# UBXN1 maintains ER proteostasis and represses UPR activation by modulating translation

Brittany A Ahlstedt [1,3], Rakesh Ganji [1], Sirisha Mukkavalli [1,4], Joao A Paulo [2], Steve P Gygi[2] & Malavika Raman [1✉]

## Abstract

**ER protein homeostasis (proteostasis) is essential for proper folding and maturation of proteins in the secretory pathway. Loss of ER proteostasis can lead to the accumulation of misfolded or aberrant proteins in the ER and triggers the unfolded protein response (UPR). In this study, we find that the p97 adaptor UBXN1 is an important negative regulator of the UPR. Loss of UBXN1 sensitizes cells to ER stress and activates the UPR. This leads to widespread upregulation of the ER stress transcriptional program. Using comparative, quantitative proteomics we show that deletion of UBXN1 results in a significant enrichment of proteins involved in ER-quality control processes including those involved in protein folding and import. Notably, we find that loss of UBXN1 does not perturb p97-dependent ER-associated degradation (ERAD). Our studies indicate that loss of UBXN1 increases translation in both resting and ER-stressed cells. Surprisingly, this process is independent of p97 function. Taken together, our studies have identified a new role for UBXN1 in repressing translation and maintaining ER proteostasis in a p97 independent manner.**

**Keywords** Unfolded Protein Response; Ubiquitin; ER Stress; Translation; p97
**Subject Categories** Post-translational Modifications & Proteolysis; Translation & Protein Quality

## Introduction

The ER is a major site for protein maturation and oversees the folding and secretion of one-third of the cellular proteome. A tight equilibrium exists between the protein folding capacity of the ER and protein load to maintain ER proteostasis. Alterations in this equilibrium due to various physiological conditions, such as disruption of ER-calcium homeostasis, mutations that impair protein folding, or cellular aging can result in protein misfolding (Kaufman, 2002). ER stress is elicited by protein misfolding and aggregation and can become pathological if not alleviated. Indeed, protein misfolding and aggregation are hallmarks of many age-associated neurodegenerative disorders. Numerous studies indicate that in neurodegenerative disorders such as Parkinson's Disease (PD) and amyotrophic lateral sclerosis (ALS), age-associated protein aggregation in the ER is causally linked to cell death (Bellucci et al, 2011; Duennwald and Lindquist, 2008; Atkin et al, 2008; Hoozemans et al, 2009). Protein aggregates can inappropriately interact with ER chaperones, disrupting ER morphology and vesicular trafficking to the Golgi apparatus (Nishitoh et al, 2008; Cooper et al, 2006; Yang et al, 2010). In the context of these age-related disorders, it is known that a decline in the efficiency and capacity of protein quality control systems diminishes ER-protein homeostasis. However, it is currently unclear how decline in specific protein quality control mechanisms impacts cell health. Recent studies suggest that aging cells rewire chaperone subnetworks and find that proteasome and autophagy capacities are altered (Vonk et al, 2020; Brehme et al, 2014; Rousseau and Bertolotti, 2016). Thus, a thorough understanding of protein quality control mechanisms is warranted to comprehend how their decline triggers disease onset.

Distinct ER-quality control mechanisms sense ER stress. These include the unfolded protein response (UPR) and ER-associated degradation (ERAD), which function to improve the folding capacity of the ER, and clear misfolded proteins from the ER through the ubiquitin-proteasome system respectively. The UPR is an adaptive ER-quality control pathway comprised of three, parallel, ER-resident transmembrane stress sensors that monitor the misfolded protein burden within the ER. These sensors include activating transcription factor 6 (ATF6), inositol requiring enzyme 1 (IRE1), and protein kinase R (PKR)-like ER kinase (PERK). Signaling pathways downstream of each sensor ultimately activates a distinct transcription factor that induces the expression of a suite of UPR-specific target genes, encoding chaperones and protein degradation machinery (Shoulders et al, 2013). The prevailing theory suggests that the ER sensors are held in an inactive state when their luminal domains are bound to the abundant ER chaperone, binding immunoglobulin protein (BiP) (also known as glucose-regulated protein 78 kDa, GRP78) (Bertolotti et al, 2000). When misfolded proteins accumulate, BiP is titrated away from the sensors, triggering their activation (Kopp et al, 2019;

[1]Department of Developmental Molecular and Chemical Biology, Tufts University School of Medicine, Boston, MA, USA. [2]Department of Cell Biology Harvard Medical School, Boston, MA, USA. [3]Present address: ALPCA diagnostics, Salem, NH, USA. [4]Present address: Dana Farber Cancer Research Institute, Boston, MA, USA.
✉E-mail: malavika.raman@tufts.edu

Pincus et al, 2010). Under these conditions, ATF6 traffics from the ER to the Golgi apparatus where site-1 and site-2 proteases cleave the cytosolic domain of ATF6 to generate an active transcription factor. In contrast, PERK dimerizes upon activation and phosphorylates eukaryotic initiation factor 2α (eIF2α), diminishing the translation initiation of most proteins (Harding et al, 2000b). In response to eIF2α phosphorylation, the translation of activating transcription factor 4 (ATF4) is selectively induced in response to reinitiation of ribosomes at the upstream atf4 coding region. Lastly, IRE1 forms oligomers and mediates the unconventional mRNA splicing of an intron from the x-box binding protein-1 (xbp1) mRNA to generate the active transcription factor xbp1s (Calfon et al, 2002; Yoshida et al, 2001). In addition, IRE1 degrades ER-targeted mRNAs in a process termed regulated IRE1-dependent decay (RIDD) (Hollien et al, 2009). The rapid degradation of mRNAs through RIDD functions to alleviate ER stress by preventing the continued influx of new polypeptides into the ER (Lipson et al, 2008; Oikawa et al, 2007). If these mechanisms fail to alleviate ER stress, PERK and IRE1 signaling will induce apoptosis through expression of the pro-apoptotic transcription factor C/EBP homologous protein (CHOP) and JNK signaling respectively (Ghosh et al, 2012; Urano et al, 2000; Kato et al, 2012).

The UPR works in tandem with ERAD, which purges terminally misfolded clients from the ER by the ubiquitin-proteasome system. During ERAD, misfolded ER proteins are recognized and handed off to ER membrane resident E3 ligases to initiate ubiquitin-dependent degradation by cytosolic proteasomes. The AAA+ ATPase p97 (also known as valosin-containing protein, VCP) is a critical intermediate in this process that recognizes ubiquitylated proteins and retro-translocates them from the ER to the cytosol prior to proteasomal degradation. The p97 homo-hexamer utilizes ATP hydrolysis to unfold substrates (Twomey et al, 2019; Pan et al, 2021; Bodnar and Rapoport, 2017) and participates in an array of cellular pathways from the cell cycle to autophagy (Meyer and Weihl, 2014; Ye et al, 2003). p97 interacts with a variety of dedicated adaptor proteins that facilitate substrate specificity and targeting of p97. The largest family of p97 adaptors contain a ubiquitin regulatory X (UBX) domain that enables binding to p97 (Raman et al, 2015). Many of the UBX domain-containing adaptors also harbor a ubiquitin-associated domain (UBA) that associates with polyubiquitin chains on client proteins. There are ~40 adaptors, though the functions and substrates of most remain poorly characterized. Studies have shown that p97 mutation can disrupt p97-adaptor binding which has been causally linked to multisystem proteinopathy 1 (MSP-1), a degenerative disorder wherein individuals present with inclusion body myopathy, Paget's disease of the bone and/or frontotemporal dementia (Buchberger et al, 2015; Zhang et al, 2015; Weihl et al, 2006, Watts et al, 2004; Schroder et al, 2005). More recently, p97 mutation has been linked to the development of familial ALS (fALS), PD, and Charcot-Marie-Tooth disease type IIB (CMTIIB) (Tang and Xia, 2016; Gonzalez et al, 2014, Gonzalez-Perez et al, 2012; Chan et al, 2012). A full understanding of adaptor function is required to better understand how their dysfunction disrupts protein quality control pathways to contribute to disease pathogenesis.

In a previous study, we reported that the p97 adaptor UBX domain protein 1 (UBXN1) associated with the multifunctional BCL2-associated athanogene 6 (BAG6) chaperone. BAG6 facilitates the insertion of tail-anchored proteins into the ER (Ganji et al, 2018;

Hegde and Keenan, 2011; Hegde and Zavodszky, 2019; Guna and Hegde, 2018; Rodrigo-Brenni et al, 2014; Mariappan et al, 2010). When clients fail to insert (due to mutations, saturation of insertion machinery at the ER, or ER stress), BAG6 recruits the E3 ligase RNF126 for ubiquitylation and degradation of the substrate (Rodrigo-Brenni et al, 2014). We showed that UBXN1 was important for the recruitment of p97 to the BAG6 triage complex to facilitate extraction and degradation of non-inserted substrates. In addition, BAG6 has also been reported to function in ERAD in concert with p97 by acting as a chaperone 'holdase' for unfolded p97 clients (Wang et al, 2011a; Ernst et al, 2011; Claessen and Ploegh, 2011). However, our studies indicated that UBXN1 is not a ERAD adaptor for p97 (Ganji et al, 2018). Here, we have identified a novel role for UBXN1, as a regulator or ER proteostasis. In this study, we demonstrate that cells depleted of UBXN1 have elevated UPR activation both in unperturbed and ER-stressed cells. Using quantitative proteomics, we find a significant increase in the abundance of ER proteins in UBXN1 deleted cells. Increased steady state protein levels are not due to the participation of UBXN1 in p97-dependent ERAD. We further demonstrate that loss of UBXN1 increases protein translation in a p97-independent manner and this contributes to ER stress. Our studies have identified a new regulator of protein translation and ER proteostasis.

## Results

### UBXN1 depletion induces activation of the unfolded protein response

Based on our prior study on p97-UBXN1 regulation of BAG6-associated protein triage, we presumed that one consequence of UBXN1 loss may be imbalance in the ER proteome and subsequent ER stress. Indeed, loss of p97 or BAG6 function can cause the accumulation of misfolded proteins in the ER via multiple mechanisms to activate the UPR. Silencing BAG6 with siRNA or inhibiting p97 with the ATP-competitive inhibitor CB-5083 caused robust activation of the UPR as expected by immunoblotting for the ER chaperone BiP and transcription factor ATF4 (Fig. EV1A–E). To assess if UBXN1 was also involved in ER stress responses, we systematically assessed UPR activation in UBXN1 knock-out (KO) HeLa Flp-In T-REx (HFT) cell lines generated by CRISPR-Cas9 gene editing. Wildtype and UBXN1 KO cells were treated with the reducing agent dithiothreitol (DTT), and lysates were probed for the ER chaperone BiP, the transcription factor ATF4, and phosphorylation of eIF2α (Fig. 1A,B). In wild-type cells, a time-dependent increase in these proteins was observed, however, a significant increase was apparent at earlier time points in UBXN1 KO cells. Notably, these markers were already upregulated in untreated UBXN1 KO cells (Fig. 1A, compare lane 5 and 1). We next monitored the levels of cleaved ATF6 in wildtype and UBXN1 KO cells treated with DTT. Cells were co-treated with the proteasome inhibitor Bortezomib (BTZ) to prevent turnover of and facilitate visualization of the N-terminus of ATF6 (Hong et al, 2004). Compared to control cells, there was a modest but significant increase in cleaved ATF6 in UBXN1 KO cells in response to ER stress (Fig. 1C,D). Under ER stress conditions, IRE1α is activated by dynamic clustering in the ER membrane (Belyy et al, 2020; Li et al, 2010). IRE1α monomers cluster into higher-order oligomers

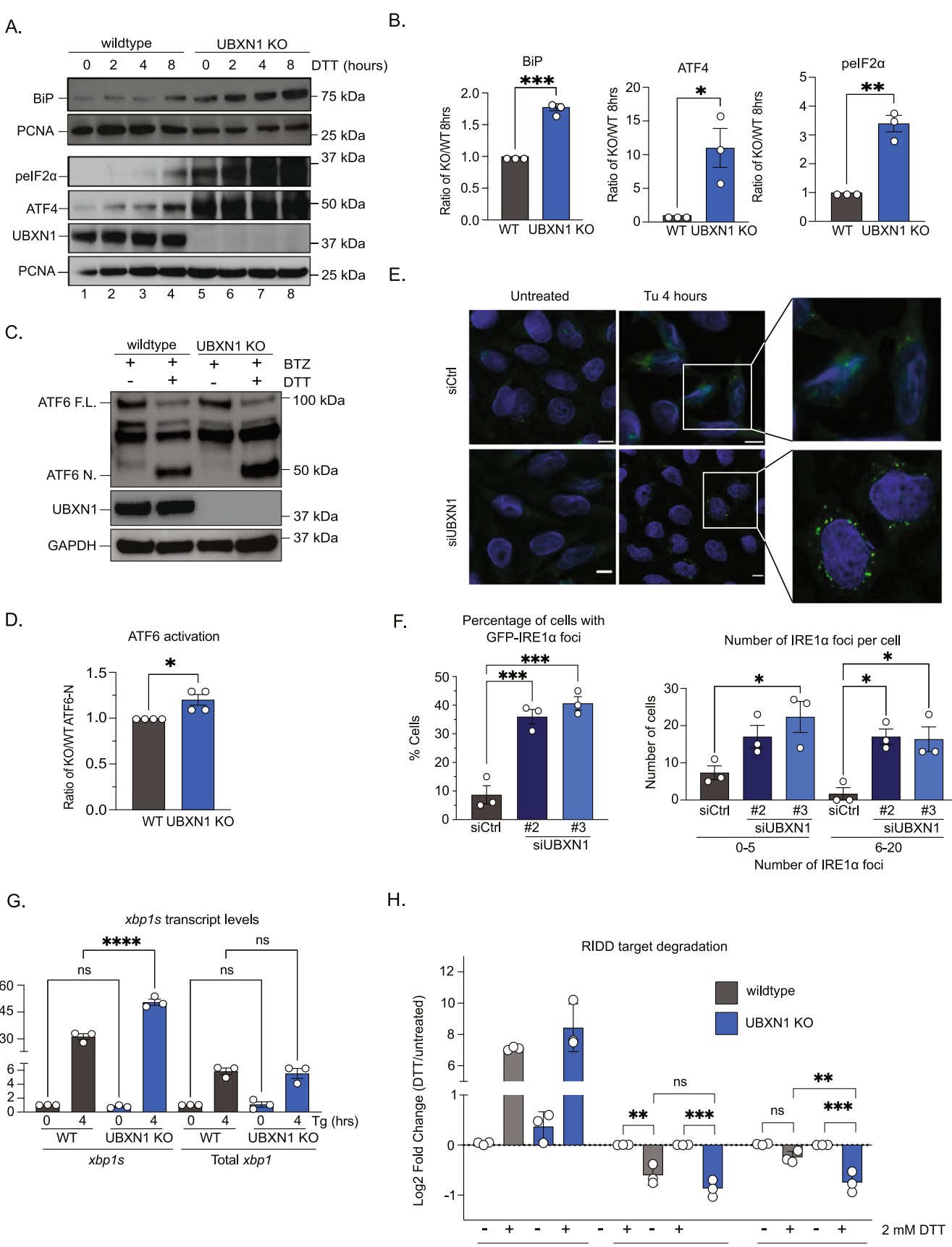

Figure 1. UBXN1 depletion induces activation of the unfolded protein response.

(A) Immunoblot of HFT wild-type and UBXN1 KO cells treated with 1.5 mM dithiothreitol (DTT) for the indicated timepoints (0–8 h). (B) Ratio (UBXN1 KO/wildtype) of the band intensity quantifications of BiP, ATF4, and peIF2α at the 8-h timepoint corresponding to A. (n = three biologically independent samples). We only provide the 8-h quantification as the signal is not always present in wild-type cells in earlier timepoints for accurate quantification. (C) Immunoblot of HFT wild-type and UBXN1 KO cells co-treated with 1.5 mM DTT and 1 μM Bortezomib (BTZ). (D) ATF6 activation was measured by band intensity quantification and calculation of the percentage of cleaved ATF6 to total ATF6. The ratio of the percentage of ATF6 activation in UBXN1 KO cells to wildtype is reported. (n = four biologically independent samples). (E) Representative immunofluorescent image of IRE1α clustering in a stable HFT cell line expressing GFP-tagged IRE1α. UBXN1 was depleted by siRNA transfection and cells were treated with 2.5 μM tunicamycin (Tu) for 4 h to induce IRE1α clustering (Scale bar: 10 μm). (F) Quantification corresponding to E reporting the percent of cells with GFP-IRE1α foci as well as the number of GFP-IRE1α foci per cell (n = three biologically independent samples). (G) Transcript levels of xbp1s and total xbp1 in HFT wild-type and UBXN1 KO cells quantified by quantitative real-time PCR. Cells were treated with 10 nM thapsigargin (Tg) for 4 h as indicated. (n = three biologically independent samples). H Transcript levels of xbp1s, bloc1s1, and cd59 were quantified by quantitative real-time PCR. Cells were treated with 2 mM DTT for 4 h where indicated. (n = three biologically independent samples). Data information: Data are means ± SEM (*, **, ***, **** where P < 0.05, 0.01, 0.001, and 0.0001, respectively.) Unpaired two-tailed t test (B, D) or one-way ANOVA with Tukey's multiple comparisons test (F–H). Source data are available online for this figure.

that can be imaged and quantified by immunofluorescence. We generated a stable cell line using a previously published doxycycline-inducible GFP-tagged IRE1α reporter to observe IRE1α clustering (Li et al, 2010). Utilizing this cell line, IRE1α clustering into oligomers was quantified by counting the number of GFP puncta. UBXN1 was depleted with two independent siRNAs and cells were treated with the ER stressor tunicamycin (Tu, inhibits the first step in N-linked glycosylation) for 4 h before cells were imaged for GFP-foci (Fig. 1E). We found that in response to ER stress, 40% of cells depleted of UBXN1 contained GFP-IRE1α foci compared to the 10% observed in cells transfected with a control siRNA (Fig. 1F). Furthermore, there were significantly more GFP-foci per cell in UBXN1-depleted cells compared to control after ER stress (Fig. 1F, right panel). We also measured IRE1α activation via downstream xbp1s expression in UBXN1 KO cells by quantitative real-time PCR using a primer set that spans the splice site removed by IRE1 (Van Schadewijk et al, 2012). We detected 19-fold greater xbp1s expression in UBXN1 KO cells compared to control in response to thapsigargin (Tg), that inhibits the ER Ca$^{2+}$ ATPase while total xbp1 levels remained unchanged (Fig. 1G). A similar phenotype was observed in a HEK-293T CRISPR-Cas9 generated UBXN1 KO cell line (Fig. EV1F). We also determined whether the RIDD activity of IRE1α was affected by loss of UBXN1. We measured bona fide RIDD target mRNAs bloc1s1 (Bright et al, 2015; Hollien et al, 2009) and cd59 (Oikawa et al, 2007) by quantitative real-time PCR and observed a trend toward an increase in bloc1s1 degradation and a significant increase in cd59 degradation in UBXN1 KO cells compared to wildtype (Fig. 1H).

To ensure that phenotypes observed in CRISPR generated KO cells were not due to adaptation with long-term cell culture, we repeated all UPR assays by acute depletion of UBXN1 and observed similar phenotypes (Fig. EV1G–J). Notably, the repression of UPR activation by UBXN1 is not due to a role for UBXN1 as a p97 adaptor for ERAD (Ganji et al, 2018; Lalonde et al, 2011). Using a panel of GFP-tagged ERAD reporters we have previously shown that p97 and its adaptor UBXD8 but not UBXN1 is required for the turnover of multiple ERAD substrates (Ganji et al, 2018).

Our group recently reported that the p97-UBXN1 complex was required for recognizing ubiquitylated protein aggregates and consolidating them in a perinuclear structure known as the aggresome (Mukkavalli et al, 2021). ERAD substrates can also be recruited to these aggregates that can associate with the ER membrane to cause ER stress. To determine whether UPR activation caused by UBXN1 loss was due to the accumulation of

ubiquitylated aggregates in ER-stressed cells, we treated wild-type and UBXN1 KO cells with tunicamycin (Tu) or thapsigargin (Tg). Cells were also treated with Bortezomib to induce aggresomes as a positive control. We visualized ubiquitylated proteins by microscopy and immunoblotting. Bortezomib-treated cells contained a single ubiquitin-positive aggresome that was disrupted in UBXN1 KO cells as we previously reported (Mukkavalli et al, 2021). However, wild-type and UBXN1 KO cells treated with Tu or Tg had no ubiquitin positive aggregates or overt accumulation of ubiquitylated proteins (Fig. EV2A,B). Thus, UPR activation upon UBXN1 deletion is not due to mishandling of aggregates destined for the aggresome. Taken together, our studies suggest a new role for UBXN1 as a repressor of UPR activation in response to distinct ER stress triggers that is independent of ERAD.

## Loss of UBXN1 activates an ER stress-related transcriptional program leading to cell death

The UPR ultimately terminates in a transcriptional response downstream of ATF4, ATF6-N, and XBP1s, therefore, we analyzed UPR target gene expression using quantitative real-time PCR of 84 UPR-specific target genes (Dataset EV1). Gene expression analysis of wild-type and UBXN1 KO cells identified a global upregulation (88%, 74 of 84 targets) of UPR-specific target genes in UBXN1 KO cells with 40 genes (47%) experiencing a log$_2$ fold change ≥ 0.5 (Fig. 2A). While this appears to be a modest change, previous microarray analysis has demonstrated that only a handful (22 out of 1868 analyzed) of UPR related genes have twofold induction in HeLa cells stressed with tunicamycin (Okada et al, 2002). Of the genes that demonstrated a log$_2$ fold change ≥ 0.5, there are known targets of each of the UPR sensors, including PERK (ddit3/chop), ATF6 (calreticulin, (calr) and herpud1), and IRE1α (sec62 and edem3), among others (Shoulders et al, 2013; Gonen et al, 2019). Hierarchical clustering analysis demonstrates that the gene expression pattern observed in untreated UBXN1 KO cells more closely resembles wild-type cells stressed with DTT than untreated wild-type cells based on similar log$_2$ fold change values (Fig. 2A). We find that the transcriptional response in UBXN1 KO cells treated with DTT is comparable to wild-type cells treated with DTT (Fig. 2A). This is possibly due to a ceiling effect that is often seen with UPR target gene expression upon ER stress (Shinjo et al, 2013; Travers et al, 2000). Interestingly, many of the downregulated transcripts are ERAD components (nploc4 and hspa1b). Accumulating evidence illustrates that selective activation of PERK, IRE1α,

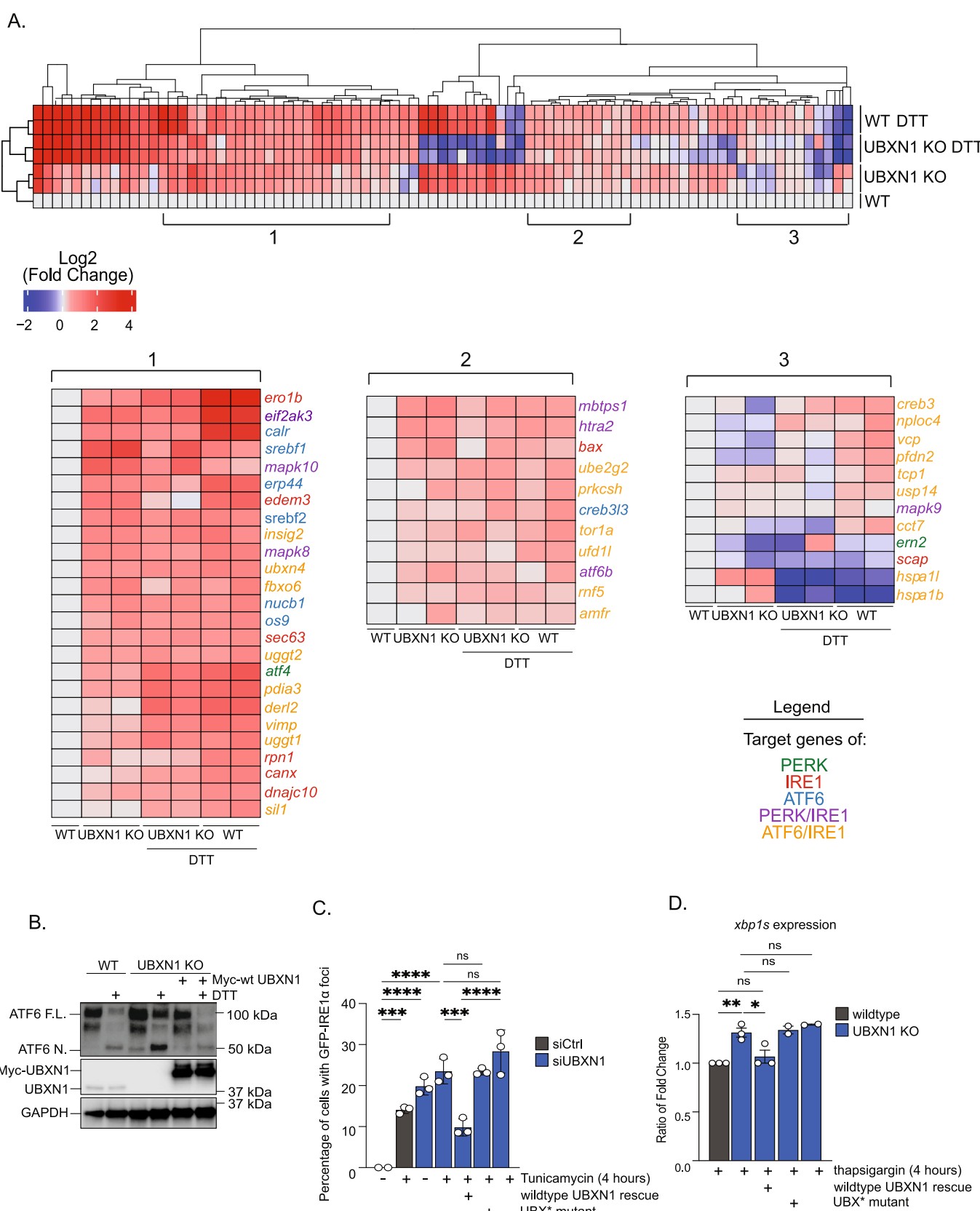

◀ **Figure 2.   Loss of UBXN1 activates an ER stress transcriptional program.**

(A) Heat map displays the hierarchical clustering of the $\log_2$ fold change of 84 UPR target genes determined by the Human Unfolded Protein Response real-time PCR profiler array. The range is from a $\log_2$ fold change of -2 (blue) to a $\log_2$ fold change of 4 (red). The untreated wild-type column is displayed as an average of biological duplicates with the $\log_2$ fold change set to zero for this column. Additional columns represent biological duplicates of untreated UBXN1 KO cells, biological duplicates of UBXN1 KO cells treated with 1.5 mM DTT, and biological duplicates of HFT wild-type cells treated with 1.5 mM DTT for 4 hours respectively. Section 1 and 2 represent UPR target genes that experience increased expression in the untreated UBXN1 KO cells as well as UBXN1 KO and wild-type cells treated with DTT. Section 3 represents a set of UPR target genes that experience decreased expression in all samples compared to untreated wild-type samples. Genes are color coded by the upstream UPR stress sensor that induces their expression: PERK (green); IRE1 (red); ATF6 (blue); PERK and IRE1 (purple); ATF6 and IRE1 (orange). ($n =$ two biologically independent samples). (B) Rescue of ATF6 processing (ATF6-N) in UBXN1 KO cells by re-expression of Myc-tagged wild-type UBXN1 ($n =$ three biologically independent samples). (C) Quantification of the percent of cells with GFP-IRE1α foci in control cells and cells depleted of UBXN1 with siRNA. Myc-tagged wildtype, UBX point mutant, and UBA point mutant were expressed into UBXN1-depleted cells before treatment with 2.5 μM tunicamycin (Tu) for 4 h to induce IRE1α clustering. ($n =$ three biologically independent samples). (D) Ratio of the relative expression of *xbp1s* in wild-type and UBXN1 KO cells treated with 10 nM thapsigargin for 4 hours. Myc-tagged wildtype, UBX point mutant, and UBA point mutant were expressed into UBXN1-depleted cells for 48 h before thapsigargin treatment. ($n \geq$ two biologically independent samples). Data information: Data are means ± SEM (*, **, ***, **** where $P < 0.05$, 0.01, 0.001, and 0.0001, respectively.) One-way ANOVA with Šídák's multiple comparisons test (C) and Tukey's multiple comparisons test (D). Source data are available online for this figure.

or ATF6 is induced by distinct physiological stimuli, but activation of all three UPR arms may indicate more widespread protein misfolding within the ER that impairs ER-proteostasis. Thus, it is possible that loss of UBXN1 promotes a transcriptional response favoring an environment conducive to protein folding rather than degradation. However, a more comprehensive, global analysis using RNA-seq is needed to investigate this further.

We next asked if we could rescue UPR activation in UBXN1 loss of function cells by complementing cells with wild type UBXN1. Indeed, ATF6 processing and IRE1 activation in UBXN1 KO cells could be restored to wild-type levels by re-expression of UBXN1 (Fig. 2B–D). We next asked if ubiquitin interaction or p97 recruitment by UBXN1 were important for UPR activation. For this, we focused our efforts on IRE1α activation (GFP-IRE1α foci formation and *xbp1* splicing) as these assays are robust and quantitative. Re-expression of wild-type UBXN1 rescued IRE1α foci formation (Fig. 2C) and *xbp1* splicing (Fig. 2D) back to wild-type levels. However expression of UBXN1 with point mutations in the UBX domain (Phe[265] Pro[266] Arg[267] truncated to Ala-Gly) which we have demonstrated previously to disrupt interaction with p97 (Mukkavalli et al, 2021), as well as a UBA domain mutant (Met[13] and Phe[15] to Ala) that fails to interact with ubiquitin (Ganji et al, 2018; Mukkavalli et al, 2021) failed to rescue the phenotype (Fig. 2C,D). These findings suggest both p97 and ubiquitin interaction are important to repress UPR.

## UBXN1 deletion sensitizes cells to ER stress with consequent reduction in cellular viability

Next, we asked whether UPR activation upon UBXN1 loss impacted cell viability in response to ER stress. UPR signaling during acute ER stress is a protective mechanism, whereas chronic ER stress and UPR activation primes cells toward apoptosis (Harding et al, 2003; Han et al, 2013; Guan et al, 2014; Krokowski et al, 2013). First, we assessed recovery from ER stress in wild-type and UBXN1 KO cells. Cells were treated with thapsigargin for a time-course spanning 30 min to 3 h and released into drug-free media (Fig. 3A). Viable cells remaining after three days were stained with crystal violet and quantified. We observed significantly reduced growth of UBXN1 KO cells relative to wild-type cells suggesting that elevated UPR signaling primes UBXN1 KO cells toward cell death (Fig. 3A,B). We corroborated these studies using a fluorescence-based cytotoxicity assay. Cells were treated with

thapsigargin, and fluorescent readings were taken 24-, 48-, and 72-h post treatment. Again, we observed a significant increase in cell death in UBXN1 KO cells at 48- and 72-h compared to wildtype (Fig. 3C). We note that UBXN1 KO cells are as viable as wildtype in the untreated condition. In agreement with this finding, we observed that UBXN1 KO cells have elevated levels of the apoptosis-inducing transcription factor C/EBP homologous protein (CHOP) compared to wild-type cells in response to DTT (Fig. 3D,E). Thus, we conclude that UBXN1 loss is detrimental to cell viability and primes cells toward cell death in the face of ER stress.

## Loss of UBXN1 leads to a perturbed ER proteome

To understand how UBXN1 loss causes ER stress, we first asked if UBXN1 was associated with the ER. We isolated ER-derived microsomes by biochemical fractionation from HEK293T cells and subjected the microsomes to proteinase K digestion in the presence and absence of detergent (Triton X-100) (Fig. EV2C). As previously reported, a significant fraction of p97 is peripherally associated with the ER to regulate ERAD (Ye et al, 2005, 2004) and this localization is sensitive to Proteinase K (Fig. EV2C). Similarly, we found that a fraction of UBXN1 was also associated with the ER membrane and was readily digested in Proteinase K treated samples (Fig. EV2C). We next asked if UBXN1 associated with the SEC61 translocon to regulate import of proteins into the ER, however in co-immunoprecipitation studies we were unable to discern an interaction between UBXN1 and the translocon components SEC61α or β whereas UBXN1 interacted with p97 as expected (Fig. EV2D,E).

We decided to take an unbiased approach to assess how UBXN1 deletion impacted ER protein homeostasis using quantitative tandem mass tag (TMT) proteomics. Peptides derived from duplicate wild-type and UBXN1 KO cells were labeled with four-plex TMT labels, combined and peptide abundance was quantified by liquid chromatography and mass spectrometry (Fig. 4A). We quantified a total of 6,673 proteins from this study of which 53 were significantly enriched in UBXN1 KO cells ($\log_2$ fold change KO:WT $\geq 1.0$ and $p \leq 0.05$) (Fig. 4B, Dataset EV2). Strikingly, gene ontology analysis identified significant enrichment of ER proteins involved in protein folding, ER-quality control, and the response to ER stress suggestive of UPR activation (Fig. 4B and Fig. EV3A,B). These include proteins involved in *N*-linked glycosylation

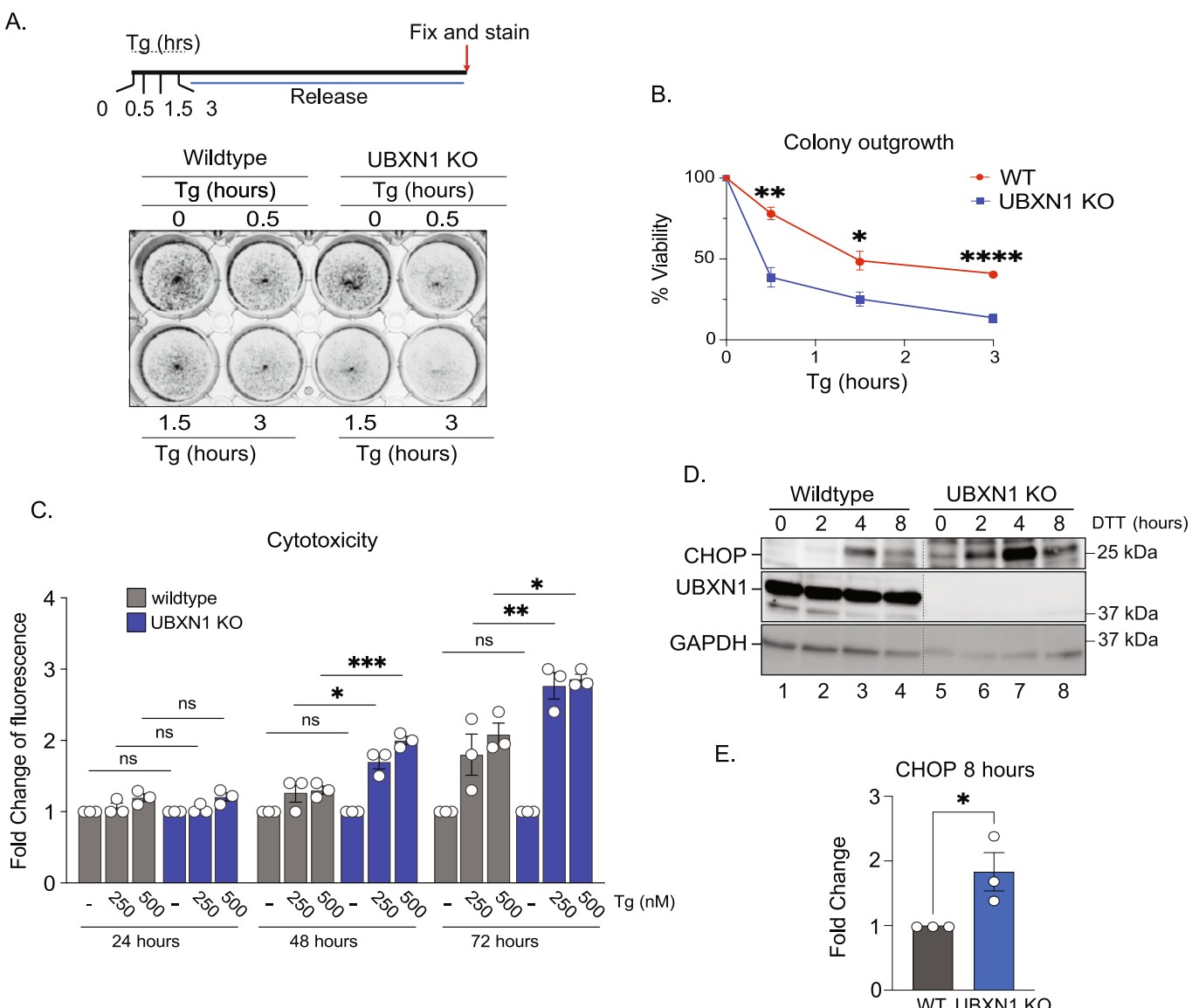

**Figure 3. UBXN1 deletion sensitizes cells to ER stress with consequent reduction in cellular viability.**

(A) Recovery from ER stress was determined by crystal violet staining. Wild-type or UBXN1 KO cells were treated with 200 nM thapsigargin (Tg) for the indicated timepoints and released into drug free media. Colonies were allowed to grow out for three days and live cells were stained with crystal violet. (B) Percent recovery calculated from the remining cells in **A**. (n = four biologically independent samples). (C) Fold change of the fluorescence identified by fluorescence-based cytotoxity assay. Wild-type and UBXN1 KO cells were treated with either 500 or 250 nM thapsigargin for 24-, 48-, or 72-h with fluorescent readings taken every 24-h. Values were normalized to untreated for each genotype. Note we observe no differences in viability between wild-type and UBXN1 KO cells in untreated conditions allowing for this normalization (n = three biologically independent samples). (D) Immunoblot of HFT wild-type and UBXN1 KO cells treated with 1.5 mM dithiothreitol (DTT) for the indicated timepoints (0–8 h). (E) Band intensity quantifications of CHOP expression at the 8-h timepoint corresponding to **C**. (n = three biologically independent samples). We only provide the 8-h quantification as the signal is not present in wild-type cells in earlier timepoints for accurate quantification. Data information: Data are means ± SEM (*, **, ***, **** where $P < 0.05$, 0.01, 0.001, and 0.0001, respectively.) Unpaired two-tailed t test (**B**, **E**) or one-way ANOVA with Tukey's multiple comparisons test (**C**). Source data are available online for this figure.

(glycosyltransferase enzymes ALG1, 3, and 5), ERAD (HRD1 E3 ligase adaptor SEL1L and the ER-anchored protein escortase TMUB1 (Wang et al, 2022)), and ER import (components of the heterotrimeric SEC61 complex: SEC61A1, SEC62, and SEC63 and the ER membrane complex, EMC) (Fig. EV3A). We also found increased abundance of diverse ER membrane proteins including: fatty acid elongase 1 (ELOVL1), cytochrome B-245 chaperone 1 (C17orf62), and Glucose-6-phosphatase catalytic subunit 3

(G6PC3), as well as several secreted proteins including the Type I and IX collagens (COL1A2 and COL9A1), indicating that increased abundance of TMT hits is not solely due to a stress-induced UPR transcriptional program alone (Fig. 4C). Only ER proteins were selectively increased in abundance in UBXN1 null cells and proteins resident in other organelles were largely unaffected (Fig. EV3C). In contrast, mitochondrial proteins were depleted in UBXN1 KO cells (Figs. 4C and EV3C, explored further below).

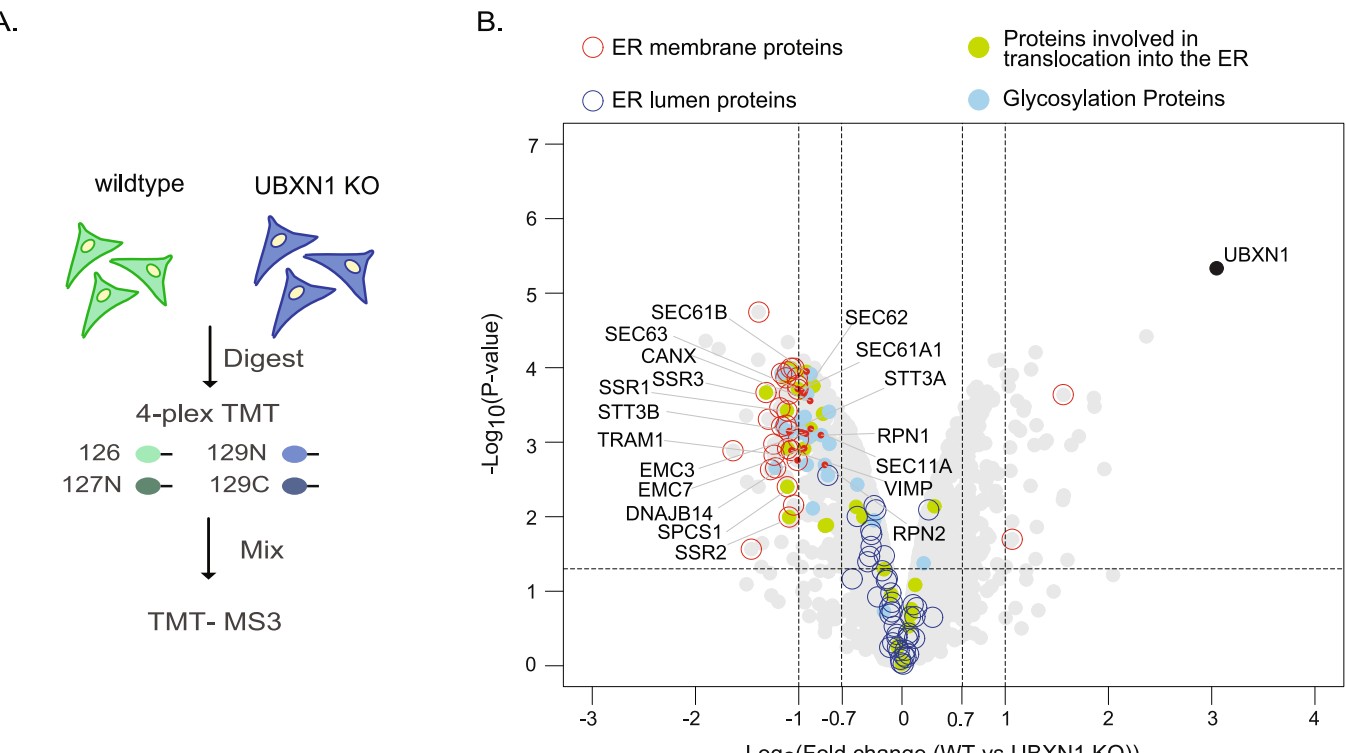

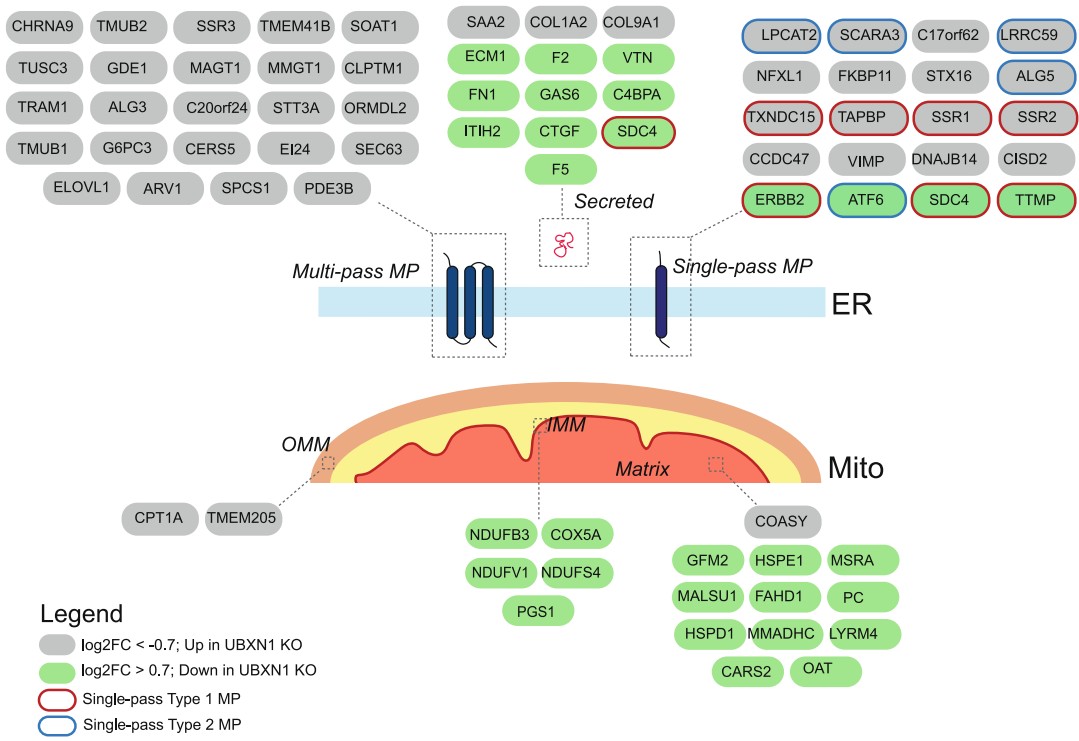

◄

**Figure 4. Loss of UBXN1 leads to a perturbed ER proteome.**

(A) Schematic workflow of quantitative tandem mass tag (TMT) proteomics in HFT wild-type and CRISPR-Cas9 UBXN1 knock-out (KO) cells. (B) Volcano plot of the ($-\log_{10}$-transformed $P$ value versus the $\log_2$-transformed ratio of wildtype/UBXN1 KO) proteins identified by TMT proteomics in wild-type and UBXN1 KO cells. A negative $\log_2$ fold change indicates proteins with increased abundance in UBXN1 KO cells. Proteins with a $\log_2$ fold change $\geq |1|$ are considered significant. ER membrane and luminal proteins are outlined by a red and blue circle respectively. Proteins involved in translocation into the ER and glycosylation are colored in green and blue respectively ($n = 2$ biologically independent samples for each genotype). (C) Clients in the secretory pathway and mitochondrial proteins identified by proteomics. Proteins were characterized by ER-targeting signal (transmembrane domain or signal sequence) and then further characterized into single-pass Type 1 (red outline) or Type 2 (blue outline) and multi-pass membrane protein. Mitochondrial proteins were further characterized into outer- and inner- mitochondrial membrane and matrix proteins. Proteins enriched in UBXN1 KO cells are in gray and those depleted in UBXN1 KO cells are in green.

Thus, we demonstrate that UBXN1 localizes to the ER and loss of UBXN1 alters the ER proteome.

## Depletion of UBXN1 increases the expression and aggregation of ER-destined proteins without perturbing their degradation

We validated hits from our proteomic studies by immunoblot and observed a significant increase in the abundance of the translocon-associated protein α (TRAPα) and alkaline phosphatase placenta-like 2 (ALPP2) in cells depleted of UBXN1 with siRNA (Fig. 5A,B,G). Interestingly, we found that this phenotype extends beyond our proteomics hits. We detected a significant increase in the abundance of ER clients such as α-galactosidase (AGAL) (Sun et al, 2023) (Fig. 5C,G), and prion protein (PrP) (Rodrigo-Brenni et al, 2014) (Fig. EV4A,B, compare lanes 5 and 1), in cells transfected with the corresponding epitope-tagged constructs and depleted of UBXN1. In agreement with our proteomic studies, only ER proteins were increased in abundance and no impact on cytosolic and nuclear proteins was observed upon loss of UBXN1 (Fig. 5D–G).

We next explored why ER proteins are increased in abundance upon loss of UBXN1. We find no significant change in transcript levels of several ER-targeted proteins in UBXN1-depleted cells ruling out increased transcription (Fig. EV4C). Next, we asked if loss of UBXN1 resulted in delayed degradation of ER clients resulting in their stabilization. We monitored the half-lives of TMT hits squalene epoxidase (SQLE) and stearoyl-CoA 9-desaturase (SCD1), HA-Prp and FLAG-AGAL after translation shut off using cycloheximide. While we observed a basal increase in the abundance of these proteins in UBXN1-depleted cells (Figs. 5H and EV4D,E), the turnover of all substrates except AGAL was not appreciably different from control cells indicating that UBXN1 did not mediate their degradation (Figs. 5H,I and EV4D–H). FLAG-AGAL was slowly turned over in UBXN1 depleted cells compared to control (Fig. EV4G,H). Given our previous findings that UBXN1 facilitated the p97-dependent degradation of mislocalized BAG6 ER clients (Ganji et al, 2018) we additionally monitored the ER versus cytosolic (mislocalized) forms of AGAL and Prp as they have weak signal sequences and readily mislocalize to the cytosol (Ganji et al, 2018; Sun et al, 2023; Kang et al, 2006). We found that UBXN1 loss specifically stabilized ER-localized forms of these proteins as verified by migration status on immunoblots and EndoH treatment suggesting that the increase in ER proteins is not due to the accumulation of the cytosolic, mislocalized form (Fig. EV4A,F,G).

An increase in the abundance of proteins in the ER can outcompete chaperone availability and result in perturbed folding

and subsequent aggregation. We monitored AGAL solubility in control and UBXN1-depleted cells treated with DTT that were fractionated into soluble and insoluble fractions. In control cells, AGAL was soluble and was not found in the insoluble fraction even upon DTT treatment (Fig. 5J compare lanes 1 to 3 to 7). Strikingly, in cells depleted of UBXN1 and treated with DTT, AGAL shifted completely into the insoluble fraction (Fig. 5J compare lanes 2 to 4 and 8). Taken together, loss of UBXN1 results in the increased abundance of ER proteins independent of transcription and degradation. At least in the case of AGAL, we find that increased protein levels lead to insolubility.

## UBXN1 depletion impacts mitochondrial proteins and alters morphology

Our proteomics study identified several mitochondrial proteins depleted in UBXN1 KO cells, but no change in other organellar proteins (Figs. 4C and EV3C). We validated these hits by immunoblot and observed a significant decrease in the expression of several mitochondrial proteins including subunits of translocase of outer mitochondrial membrane complex (TOMM20, TOMM70), inner membrane complex (TIMM23 and TIMM17A), and the matrix protein complex III subunit cytochrome c1 (CYC1) when UBXN1 was depleted (Fig. 6A–E).

Numerous studies have demonstrated that contact sites between the ER and mitochondria act as sites for calcium and lipid exchange between these organelles (Hori et al, 2002; Bravo et al, 2011). ER stress drives calcium uptake into the mitochondria to elevate ATP production thereby facilitating the folding capacity of ATP-dependent ER chaperones (Chami et al, 2008). However, chronic ER stress delivers excessive calcium into mitochondria which triggers mitochondria membrane permeability, fragmentation, and causes apoptosis (Gupta et al, 2010). To prevent mitochondrial fragmentation, recent studies have found that PERK-peIF2α induced translation attenuation prompts two distinct mitochondrial protective mechanisms. First, PERK-regulated translation attenuation induces the protective degradation of the TIMM17A subunit of the TIMM23 import complex by the mitochondrial metalloprotease YME1L (Rainbolt et al, 2013) thereby inhibiting protein import into mitochondria to prevent mitochondrial dysfunction (Rainbolt et al, 2013). We observed significant depletion of TIMM17A in UBXN1-depleted cells that is further depleted upon DTT treatment (Fig. 6D,E). Whether the other mitochondrial membrane proteins we observed to be depleted (TIMM23, TOMM20, and TOMM70) are similarly downregulated downstream of eIF2α phosphorylation is currently unknown.

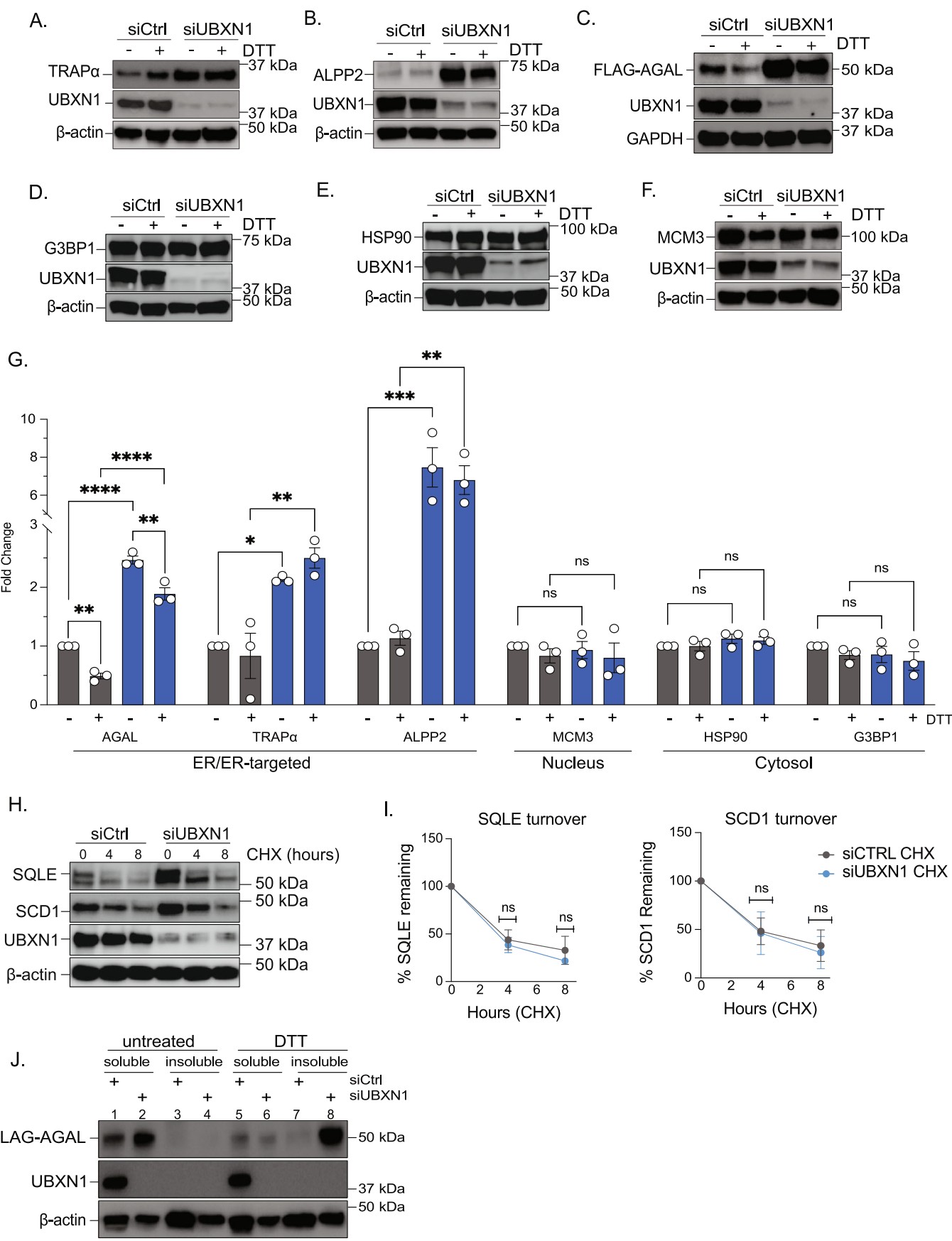

◄ **Figure 5. Depletion of UBXN1 increases the expression of clients in the secretory pathway.**

(A, B) Representative immunoblots of proteins identified to be enriched in the proteomics study. Protein levels of the ER (TRAPα) and secretory (ALPP2) proteins were measured in control cells or cells depleted of UBXN1 with siRNA. Cells were treated with 10 mM DTT for 4 h where indicated. (C) Immunoblot of a stable HFT cell line expressing doxycycline inducible FLAG-tagged α-galactosidase (AGAL). UBXN1 was depleted with siRNA before AGAL induction. Cells were treated with 10 mM DTT for 4 hours where indicated. (D–F) Representative immunoblots of cytosolic (G3BP1 and HSP90) and nuclear (MCM3) proteins measured in control or UBXN1-depleted cells. Cells were treated with 10 mM DTT for 4 hours where indicated. (G) Band intensity quantifications of protein expression corresponding to A–F. (n = three biologically independent samples). (H) Immunoblots demonstrating protein turnover of the proteomics hits (SQLE and SCD1) measured by cycloheximide chase time course in siRNA UBXN1-depleted cells. (I) Quantifications of SQLE and SCD1 turnover from H. (n = three biologically independent samples). (J) Immunoblot of soluble and insoluble fractions of FLAG-tagged AGAL in control cells or cells depleted of UBXN1 with siRNA. Cells were treated with 10 mM DTT for 4 hours where indicated. (In this experiment only, we quantified and ran 2.5x less siUBXN1 protein so we could accurately visualize and compare siControl to siUBXN1. The AGAL expression in siUBXN1 cells is much stronger than what can be observed with siControl.). Data information: Data are means ± SEM (*, **, ***, **** where P < 0.05, 0.01, 0.001, and 0.0001, respectively.) One-way ANOVA with Tukey's multiple comparisons test (G) or Unpaired two-tailed t test (I). Source data are available online for this figure.

In parallel, the PERK-peIF2α pathway also promotes ER stress-induced mitochondrial hyperfusion (SIMH) (Lebeau et al, 2018; Adachi et al, 2016; Bobrovnikova-Marjon et al, 2012; Perea et al, 2023). SIMH aids in cellular recovery after ER stress by preventing premature mitochondrial fragmentation, thereby enhancing the activity of the electron transport chain (Lebeau et al, 2018; Adachi et al, 2016; Bobrovnikova-Marjon et al, 2012; Perea et al, 2023). We evaluated mitochondrial morphology by immunofluorescence of TOMM20 in wild-type and UBXN1 KO cells treated with ER stress. Hyperfusion was quantified by measuring the length of mitochondrial networks using published image analysis scripts (Valente et al, 2017). The Complex III inhibitor, antimycin A, and ATP synthase inhibitor, oligomycin A (AO), were used as controls as they cause mitochondrial fragmentation (reduced fusion) (Fig. 6F,G). Strikingly, we observed that mitochondria were more fused in unstressed UBXN1 KO cells in a manner comparable to wild-type cells treated with ER stress (Fig. 6F,G). Furthermore, ER-stress induction further enhanced mitochondrial fusion in UBXN1 KO cells (Fig. 6F,G). Taken together, these studies suggest that increased ER stress in UBXN1 KO cells causes activation of known protective mechanisms in the mitochondria thereby leading to downregulation of mitochondrial proteins and increased mitochondrial fusion.

## UBXN1 represses protein translation

Our findings thus far suggested that loss of UBXN1 increased ER protein abundance independent of transcription and degradation. We therefore asked if UBXN1 directly impacted protein synthesis. We utilized puromycin incorporation into newly synthesized proteins as a proxy for translation (Schmidt et al, 2009). Puromycin is structurally similar to aminoacyl-tRNAs and can occupy the acceptor (A) site on a ribosome to be incorporated into the nascent polypeptide (Yarmolinsky and Haba, 1959; Nathans, 1964). Puromycilated proteins are pre-maturely released from the ribosome and can be visualized by immunoblotting with a puromycin antibody (Ravi et al, 2020). Wild-type and UBXN1 KO cells were briefly labeled (~30 min) with puromycin, and the levels of puromycilated proteins were quantified. We observed a significant increase in puromycilated proteins in UBXN1 loss of function cells suggestive of increased translation (Figs. 7A,B and EV5A). We additionally treated cells with DTT or thapsigargin to induce ER stress and shut down protein translation through eIF2α phosphorylation. DTT, thapsigargin, or cycloheximide treatment resulted in a complete shutdown of protein synthesis

in wild-type cells as evidenced by the loss in puromycin signal (Fig. 7A compare lanes 7 and 5 to 3, Fig. EV5A,B). However, although significantly reduced in UBXN1 KO cells, some translation could still be observed (Figs. 7A, and EV5A,B). To assess the role of p97 in this process, we repeated these studies in cells depleted of p97. Surprisingly, we found that p97 depletion did not impact translation (Fig. EV5C,D, compare lanes 2 to 1). In fact, thapsigargin treatment resulted in an even more robust termination of translation in p97-depleted cells compared to control (Fig. EV5C, D, compare lanes 5 to 4).

The finding that p97 was not involved in UBXN1 translation repression was somewhat surprising given that p97 adaptors are believed to be obligate p97 interactors. Therefore, we next asked whether UBXN1 mutants that cannot bind ubiquitin or p97 could rescue the translation phenotype. While transfection of wild-type UBXN1 into UBXN1 KO cells restored translation to wild-type levels (Fig. 7C,D, compare lane 3 to lane 1), so did the UBXN1 UBX domain mutant (Fig. 7C,D, compare lane 5 to 1). However, the UBA domain mutant (Fig. 7C,D compare lanes 6 to 1) did not rescue the increase in protein synthesis indicating that ubiquitin but not p97 association is important for translation repression.

To further investigate the increased translation phenotype, we isolated polysomes to directly analyze the amount of actively translating ribosomes present on mRNA. Inhibiting protein synthesis with cycloheximide halts ribosomes on mRNA such that they can be isolated by sucrose gradient centrifugation. We found that UBXN1 KO cells have an increase in the levels of actively translating polysomes compared to wild-type cells (Fig. 7E). Strikingly, while DTT treatment in wild-type cells halted protein synthesis and collapses polysomes to 80 S monosomes, translation persists in UBXN1 KO cells (Fig. 7E). Furthermore, depletion of p97 decreased polysomes in untreated cells that was further depleted upon DTT treatment supporting the puromycin studies (Figs. 7F and EV5E). To determine if UBXN1 could potentially interact with ER-bound polysomes, we analyzed the polysome fractions by immunoblot to inspect the localization of UBXN1 and p97. We found that UBXN1 co-sediments with the 40S, 60S, and 80S ribosome fractions as well as actively translating polysome fractions (Fig. EV5F). In contrast, p97 appears to associate largely with the 40S, 60S, and 80S ribosomal fractions but not with the translating polysomes (Fig. EV5F). Taken together UBXN1 represses translation in a p97-independent but ubiquitin-dependent manner.

We asked if the increase in protein translation in UBXN1 KO cells contributes to activation of the UPR. If so, we hypothesized

A.

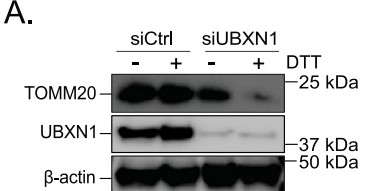

B.

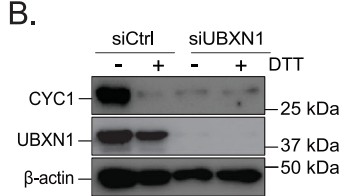

C.

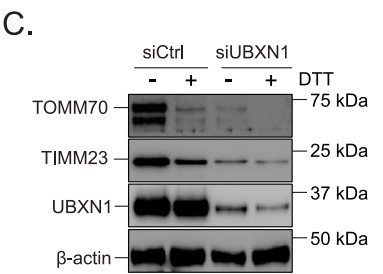

D.

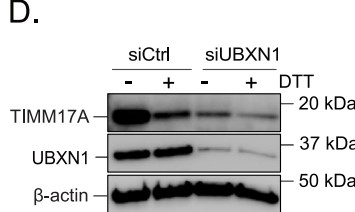

E.

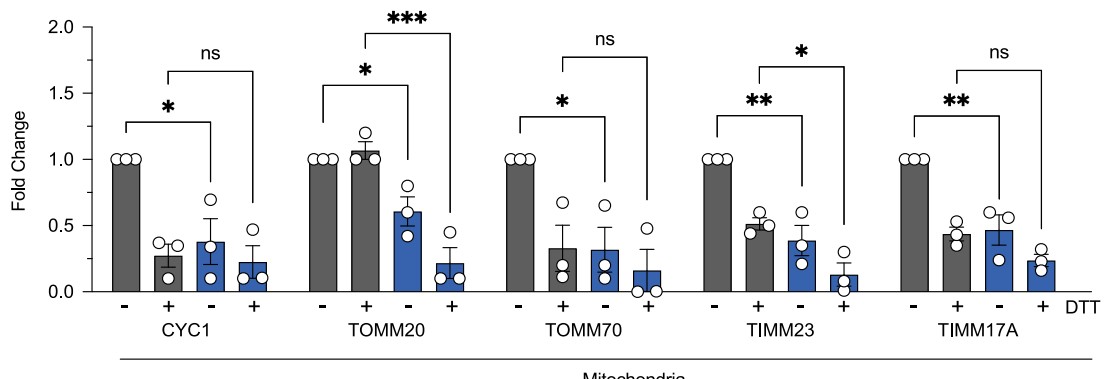

F.

G.

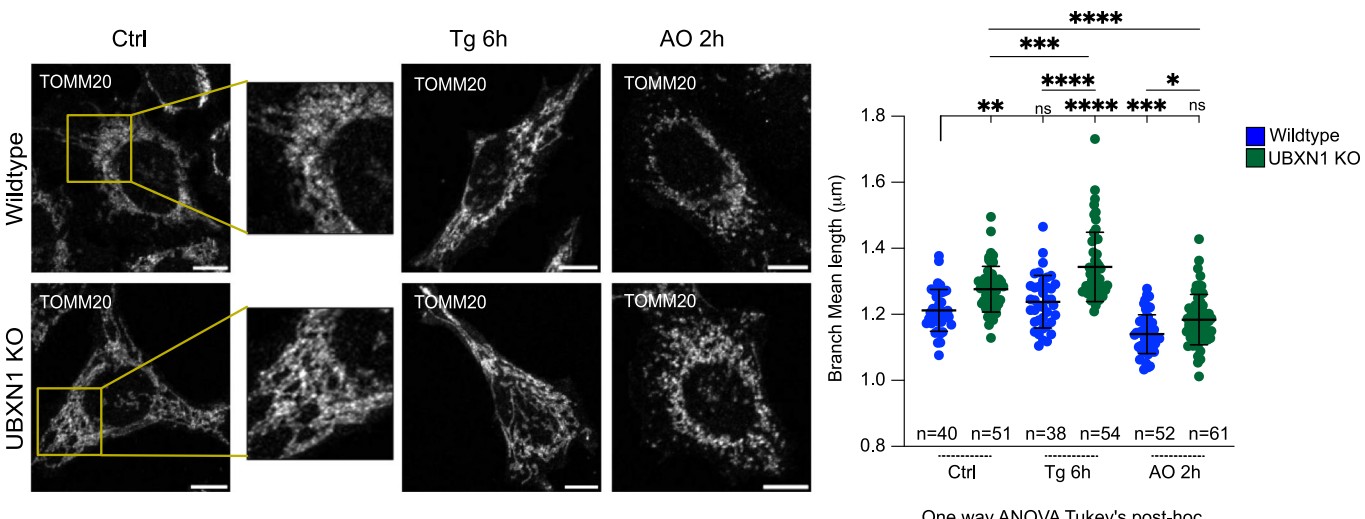

Figure 6. UBXN1 depletion impacts mitochondrial proteins and alters morphology.

A–D Representative immunoblots of mitochondrial proteins (TOMM20, CYC1, TOMM70, TIMM23, and TIMM17A) in siRNA UBXN1 depleted cells. Cells were treated with 10 mM DTT (4 h) where indicated. (E) Band intensity quantifications of protein expression corresponding to A–D (n = three biologically independent samples). (F) Representative immunofluorescent image of wild-type and UBXN1 KO cells stained with TOMM20 to image mitochondria. Cells were treated with 1.5 μM thapsigargin for 6 h or antimycin A with oligomycin A (AO) for 2 h where indicated. (Scale bar: 10 μm) (G) Quantification of the mitochondrial branch mean length (μm) of wild-type and UBXN1 KO cells from F. (n = 38–61 cells per condition as indicated in the bar graph, from a total of three biologically independent samples). Data information: Data are means ± SEM (*, **, ***, **** where P < 0.05, 0.01, 0.001, and 0.0001, respectively.) One-way ANOVA with Tukey's multiple comparisons test (E, G). Source data are available online for this figure.

that inhibiting translation should rescue UPR activation and cell death phenotypes in UBXN1 KO cells. We measured the levels of *xbp1s* in wild-type and UBXN1 KO cells treated with thapsigargin in the absence or presence of cycloheximide (Fig. 8A). As we found previously, *xbp1s* was robustly induced in UBXN1 KO cells relative to wildtype in response to thapsigargin treatment (Fig. 8A). Strikingly, co-treatment with cycloheximide reduced *xbp1s* close to wild-type levels suggesting that repressing elevated translation in UBXN1 KO cells relieves UPR activation (Fig. 8A). Furthermore, this is replicated in terms of cell viability. We found that co-treatment of thapsigargin with cycloheximide in UBXN1 KO cells reversed cell death and was protective (Fig. 8B). As noted above, UBXN1 KO cells are as viable as wild-type cells under resting conditions; only in the presence of ER stress is cell death exacerbated (Fig. EV5G). Taken together, we have identified UBXN1 as a novel repressor of protein translation. UBXN1 functions to maintain ER proteostasis and loss of UBXN1 increases ER protein abundance and UPR activation leading to reduced cell viability.

## Discussion

Cellular stress can provoke an imbalance between the protein folding machinery and ER-protein load. Numerous studies have illuminated how protein misfolding and consequent ER stress is a major disease mechanism that exacerbates the pathogenesis of a variety of disease states. Failure to adapt to ER stress has been linked to inflammation and UPR-induced apoptosis in cardiovascular disease, liver fibrosis, and diabetes mellitus (Kim et al, 2018; Okada et al, 2004; Ljubkovic et al, 2019; Ramalingam et al, 2020). Thus, it is critical to understand the protein quality control mechanisms that function to prevent ER stress and maintain ER proteostasis to prevent disease.

In the current study, we have identified a novel role for the p97 adaptor protein UBXN1 as a repressor of the UPR and translation (Fig. 8C). We found that cells devoid of UBXN1 have significant activation of pathways downstream of PERK, IRE1α, and ATF6 irrespective of the type of ER stress (Fig. 1) resulting in a transcriptional program to reset ER proteostasis (Fig. 2). Using quantitative proteomics, we observed that UBXN1 KO cells have a perturbed ER-proteome with an enrichment of proteins that include proteins involved in ER quality control as well as clients in the secretory pathway (Fig. 4). Our proteomic studies further found that loss of UBXN1 resulted in the depletion of several mitochondrial proteins. We validated these findings and further show that SIMH is upregulated in UBXN1 null cells as a compensatory mechanism to protect mitochondria from overt ER stress (Fig. 6).

How does UBXN1 maintain homeostasis of the ER proteome? Surprisingly, we find that UBXN1 is a translation repressor and loss of UBXN1 increases protein translation under basal conditions, which persists in response to ER stress (Fig. 7). This increase in abundance of secretory pathway clients can overwhelm the folding capacity of the ER to cause ER stress. Indeed, previous studies reported that aberrant import of proteins into the ER increased aggregation in response to ER stress (Kang et al, 2006). We find that loss of UBXN1 increases the propensity for AGAL aggregation in response to ER stress (Fig. 5). A more comprehensive analysis to determine whether loss of UBXN1 causes widespread aggregation of ER proteins is needed. In theory, we expected that IRE1-dependent RIDD would prevent aberrant translation into the ER by degrading ER-associated mRNAs. However, only a handful of mRNAs have been characterized as bona fide RIDD targets, which at present does not include all ER-targeted mRNAs. Along with *xbp1*, RIDD target mRNAs are characterized by a stem loop structure containing conserved residues that enhance their accessibility to the endoribonuclease domain of IRE1α (Moore and Hollien, 2015). Thus, it is unlikely that all the ER-targeted mRNAs would be degraded by RIDD before translation. Whether UBXN1 represses translation globally or just the synthesis of ER proteins is unclear at present. However, based on our proteomics and subsequent validation studies, ER proteins seem especially sensitive to loss of UBXN1 (Fig. 5). Surprisingly, we observed that the role of UBXN1 in repressing protein translation is independent of p97 function but reliant on its ability to bind ubiquitin (Fig. 7). In support of our findings, a previous study found that p97 inhibition also suppressed global translation (Moon et al, 2020). Surprisingly, we find that both p97 and ubiquitin interaction by UBXN1 are required for repressing the UPR (Fig. 2). While we find that UBXN1 does not participate in the degradation of several of our proteomic hits (SCD1, SQLE) or degradation of ERAD reporters(Ganji et al, 2018), it is possible that some ER proteins (such as AGAL) may be degraded in a p97-UBXN1 dependent manner and thus loss of p97 interaction would contribute to UPR activation. Nevertheless, we find that translational repression by UBXN1 is an important mechanism that represses UPR activation as inhibition of translation rescues UPR activation and consequent cell death in UBXN1 KO cells (Fig. 8). Indeed, previous studies have found that translation inhibition alleviates ER stress associated cytotoxicity (Kang et al, 2006; Oyadomari et al, 2006; Han et al, 2013).

How UBXN1 represses translation is currently under investigation, but one possibility is that UBXN1 may be recruited via its UBA domain to ER-bound ribosomes to specifically repress the synthesis of ER clients. In support of this idea, we show that UBXN1 localizes peripherally to ER membranes (Fig. EV2) and co-sediments with translating polysomes (Fig. EV5). Recent studies

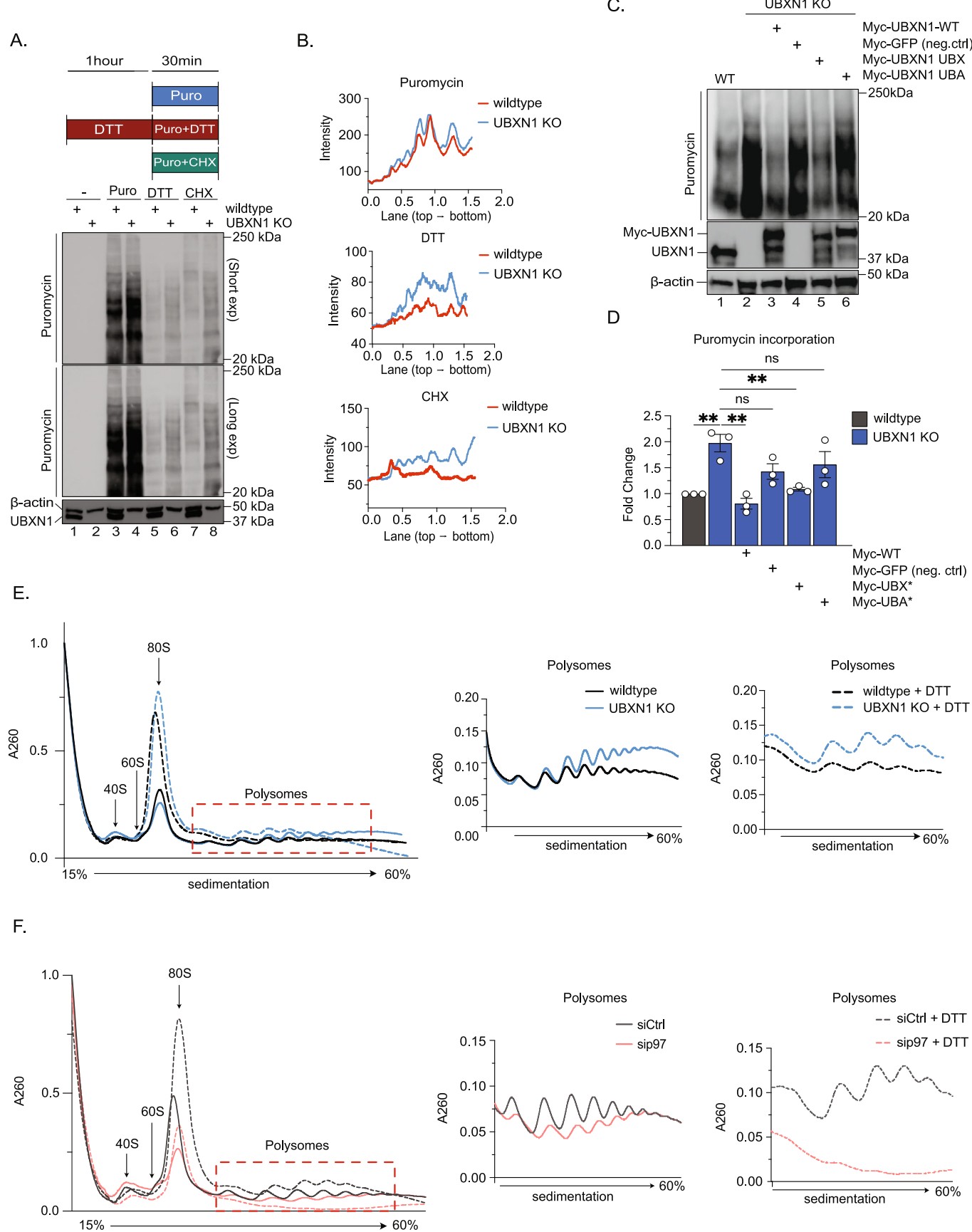

**Figure 7. UBXN1 represses protein translation.**

(A) Immunoblot of puromycin incorporation into HFT wild-type and UBXN1 KO cells. Cells were pulsed with 1 μM puromycin or 1 μM puromycin in combination with 10 μg/mL cycloheximide (CHX) for 30 min. When applicable, cells were pretreated with 1.5 mM DTT for 1 h before 1 μM puromycin pulse for 30 min. The level of puromycin incorporation reflects the rate of protein synthesis. (B) Puromycin, DTT, and CHX plots correspond to the intensity of each lane in the immunoblot in A. The intensity of the traces for each wild-type and UBXN1 KO sample was determined by the plot profile feature in Fiji. The x-axis represents the distance along the lane. (C) Myc-tagged GFP, wildtype, UBX domain mutant, or UBA domain mutant UBXN1 was expressed in cells for 48 hours before 1 μM puromycin pulse for 30 minutes. (D) Quantification of each lane intensity from C. (n = three biologically independent samples). (E) Polysome profile traces of HEK-293T wild-type and UBXN1 KO cells. Cells were treated with 2 mM DTT for 60 minutes where indicated. The 40S, 60S, and 80S ribosomal subunits are labeled as well as actively translating polysomes. Traces to the right correspond to the polysomes seen in the main trace. (F) Polysome profile traces of HEK-293T cells depleted of p97 with siRNA. Cells were treated with 2 mM DTT for 60 min where indicated. The 40S, 60S, and 80S ribosomal subunits are labeled as well as actively translating polysomes. Traces to the right correspond to the polysomes seen in the main trace. Data information: Data are means ± SEM (** where $P < 0.01$) One-way ANOVA with Tukey's multiple comparisons test (D). Source data are available online for this figure.

have demonstrated the surprising heterogeneity of translating ribosomes and have identified several ribosome-associated proteins that specifically regulate the synthesis of ER-bound mRNAs (Simsek et al, 2017; Noeske and Cate, 2012). For example, the RNA binding protein LIN28A localizes to the ER in human embryonic stem cells and specifically represses the synthesis of ER-integral membrane and secretory proteins while leaving nuclear and cytoplasmic proteins unaffected (Cho et al, 2012; Mayr and Heinemann, 2013). Similarly, pyruvate kinase muscle (PKM) is enriched on ER-bound ribosomes and activates the translation of a set of ER-associated mRNAs (Simsek et al, 2017). Previous studies have shown that the UBA domain of UBXN1 was necessary and sufficient to inhibit antiviral signaling in a p97-independent manner (Wang et al, 2013; Hu et al, 2017; Kawai et al, 2005). One possibility that we are exploring is whether UBXN1 is recruited to a ubiquitylated ribosomal protein to halt translation under certain circumstances. Several distinct ubiquitination events on ribosomal proteins (Sung et al, 2016; Juszkiewicz and Hegde, 2018) regulate ribosome abundance (An and Harper, 2020; Kraft et al, 2008) via degradation by the 26S proteasome or autophagy, ribosome disassembly through ribosome quality control pathways (Juszkiewicz and Hegde, 2017; Juszkiewicz et al, 2018; Sundaramoorthy et al, 2017; Higgins et al, 2015; Sugiyama et al, 2019), and translation initiation on problematic mRNAs (Hickey et al, 2020). It is possible UBXN1 could function to destabilize the ribosome to prevent translation of ER-localized mRNAs and is an area of current study.

The increased translation in UBXN1 null cells is somewhat surprising given the robust levels of eIF2α phosphorylation we observe that should attenuate global translation. Over-expression of CHOP and ATF4 after ER stress induces the expression of several aminoacyl-tRNA synthetases and ribosomal proteins leading to increased translation (Han et al, 2013). In addition, CHOP induces the expression of growth arrest and DNA damage-inducible protein (GADD34) (also known as protein phosphatase 1 regulatory subunit 15 A (PPP1R15A)), which forms a complex with protein phosphatase 1 (PP1) to dephosphorylate eIF2α and restore translation (Harding et al, 2000a; Fawcett et al, 1999; Novoa et al, 2001; Harding et al, 2009). Re-initiation of protein synthesis before the restoration of ER proteostasis can induce cell death in response to ER stress due to elevated reactive oxygen species (ROS) (Han et al, 2013; Guan et al, 2014). Indeed, MEFs harboring an S51A mutation in eIF2α, which prevents phosphorylation of eIF2α by PERK were significantly sensitized to ER stress, whereas ATF4 null cells, which more slowly restored protein synthesis, were resistant to ER-stress induced cell death (Han et al, 2013). We find robust

expression of ATF4 (Fig. 1) and CHOP (Fig. 3) in UBXN1 null cells and these cells are more sensitive to ER stress than their wild-type counterparts (Fig. 3) suggesting that loss of UBXN1 may allow for premature translation restart through ATF4-CHOP. This possibility is currently under investigation.

We propose that translational repression by UBXN1 may represent a novel "pre-emptive" ER-quality control pathway. Emerging studies indicate the presence of numerous pre-emptive ER-quality control systems (for example the BAG6-RNF126 mediated degradation of mislocalized ER proteins) (Rodrigo-Brenni et al, 2014; Ganji et al, 2018; Wang et al, 2011a; Kang et al, 2006) that preclude protein misfolding in the ER by preventing the translocation of aberrant proteins. Ufmylation of the ribosomal protein uL24 (Rpl26) enables the degradation of stalled polypeptides in the translocon (Walczak et al, 2019; Wang et al, 2020). It is likely that multiple measures exist to maintain an optimal ER proteome prior to UPR and ERAD activation. Our study suggests that UBXN1-dependent translation repression may be one such mechanism to maintain ER proteostasis.

## Methods

### Cell culture, transfections, immunoblot, and immunoprecipitation

HeLa-Flp-IN-TREX (HFTs (ThermoFisher Cat# R71407) with introduced Flp-In site (Flp-In™ T-REx™ Core Kit, Cat# K650001; Thermofisher Scientific is a gift from Brian Raught, University of Toronto) and HEK-293T (ATCC# CRL-3216™) cells were cultured in Dulbecco's modified Eagle's medium (DMEM) supplemented with 10% fetal bovine serum (FBS) and 100 U/mL penicillin and streptomycin. Cells were grown in a humidified, 5% $CO_2$ atmosphere at 37 °C and tested routinely for mycoplasma with MycoAlert (Lonza #LT07-118). For siRNA transfections, cells were either forward or reverse transfected with 20 nM siControl, siUBXN1, siBAG6, or sip97 using Lipofectamine RNAiMax (Invitrogen) in six-well plates. After 8 or 24 h depending on the study, cells were split into 12-well plates for treatment. Forty-eight hours post transfection, cells were treated with the indicated drugs for the indicated timepoints before harvest for immunoblot or fixation for immunofluorescence. For DNA transfections, 1 μg HA-prion protein (PrP), 0.5 μg Myc-tagged wild-type UBXN1, 0.5 μg of either Myc-tagged UBX mutant (Phe[265] Pro[266] Arg[267] truncated to Ala-Gly), or Myc-tagged UBA mutant (Met[13] and Phe[15] to Ala), and 50 ng Myc-tagged GFP constructs were forward transfected into

HFT cells seeded in a six-well plate using Lipofectamine 3000 (Invitrogen). The cells were harvested 48- or 72-h post transfection. To establish an HFT cell line stably expressing doxycycline

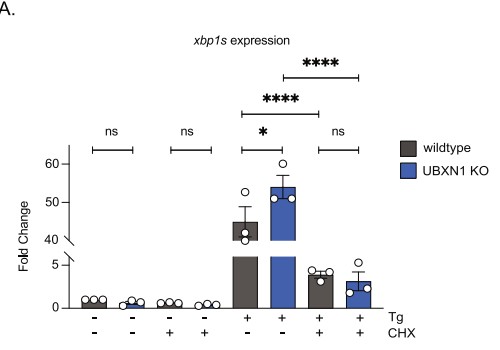

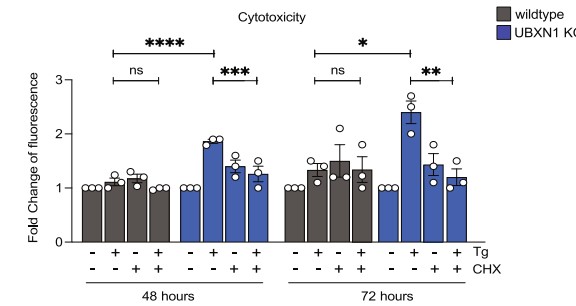

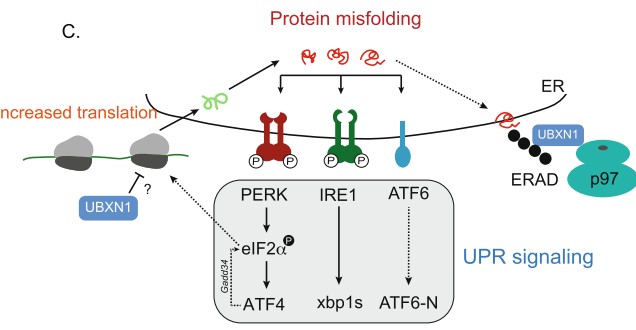

**Figure 8.  Inhibiting translation rescues the UPR and cell death phenotypes in UBXN1 KO cells.**

(A) Relative *xbp1s* expression quantified by quantitative real-time PCR. Cells were treated with 10 nM thapsigargin in combination with 10 µg/ml cycloheximide where indicated. (*n* = three biologically independent samples. (B) Fold change of the fluorescence measured by fluorescence-based cytotoxicity assay. Cells were treated with 1.5 µM thapsigargin in combination with 10 µg/ml cycloheximide where indicated. Values were normalized to untreated for each genotype. Note we observe no differences in viability between wild-type and UBXN1 KO cells in untreated conditions allowing for this normalization (*n* = three biologically independent samples. (C) Model Figure. UBXN1 plays an important role in ER-quality control as a repressor of translation and the UPR. Loss of UBXN1 increases protein synthesis. Increased abundance of ER proteins leads to protein misfolding and activation of the UPR which reduces cell viability. Some p97 ERAD clients may require UBXN1 for degradation. Data information: Data are means ± SEM (*, **, ***, **** where $P < 0.05$, 0.01, 0.001, and 0.0001, respectively.) One-way ANOVA with Tukey's multiple comparisons test (A, B). Source data are available online for this figure.

inducible FLAG-AGAL, 1 µg FLAG-AGAL construct and 4.5 µg pOG44 was transfected into HFTs seeded onto a six-well plate using Lipofectamine 2000 (Invitrogen). After 48 h the transfected cells were selected with hygromycin (200 µg/mL) for about 10 days.

Cell pellets were lysed in either radioimmunoprecipitation assay (RIPA) buffer (25 mM Tris-Cl [pH 7.6], 150 mM NaCl, 1% NP-40, 1% sodium deoxycholate, 0.1% SDS, HALT protease inhibitors (Pierce PI-78425), sodium vanadate, sodium fluoride) or in 1% SDS lysis buffer (1% SDS, 50 mM Tris-Cl, 150 mM NaCl, 0.5% NP-40, HALT protease inhibitors, sodium vanadate, sodium fluoride). FLAG-AGAL and HA-PrP cell pellets were resuspended in RIPA buffer and incubated on ice for 20 minutes with a 10 s vortex pulse every 5 minutes. Lysates were then centrifuged at 14,000 rpm for 15 min at 4 °C. For UPR induction studies in Fig. 1, cell pellets were resuspended in SDS lysis buffer, vortexed briefly, boiled for 10 min at 95 °C, and briefly vortexed. Lysates were incubated at 65 °C for an additional 5 min before centrifugation at 14,000 rpm at room temperature. The supernatant was collected, and protein concentration was estimated utilizing the DCA protein assay kit (Biorad). Protein expression was analyzed by SDS-PAGE and immunoblot.

For immunoprecipitations, cell pellets were lysed in mammalian cell lysis buffer (50 mM Tris-HCl [pH 7.6], 150 mM NaCl, 1% Nonidet P-40, HALT protease inhibitors (Pierce), and 1 mM DTT). Protein G agarose beads (Thermo Scientific 20398) were pre-washed with MCLB before the addition of the cell lysate and antibody (UBXN1, Sec61β, or rabbit IgG control). Samples were incubated with rotation overnight at 4 °C. Beads were collected by centrifugation at 3000 rpm for 1 min and was washed four times with MCLB buffer. Beads were resuspended in 50 µl SDS sample buffer, briefly boiled, and processed for SDS-PAGE and immunoblotting.

## Antibodies, siRNA, and reagents

The rabbit UBXN1 (16135-1-AP 1:7000 dilution), p97 (10736-1-AP), BAG6 (26417-1-AP), ATF6 (24169-1-AP), Sec61β (51020-2-AP 1:3000 dilution), TIMM17A (11189-1-AP), Aconitase 2 (11134-1-AP), calnexin (10427-2-AP), CYC1 (10242-1-AP), and G3BP1 (13057-2-AP) antibodies were obtained from Proteintech Inc; rabbit BiP (3177 dilution), peIF2α (3398 1:500 dilution), and MCM3 (4012S) antibodies were obtained from Cell Signaling Technology; mouse PCNA (sc-56 1:3000 dilution), ATF4 (sc-390063 1:500 dilution), TOMM20 (sc-17764), TOMM70 (sc-390545), TIMM23 (sc-514463), ubiquitin (P4D1; sc8017), β-actin (sc-69879 1:3000 dilution), GAPDH (sc-47724 1:3000 dilution), c-myc (sc-40 1:3000 dilution), Sec61α (sc-393182 1:500 dilution), and TRAPα (sc-373916) were from Santa Cruz Biotechnology; mouse ubiquitin FK2 (04-263 1:100 dilution) for immunofluorescence and Anti-Puromycin, clone 12D10 (MABE343 1:3000 dilution) was from EMD Millipore; mouse FLAG-M2 antibody (F3165 1:1000 dilution) was from Sigma Aldrich. The rabbit anti-Prion protein PrP antibody (ab52604) was purchased from Abcam. The rabbit ALPP/ALPP2 was purchased from St. John's Laboratory (STJ96740). The mouse HSP90AB1 was purchased from Origene (TA500494). All antibodies were used at a dilution of 1:1000 unless otherwise specified. Secondary antibodies, HRP conjugated anti-rabbit (W4011) and anti-mouse (W4021) were from Promega and used at a dilution of 1:10,000, CellTox Green Cytotoxicity assay (G8741) was also purchase from Promega. Thapsigargin

(5860051MG), Tunicamycin (ICN15002801), Proteinase K (FEREO0491), and Hoechst (51-17 1:10000 dilution) for immunofluorescence were purchased from Fisher Scientific. Dithiothreitol (DTT25) and puromycin (P-600-100) were purchased from Gold-Bio. Cyclohexmide (97064-724) and Triton X-100 (97063-864) was purchased from VWR. CB-5083 were purchased from Cayman Chemical Company (19311). Bortezomib (S1013) was purchased from Selleckchem. EndoH (P0702S) was purchased from New England Biolabs. Crystal violet was purchased from Sigma-Aldrich (C6158-50G). Doxycycline was purchased from EMD Millipore (324385-1GM). Wild-type PrP (HA) was generously gifted by Ramanujan Hegde (MRC, UK). FLAG-tagged AGAL were generously gifted by Malaiyalam Mariappan (Sun et al, 2023). siRNAs against UBXN1-2 (D-008652-02), UBXN1-3 (D-008652-03), and UBXN1-4 (D-008652-04) were purchased from Dharmacon. siRNA against BAG6 (s15467) was purchased from Ambion (Themo Fisher Scientific). siControl (SIC001) was from Millipore Sigma. siRNA against p97 was purchase from Invitrogen (HSS111263).

## Generation of CRISPR gene knockout cell lines

UBXN1 knockout cells were generated in both HFT and HEK-293T cells using the CRISPR-Cas9 gene editing method. The guide sequence 5′-GCCGTCCCAGGATATGTCCAA-3′ was cloned into the pX459 vector carrying hSpCas9 as previously published (Ganji et al, 2018). 500 ng of the construct was transiently transfected into HFT and HEK-293T cells seeded in a 6-well plate with Lipofectamine 3000. 36 h post-transfection, the cells were cultured with 1 µg/mL puromycin for an additional 24 h. The surviving cells were counted and serially diluted to a concentration of 1 cell per well in a 96-well plate. Protein expression of UBXN1 was examined by immunoblot.

## Crystal violet cell viability assay

Wild-type and UBXN1 KO cells were plated at a density of 8000 cells/well in a 12-well dish. The next day, cells were treated with 200 nM thapsigargin for the indicated timepoints. After treatment, wells were washed twice with PBS before re-addition of fresh media. Every day for three days, cells were washed once with PBS and media was replaced. On the third day, cells were washed twice with PBS before incubation with 0.5% crystal violet staining solution for 20 min at room temperature on an orbital shaker. After staining, wells were washed thrice with PBS to remove unbound stain. 1% SDS-solution was added to each well to solubilize bound stain and the plate was incubated for one hour at room temperature on an orbital shaker until color was uniform in solution. The absorbance was read at 570 nm on a multiwell plate reader (SpectraMax iD3). The average absorbance of wells without cells was used to background subtract.

## Real-time PCR

For all real-time PCR experiments, total RNA was isolated using the *Quick*-RNA Miniprep Kit (Zymo Research cat. no. R1055). The purified RNA was quantified by NanoDrop and 1 µg of RNA for each sample was used to generate cDNA using the iScript cDNA synthesis kit (Biorad cat. no. 1708890). Real-time PCR was performed with PowerUp SYBR Green Master Mix (Applied Biosystems cat. no. A25741) on an Applied Biosystems

StepOnePlus real-time PCR system. Data analyses utilized the $2^{-\Delta\Delta Ct}$ method and GAPDH was used as a housekeeping gene to normalize transcript expression across samples. The XBP1s (Van Schadewijk et al, 2012), CD59 (Ouyang et al, 2016), and BLOC1S1 (Bright et al, 2015) primers were previously published. In studies measuring xbp1s, a low dose of thapsigargin was used to maintain dynamic range in the assay and avoid the ceiling effect observed with UPR target gene expression under severe ER stress (Shinjo et al, 2013; Travers et al, 2000).

> AGAL forward: 5′ GCTGGAAGGATGCAGGTTAT 3′
> AGAL reverse: 5′ ATGCTTCTGCCAGTCCTATTC 3′
> ALPP2 forward: 5′ CAACGAGGTCATCTCCGTGATG 3′
> ALPP2 reverse: 5′ TACCAGTTGCGGTTCACCGTGT 3′
> BLOC1S1 forward: 5′ CCCAATTTGCCAAGCAGACA 3′
> BLOC1S1 reverse: 5′ CATCCCCAATTTCCTTGAGTGC 3′
> CD59 forward: 5′ ATGGGAATCCAAGGAGGGTC 3′
> CD59 reverse: 5′ ATTGACGGCTGTTTTGCAGT 3′
> GAPDH forward: 5′ ATCAAGAAGGTGGTGAAGCAG 3′
> GAPDH reverse: 5′ CAGCAGTGAGGGTCTCTC 3′
> HA-PrP forward: 5′ CAGCCTCATGGTGGTGGCTG 3′
> HA-PrP Reverse: 5′ GTCGGTCTCGGTGAAGTTCTCC 3′
> SCD1 forward: 5′ CGACGTGGCTTTTTCTTCTC 3′
> SCD1 reverse: 5′ CCTTCTCTTTGACAGCTGGG 3′
> SQLE forward: 5′ ACCCGAGTCCAGTTCTCATCTA 3′
> SQLE reverse: 5′ CCTTGGCATTTCTCCTCTAATG 3′
> TRAPα forward: 5′ GTGAAGTTCCTGGTAGGCTTTA 3′
> TRAPα reverse: 5′ CCATCTAACCCATCCTCTCTTTC 3′
> XBP1s forward: 5′ TGCTGAGTCCGCAGCAGGTG 3′
> XBP1s reverse: 5′ GCTGGCAGGCTCTGGGGAAG 3′
> XBP1 total forward: 5′ AAACAGAGTAGCTCAGACTGC 3′
> XBP1 total reverse: 5′ TCCTTCTGGGTAGACCTCTGGGAG 3′

## RT² Profiler PCR array

Wild-type or UBXN1 KO HeLa Flp-in T-REX cells were grown in 10 cm tissue culture dishes and either left untreated or treated with 1.5 mM DTT for 4 h. Post-harvest, RNA was extracted from each sample with the *Quick*-RNA MiniPrep Kit (Zymo Research cat. no. R1055) and cDNA was generated with the RT² First Strand Kit (Qiagen cat no. 330401). Each Human Unfolded Protein Response RT² Profiler PCR Array (cat no. 330231 PAHS-089ZA) plate consists of 84 UPR specific target genes as well as a panel of housekeeping genes and PCR efficiency controls. Each sample was distributed to an individual plate for two biological replicates each. The GeneGlobe Data Analysis Center was used to determine the fold change between samples. The data were normalized by automatic selection from the housekeeping gene panel utilizing the geometric mean. The untreated wild-type HFT samples were defined as the "control" sample for the data set and used to calculate fold changes. Heatmap visualizations for PCR array data were performed using the RStudio software (v2022.07.1 Build 554), including ggplot2 (v 3.3.5), circlize (v0.4.14), ComplexHeatmap version 2.10.0, and InteractiveComplexHeatmap package to export heatmap generated by ComplexHeatmap into an interactive Shiny app (Gu et al, 2016).

## Immunofluorescence and microscopy

To examine aggresome formation, cells were grown on coverslips (12 mm) in a 24-well plate and treated with 1 µM Tunicamycin

(Tu), Thapsigargin (Tg), or bortezomib (Btz) for 8 hours or 500 nM Tu, 500 nM Tg, or 1 μM Btz for 18 hours. Tunicamycin and thapsigargin were used in this study as they are less toxic than DTT and thus more amenable to imaging studies. To examine mitochondrial morphology, cells were grown on coverslips (12 mm) in a 24-well plate and treated with either 1.5 μM thapsigargin for 6 h or 10 μM antimycin A with 5 μM oligomycin A for 2 h. Cells were washed briefly in PBS and fixed for 10 min with ice-cold methanol at −20 °C. Cells were washed briefly in PBS then blocked for 1 h in 2% bovine serum albumin (BSA) with 0.3% Triton X-100 in PBS. The coverslips were incubated with the FK2 (ubiquitin) antibody (aggresomes) or TOMM20 antibody (mito-chondria morphology) in a humidified chamber at 4 °C overnight. Coverslips were washed with PBS and incubated with an Alexa Fluor-mouse conjugated secondary antibody (Molecular Probes) for 1 h at room temperature in the dark. Coverslips were briefly washed with PBS and nuclei were stained with Hoechst 33342 dye and mounted onto slides. All images were collected using a Nikon A1R scan head with a spectral detector and resonant scanners on a Ti-E motorized inverted microscope equipped with a 60× Plan Apo 1.4 NA objective lens as previously published (Mukkavalli et al, 2021). The indicated fluorophores were excited with either a 405-nm, 488-nm or 594-nm laser line. Images were analyzed by using FIJI (https://imagej.net/Fiji). Mitochondrial branch length was calculated utilizing the published Mitochondrial Network Analysis (MiNa) toolset of ImageJ plug-ins (Valente et al, 2017). Images were acquired in a blinded manner using Hoechst to identify cells. All image analysis was carried out using pre-published scripts to avoid bias.

## IRE1α-GFP foci imaging and quantification

Stable HFT wild-type cells expressing doxycycline inducible IRE1α-3xFLAG-6xHis-GFP (construct a generous gift from Peter Walter, UCSF) were seeded onto a six-well plate and reverse transfected with an siControl or siUBXN1 (Li et al, 2010). Next day, cells were trypsinized and seeded onto coverslips in a 12-well plate. 48 h after siRNA transection, IRE1α-GFP was induced with 4 μg/mL doxycycline for an additional 24 h. Next day, cells were either left untreated or treated with 2.5 μM tunicamycin for 4 h. Coverslips were briefly washed with phosphate-buffered saline (PBS) and fixed with ice-cold methanol for 10 min at −20 °C. Coverslips were washed with PBS and nuclei were stained with Hoechst 33342 dye and mounted on slides. Images were collected using either a 405-nm or 488-nm laser. One hundred cells were counted by creating a cell mask and utilizing the "Find Maxima" function in FIJI (https://imagej.net/Fiji). Images were acquired in a blinded manner using Hoechst to identify cells.

## Subcellular fractionation of ER-derived microsomes

HEK-293T cells were seeded into four 150 mm sterile tissue culture dishes. Cells were scraped off the plate into PBS, centrifuged at 600 × g for 5 min at 4 °C, and resuspended in ice-cold homogenization buffer (225 mM Mannitol, 75 mM Sucrose, 30 mM Tris [pH 7.4]). Cells were homogenized with a Dounce homogenizer on ice. Every 150 passes of the homogenizer, cells were centrifuged for 5 min at 600 × g, to remove non-lysed cells and nuclei. The integrity of the homogenized cells was examined under a light microscope

until 80–90% cell lysis was attained. The supernatant was collected as the post-nuclear supernatant (PNS) fraction. The clarified lysate was centrifuged at 7000 × g for 10 min at 4 °C, isolating the supernatant including ER-microsomes. The supernatant was collected, centrifuged at 20,000 × g for 30 min at 4 °C to clear lysosomal and plasma membrane fractions. The resulting super-natant was further centrifuged at high speed, 100,000 × g for 1 h in a swinging-bucket rotor SW-41 and Beckman Coulter ultracen-trifuge. This results in the isolation of pure ER-microsomes (pellet) and cytosolic fraction (supernatant). PNS, cytosolic, and ER-microsomes were solubilized using 0.5% (v/v) SDS and protein concentrations were determined by DC protein assay. An equivalent amount of ER-microsomes were used in the protease protection assay. Of four aliquots, one was left untreated, one was incubated with 1% (v/v) Triton X-100, and the other two were incubated with 10 μg/mL proteinase K with or without 1% (v/v) Triton X-100. All samples are incubated at 37 °C for 10 min before reaction termination with 2 mM phenylmethylsulfonyl fluoride (PMSF). 5X-Laemmli sample buffer was added to each sample before SDS-PAGE and immunoblot.

## CellTox green cytotoxicity assay

HFT wild-type and UBXN1 KO cells were seeded in a 96-well black, clear bottom plate at a density of 10,000 cells/well. The CellTox Green dye was added to the cells at the time of seeding. Next day, cells were treated with either 500 nM or 250 nM thapsigargin. Fluorescence readings were taken at 24-, 48-, and 72-h post thapsigargin treatment at a wavelength of 485 nm excitation and 535 nm emission. Graphed values are the fold changes of the fluorescence values normalized to untreated control wells of each genotype.

## Puromycilation translation assay

Protein synthesis was measured in HFT wild-type and UBXN1 KO cells utilizing the non-radioactive SUnSET assay (Ravi et al, 2020). HFT wild-type or UBXN1 KO cells were seeded onto a six-well plate at 400 k cells/well to achieve 70–80% confluence at the time of puromycin addition. Cells were incubated with 1 μM puromycin with or without 100 μg/ml cycloheximide for 30 min in complete medium. Samples treated with DTT or thapsigargin were pre-incubated with 1.5 mM DTT for 1 hour or 1 μM thapsigargin for 30 min before co-treatment of 1 μM puromycin with 1.5 mM DTT for 30 min in complete medium. At the end of the incubation period, cells were washed with ice-cold PBS and the plate was immediately placed on ice. Ice-cold RIPA buffer was added to the wells and cells were gently scraped into the buffer using a cell scraper. Lysates were placed on ice and vortexed for 10 seconds every 5 min for 30 min before centrifugation at 12,000 rpm for 10 min at 4 °C. Protein concentration was estimated using the DC assay (Bio-Rad). An equivalent amount of protein for each sample was resolved by SDS-PAGE and probed with the anti-puromycin antibody, clone 12D10.

## Polysome profiling

HEK-293T wild-type and UBXN1 KO cells were plated in a 150 mm dish and allowed to grow for two days before polysome

isolation. For p97 depletion experiments, HEK-293T cells were plated in a 150 mm dish and forward transfected next day with siRNA against p97 or an siControl. Twenty-four hours later the plate was split into two 150 mm dishes and the cells were grown for two additional days. The day before profiling, sucrose gradients were prepared with fresh 15% and 60% sucrose solutions using 1X low-salt buffer (5 mM Tris-HCl [pH 7.5], 2.5 mM MgCl₂, 1.5 mM KCl, 1 mM PMSF). To generate a continuous gradient, 5 mL of 60% sucrose solution was placed at the bottom of an ultra-clear Beckman Coulter centrifuge tube. 5 mL of 15% sucrose solution was layered on top and the gradient was inverted horizontally and left on its side at room temperature for 3 h. After 3 h, the tube was brought back to its vertical position and stored at 4 °C overnight. When cells reached about 70–80% confluency, cells were either pre-treated with 2 mM DTT for 60 min or left untreated. After treatment, cells were incubated with 100 µg/mL cycloheximide at 37 °C for 15–20 min. Cells were then washed twice with ice cold PBS supplemented with 100 µg/mL cycloheximide and 1 mM PMSF, scraped off the plate, and pelleted at $200 \times g$ for 5 min at 4 °C. Cells were washed once with ice cold PBS, pelleted, and resuspended in 480 µL 1X low salt buffer supplemented with 100 µg/mL cycloheximide, 2 µM DTT, 1 mM PMSF, and 1 mM HALT protease inhibitor cocktail. 60 units of RNaseOUT Recombinant RNase Inhibitor (Invitrogen, 10777019) was added to each sample and vortexed for 5 s. Triton X-100 and sodium deoxycholate was added to each sample to a final concentration of 0.5% and samples were vortexed for an additional 5 s. Samples were kept on ice for 5 minutes before centrifugation at $16,000 \times g$ for 7 min at 4 °C. Samples were quantified at 260 nm in UV compatible plates on a multiwell plate reader. 20 units of each sample was loaded onto the continuous sucrose gradient and spun at 24,200 rpm for 3 h 30 min using an SW-41 swing bucket rotor in a Beckman Coulter ultracentrifuge. After the spin, the samples were run on a continuous UA-6 detector, density gradient fractionator system with peak detection software. Values determined by the fraction collector were graphed on GraphPad Prism.

## TMT-based proteomics

### Sample preparation, digestion, and tandem-mass tag (TMT) labeling
The TMT-based proteomics was performed exactly as previously described (Ganji et al, 2023). Briefly, 100 µg protein of each sample was obtained by cell lysis in lysis buffer (8 M Urea, 200 mM N-(2-Hydroxyethyl)piperazine-N′-(3-propanesulfonic acid) (EPPS) pH 8.5) followed by reduction using 5 mM tris(2-carboxyethyl) phosphine (TCEP), alkylation with 14 mM iodoacetamide and quenched using 5 mM dithiothreitol. The reduced and alkylated protein was precipitated using methanol and chloroform. The protein mixture was digested with LysC (Wako) overnight followed by Trypsin (Pierce) digestion for 6 h at 37 °C. The trypsin was inactivated with 30% (v/v) acetonitrile. The digested peptides were labeled with 0.2 mg per reaction of 4-plex TMT reagents (Thermo-Fisher scientific) (126, 127 N, 129 N, and 129 C) at room temperature for 1 hour. The reaction was quenched using 0.5% (v/v) Hydroxylamine for 15 min. A 2.5 µL aliquot from the labeling reaction was tested for labeling efficiency. TMT-labeled peptides from each sample were pooled together at a 1:1 ratio. The pooled peptide mix was dried under vacuum and resuspended in 5% formic acid for 15 min. The resuspended peptide sample was

further purified using C18 solid-phase extraction (SPE) (Sep-Pak, Waters).

### Off-line basic pH reverse-phase (BPRP) fractionation
We fractionated the pooled, labeled peptide sample using BPRP HPLC(Wang et al, 2011b). We used an Agilent 1200 pump equipped with a degasser and a detector (set at 220 and 280 nm wavelength). Peptides were subjected to a 60-min linear gradient from 5 to 35% acetonitrile in 10 mM ammonium bicarbonate pH 8 at a flow rate of 0.25 mL/min over an Agilent 300Extend C18 column (3.5 µm particles, 2.1 mm ID and 250 mm in length). The peptide mixture was fractionated into a total of 96 fractions, which were consolidated into 24 super-fractions (Paulo et al, 2016a). Samples were subsequently acidified with 1% formic acid and vacuum centrifuged to near dryness. Each consolidated fraction was desalted via StageTip, dried again via vacuum centrifugation, and reconstituted in 5% acetonitrile, 5% formic acid for LC-MS/MS processing.

### Liquid chromatography and tandem mass spectrometry
Mass spectrometric data were collected on an Orbitrap Fusion mass spectrometer coupled to a Proxeon NanoLC-1000 UHPLC. The 100 µm capillary column was packed with 35 cm of Accucore 150 resin (2.6 µm, 150 Å; ThermoFisher Scientific). The scan sequence began with an MS1 spectrum (Orbitrap analysis, resolution 120,000, 350–1400 Th, automatic gain control (AGC) target $5 \times 10^5$, maximum injection time 100 ms). Data were acquired for 150 min per fraction. SPS-MS3 analysis was used to reduce ion interference (Gygi et al, 2019, Paulo et al, 2016b). MS2 analysis consisted of collision-induced dissociation (CID), quadrupole ion trap analysis, automatic gain control (AGC) $1 \times 10^4$, NCE (normalized collision energy) 35, $q$-value 0.25, maximum injection time 60 ms), isolation window at 0.7 Th. Following acquisition of each MS2 spectrum, we collected an MS3 spectrum in which multiple MS2 fragment ions were captured in the MS3 precursor population using isolation waveforms with multiple frequency notches. MS3 precursors were fragmented by HCD and analyzed using the Orbitrap (NCE 65, AGC $2.0 \times 10^5$, isolation window 1.3 Th, maximum injection time 150 ms, resolution was 50,000).

### Data analysis
Spectra were converted to mzXML via MSconvert (Chambers et al, 2012). Database searching included all entries from the Human UniProt Database (downloaded: April 2016). The database was concatenated with one composed of all protein sequences for that database in the reversed order. Searches were performed using a 50-ppm precursor ion tolerance for total protein level profiling. The product ion tolerance was set to 0.9 Da. These wide mass tolerance windows were chosen to maximize sensitivity in conjunction with Comet searches and linear discriminant analysis (Beausoleil et al, 2006, Huttlin et al, 2010). TMT tags on lysine residues and peptide N-termini (+229.163 Da for TMT) and carbamidomethylation of cysteine residues (+57.021 Da) were set as static modifications, while oxidation of methionine residues (+15.995 Da) was set as a variable modification. Peptide-spectrum matches (PSMs) were adjusted to a 1% false discovery rate (FDR) (Elias and Gygi, 2010, Elias and Gygi, 2007). PSM filtering was performed using a linear discriminant analysis, as described previously (Huttlin et al, 2010)

and then assembled further to a final protein-level FDR of 1% (Elias and Gygi, 2007). Proteins were quantified by summing reporter ion counts across all matching PSMs, also as described previously (Mcalister et al, 2012). Reporter ion intensities were adjusted to correct for the isotopic impurities of the different TMT reagents according to manufacturer specifications. The signal-to-noise (S/N) measurements of peptides assigned to each protein were summed and these values were normalized so that the sum of the signal for all proteins in each channel was equivalent to account for equal protein loading. Finally, each protein abundance measurement was scaled, such that the summed signal-to-noise for that protein across all channels equaled 100, thereby generating a relative abundance (RA) measurement.

Downstream data analyses for TMT datasets were carried out using the R statistical package (v4.0.3) and Bioconductor (v3.12; BiocManager 1.30.10). TMT channel intensities were quantile normalized and then the data were log-transformed. The log transformed data were analyzed with limma-based R package where p-values were FDR adjusted using an empirical Bayesian statistical. Differentially expressed proteins were determined using a $\log_2$ (fold change (WT vs UBXN1 KO)) threshold of $>+/-0.65$.

### Gene ontology (GO) functional enrichment analyses of proteomics data

The differentially expressed proteins were further annotated and GO functional enrichment analysis was performed using Metascape online tool (http://metascape.org) (Zhou et al, 2019). The GO cluster network and protein-protein interaction network generated by metascape and the STRING database (https://string-db.org/), respectively, were imported into Cytoscape software (v3.8.2) to add required attributes (fold changes, p-values, gene number, and conditions) and prepared for the visualization. Other proteomic data visualizations were performed using the RStudio software (v1.4.1103), including hrbrthemes (v0.8.0), viridis (v0.6.1), dplyr (v.1.0.7), and ggplot2 (v 3.3.5).

### Quantification and statistical analysis

Immunoblots were quantified using densitometry on FIJI and protein levels were normalized to the loading control for that experiment (GAPDH or β-actin). Fold changes, SEM, and statistical significance was calculated using GraphPad Prism (version 9.4). Statistical tests performed are indicated in figure legends and include unpaired two-tailed Student's *t*-test or one-way ANOVA with post-hoc analysis indicated in figure legends. No statistical method was used to predetermine sample size. In general, all relevant studies were repeated at least three independent times. Detailed n number are provided in the figures and/or figure legends. Representative image from these repeats were presented, where applicable in the figures. For proteomics studies, sample size was chosen based on previously published studies by the Raman, and Gygi groups (PMID: 26389662; PMID: 28375945).

## Data availability

The mass spectrometry proteomics data have been deposited to the ProteomeXchange Consortium via the PRIDE (Perez-Riverol et al, 2022) partner repository (https://www.ebi.ac.uk/pride/) with the dataset identifier PXD043686.

## Peer review information

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

## Acknowledgements

We thank Peter Juo, Michelle Johnson, and Rakesh Ganji for critical reading of the manuscript. We are grateful to Peter Walter (University of California San Francisco and Altos Labs) for the GFP-IRE1 construct, and Mals Mariappan (Yale University) for the FLAG-AGAL construct. This work is supported by the NIH grants GM127557, NS123631 and a Research Scholar Grant RSG-19-022-01 from the American Cancer Society to MR. This work was funded in part by NIH/NIGMS grant GM67945 (SPG) and R01 GM132129 (JAP).

## Author contributions

**Brittany Ann Ahlstedt**: Conceptualization; Data curation; Formal analysis; Validation; Investigation; Methodology; Writing—original draft; Project administration; Writing—review and editing. **Rakesh Ganji**: Investigation; Methodology. **Sirisha Mukkavalli**: Investigation; Methodology. **Joao A Paulo**: Funding acquisition; Investigation; Methodology. **Steven P Gygi**: Funding acquisition. **Malavika Raman**: Conceptualization; Resources; Formal analysis; Supervision; Funding acquisition; Visualization; Methodology; Writing—original draft; Project administration; Writing—review and editing.

## Disclosure and competing interests statement

The authors declare no competing interests.

# Expanded View Figures

**Figure EV1. Acute BAG6 and UBXN1 depletion and pharmacological inhibition of p97 induces activation of the unfolded protein response.** ▶

(A) Immunoblot of BiP and ATF4 expression levels in HFT wildtype cells treated with 1 µM thapsigargin (Tg), 5 µM of the ATP-competitive p97 inhibitor CB-5083, or both for the indicated timepoints (0–6 hours). (B, C) Band intensity quantifications of BiP (B) and ATF4 (C) corresponding to Figure EV1A reporting the fold change compared to the untreated condition ($n$ = three biologically independent samples). (D) Immunoblot of HFT wildtype cells depleted of BAG6 with siRNA for 48 hours before treatment with 1.5 mM DTT for the indicated timepoints (0–8 hours). (E) Band intensity quantifications of BiP and ATF4 corresponding to Figure EV1D at the 8-hour timepoint. ($n$ = four biologically independent experiments). (F) Transcript levels of *xbp1s* and total *xbp1* in HEK-293T wildtype and UBXN1 KO cells quantified by quantitative real-time PCR. Cells were treated with 10 nM thapsigargin (Tg) for 4 hours as indicated. ($n$ = three biologically independent samples). (G) Immunoblot of BiP and ATF4 expression in cells depleted of UBXN1 with siRNA. Cells were treated with 1.5 mM DTT for the indicated time points. (H) Band intensity quantifications of BiP and ATF4 from Figure EV1G at the 8-hour time point. ($n$ = four biologically independent experiments). (I) Immunoblot of ATF6 activation in cells depleted of UBXN1 with siRNA. Cells were treated with 1.5 mM DTT and 1 µM Bortezomib (BTZ). (J) ATF6 activation was measured by band intensity quantification and calculation of the percentage of cleaved ATF6 to total ATF6. The ratio of the percentage of ATF6 activation in UBXN1 KO cells to wildtype is reported. ($n$ = three biologically independent samples). Data information: Data are means ± SEM (*, **, ***, **** where $P < 0.05$, 0.01, 0.001, and 0.0001, respectively.) One-way ANOVA with Dunnetts multiple comparisons test (B, C). Unpaired two-tailed $t$ test (E, H, J). One-way ANOVA with Tukey's multiple comparisons test (F).

A.

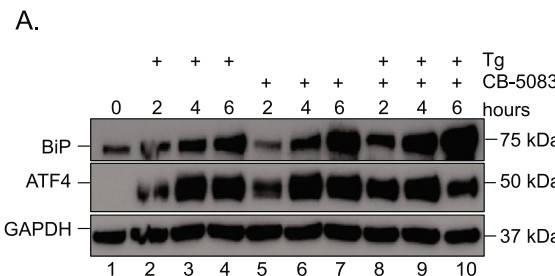

B.

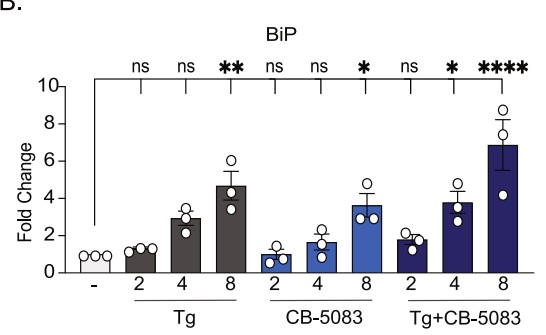

C.

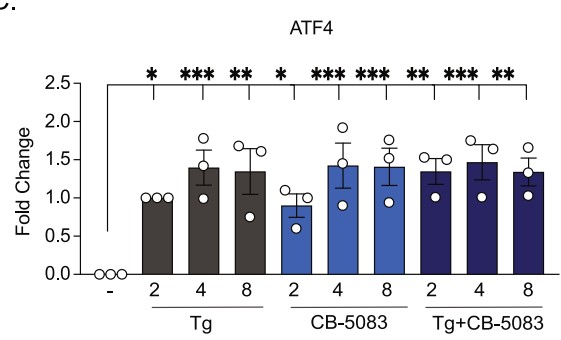

D.

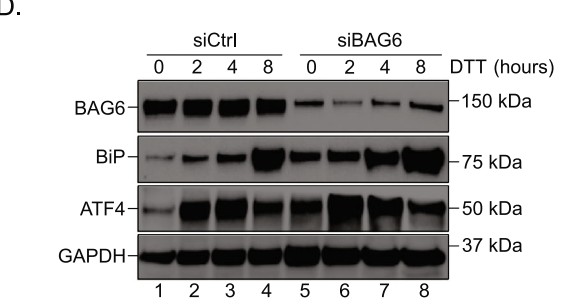

E.

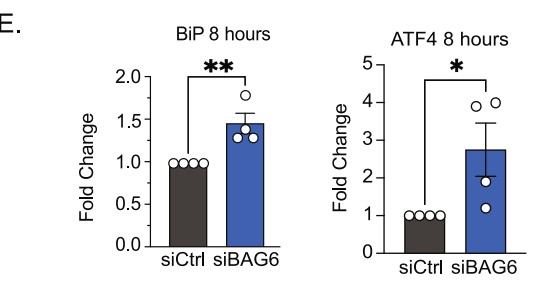

F.

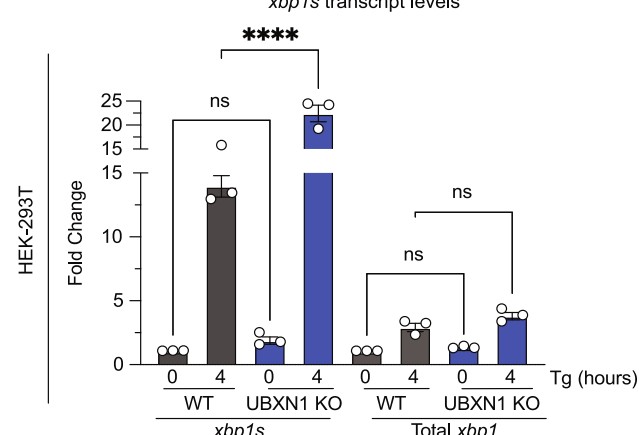

G.

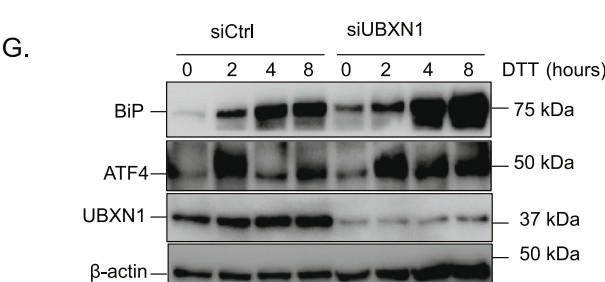

H.

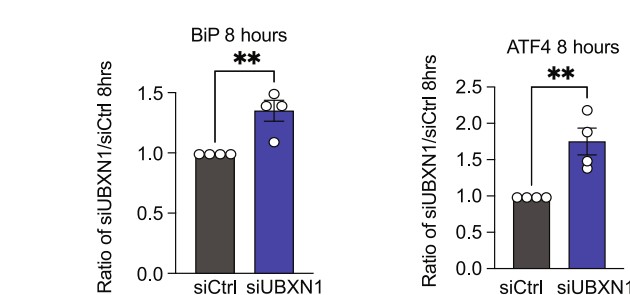

I.

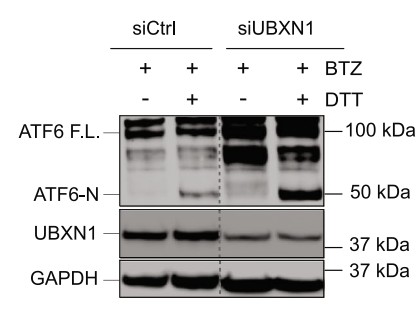

J.

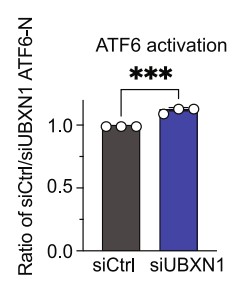

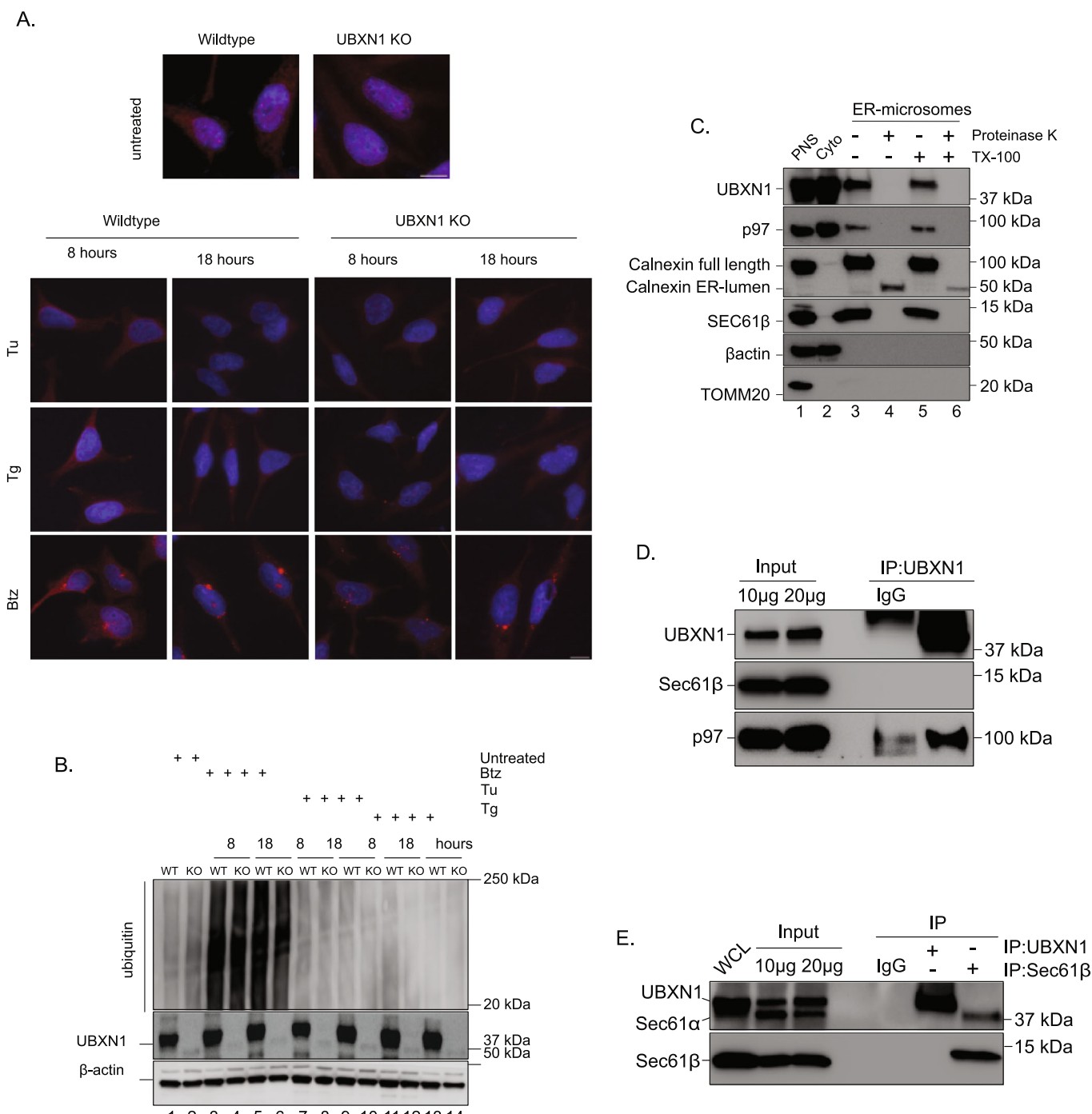

**Figure EV2.  UBXN1 localizes to ER-microsomes.**

(A) Immunofluorescent staining of ubiquitin (FK2) and nuclei (Hoechst dye) in HFT wildtype and UBXN1 KO cells. Cells were treated with 1 μM of the proteasome inhibitor bortezomib (BTZ), Tunicamycin (Tu), or Thapsigargin (Tg) for 8 hours or 1 μM BTZ, 500 nM Tu, or 500 nM Tg for 18 hours. (Scale bar: 10 μm) (B) Immunoblot assessing total ubiquitin levels corresponding to Figure EV2A. (C) Subcellular fractions enriched in ER-derived microsomes were isolated from HEK-293T cells by biochemical fractionation. Protease protection assay was used to localize UBXN1 to the ER periphery by immunoblot. Calnexin and Sec61β are ER specific markers, β-actin is a cytosolic marker, and TOMM20 is a mitochondrial marker. (PNS: post-nuclear supernatant sample, cyto: cytosolic sample). (D) Immunoblot for the immunoprecipitation of endogenous UBXN1 from HEK-293T lysates. 10 and 20 μg of whole cell lysate was used for the input. Sec61β was used as a marker for the Sec61 translocon. (E) UBXN1 and Sec61β were separately immunoprecipitated from ER-microsomes isolated from HEK-293T cells. 10 and 20 μg of whole cell lysate was used for the input.

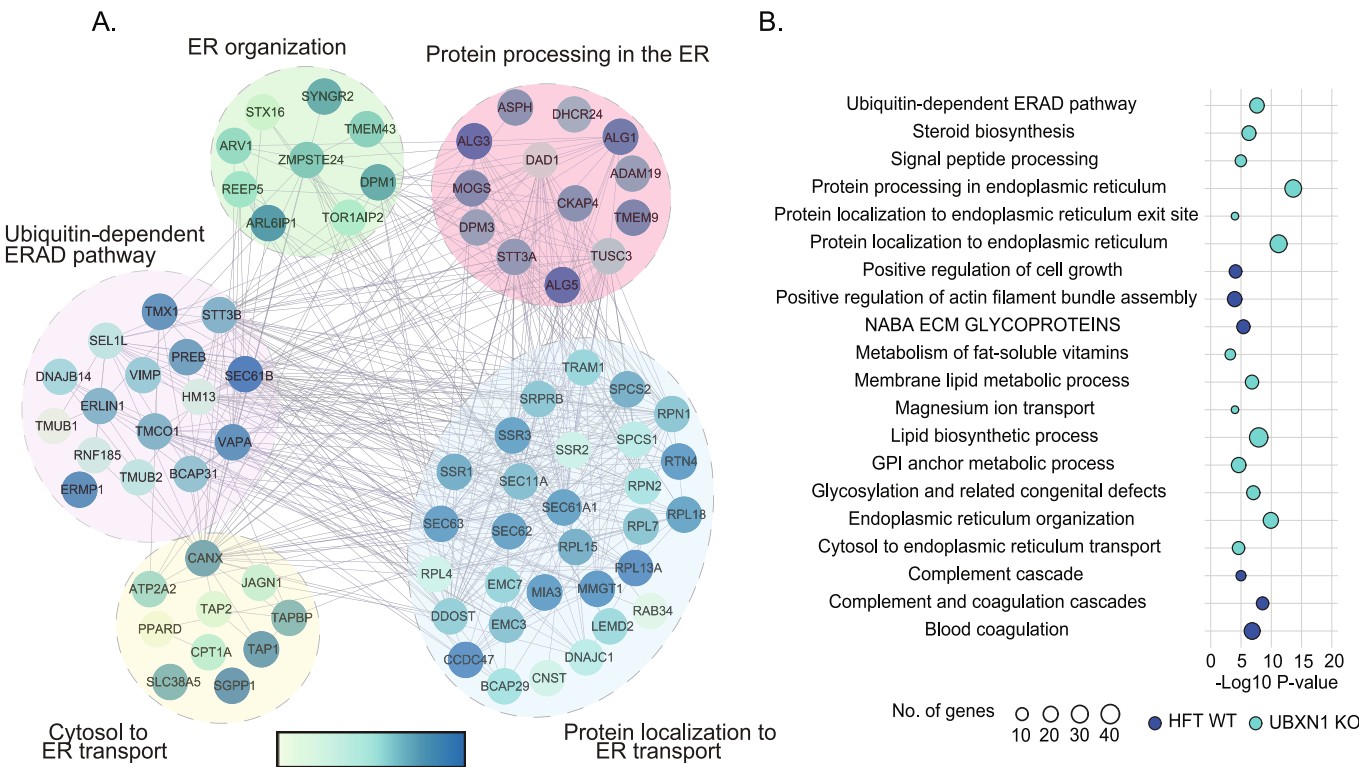

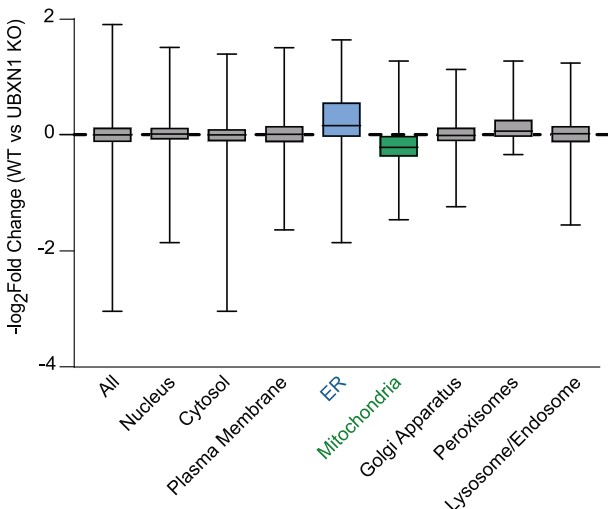

**Figure EV3.  Loss of UBXN1 leads to a perturbed ER proteome.**

(**A**) Interaction network of the gene ontology clusters from the wildtype/UBXN1 KO quantitative proteomics study with a significant increase in abundance in UBXN1 KO cells. Proteins found in each node are labeled and the color of each circle corresponds to the $-\log_{10}$-transformed $P$ value. (**B**) Bubble plot corresponding to significantly enriched gene ontology groups in the wildtype (navy) and UBXN1 KO (teal). The size of the circle corresponds to the number of genes identified in each term. (**C**) Proteins identified by proteomics categorized by organellar compartment. Proteins associated with the ER experience a significant increase in abundance in UBXN1 KO cells where mitochondrial proteins are significantly decreased $n =$ two biological replicates. Box plots show median -log fold change (wildtype: knockout), upper quartile represents the 75th percentile and lower quartile represents 25th percentile. The whiskers represent minimum and maximum -log fold change (wildtype: knockout) for each category.

A.

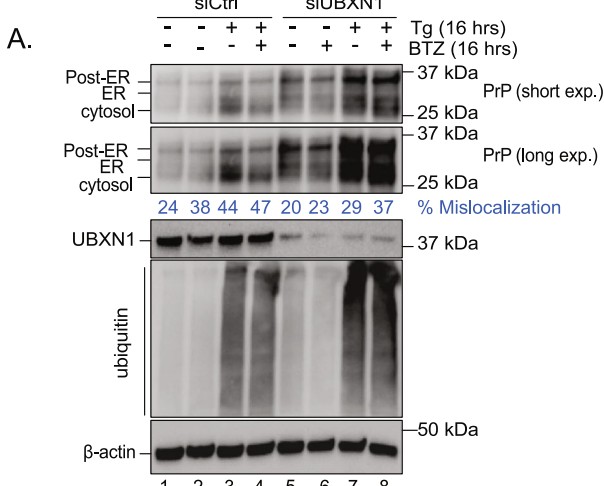

B.

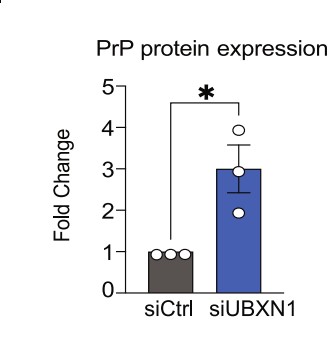

PrP protein expression

C.

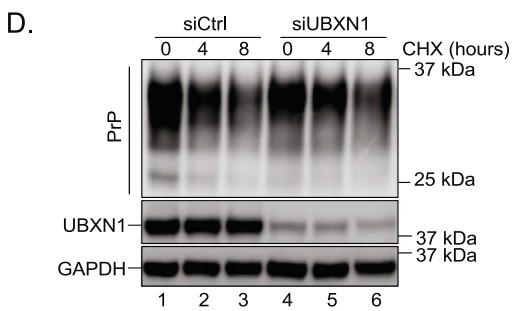

mRNA expression

F.

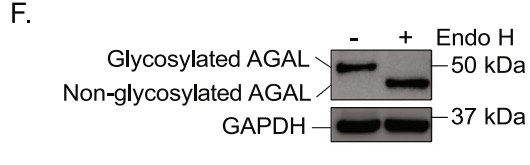

G.

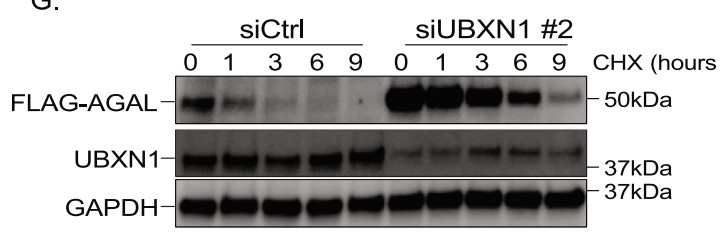

D.

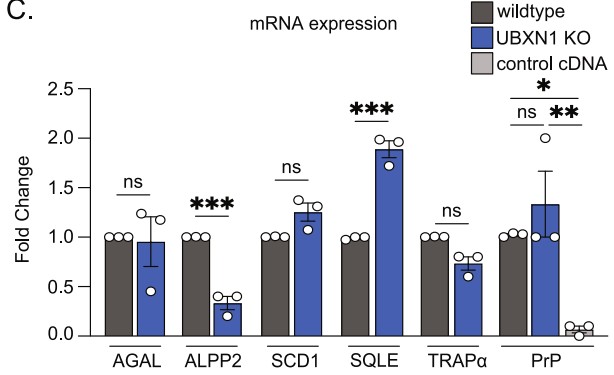

H.

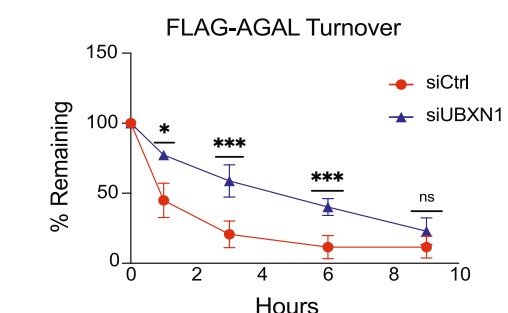

FLAG-AGAL Turnover

E.

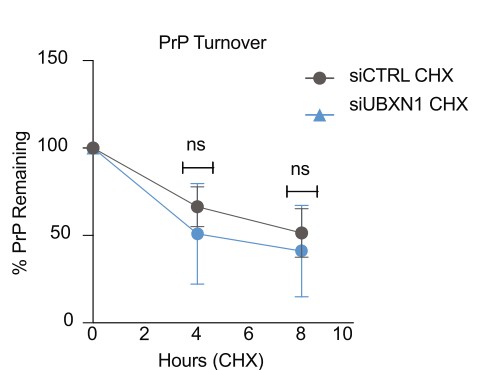

PrP Turnover

**Figure EV4.  Depletion of UBXN1 increases the expression of ER-destined proteins.**

(**A**) Immunoblot of HFT wildtype cells expressing HA-tagged prion protein (PrP). PrP was expressed in HeLa cells for before siRNA mediated depletion of UBXN1. Cells were treated with 500 nM thapsigargin, 1 μM bortezomib (BTZ), or both for 16 hours. The percent mislocalization was calculated from the ratio of the mislocalized (cytosolic) form to all forms of PrP for that sample. (**B**) Total PrP protein expression was calculated by the sum of all forms of PrP for that sample. The untreated condition was quantified (lanes 1 and 5) ($n$ = three biologically independent experiments). (**C**) Transcript levels of several ER-destined proteins from cells depleted of UBXN1 by siRNA quantified by real-time PCR. Control is cDNA generated from cells not expressing HA-PrP. ($n$ = three biologically independent experiments). (**D**) Immunoblot examining the turnover of HA-PrP in siRNA control and UBXN1 depleted cells. Cells were treated with 10 μg/mL cycloheximide (CHX) for the indicated timepoints (0–8 hours). (**E**) Quantification of PrP turnover from Figure EV4D. (**F**) Immunoblot of a stable HFT cell line expressing doxycycline inducible FLAG-tagged AGAL. Glycans were digested with EndoH for 4 hours to generate the non-glycosylated form. (**G**) Turnover of FLAG-AGAL in HEK293T cells depleted of UBXN1 and treated with cycloheximide to monitor half life. (**H**) Quantification of EV4G. For turnover studies, the zero-hour time point was set to 100% for each siRNA and the % of PrP or AGAL remaining for each time point was quantified as a ratio from the zero-hour timepoint. ($n$ = three biologically independent experiments). Data information: Data are means ± SEM (*, **, and *** where $P < 0.05$, 0.01, and 0.001 respectively.) Unpaired two-tailed $t$ test (**B, C, E, H**).

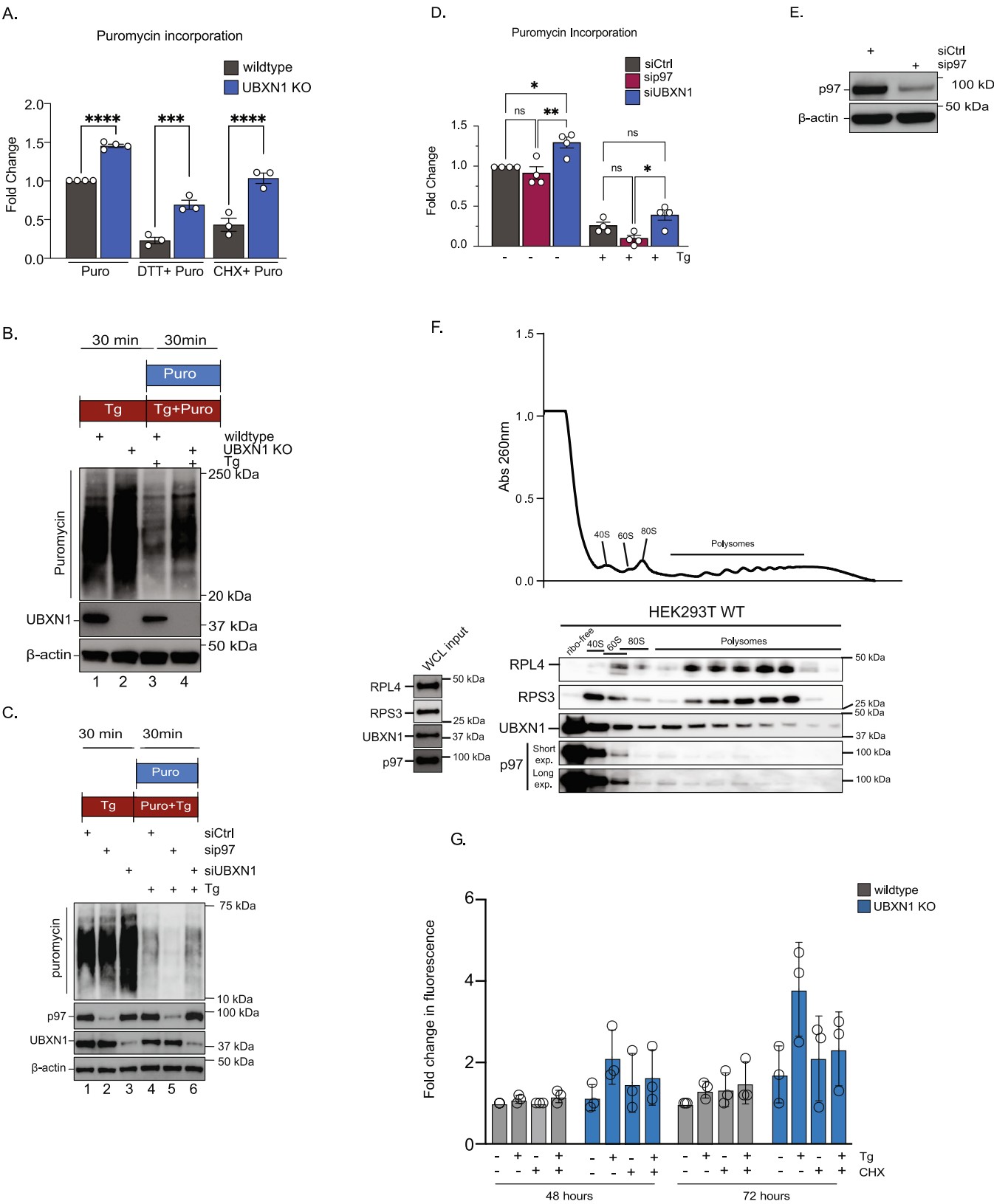

**Figure EV5.  UBXN1 suppresses protein synthesis.**

(A) Densitometry quantifications of the entire lanes corresponding to Fig. 7A ($n \geq$ three biologically independent experiments). (B) Immunoblot of HFT wildtype and UBXN1 KO cells pulsed with 1 µM puromycin for 30 min after pre-treatment with 1.5 µM thapsigargin for 30 minutes where indicated. (C) Immunoblot of 1 µM puromycin incorporation into control cells, or cells depleted of p97 or UBXN1 with siRNA for 48 hours. Cells were pre-treated with 1 µM thapsigargin for 30 minutes where indicated. (D) Quantification of the whole lane corresponding to each lane in the immunoblot in Figure EV5C ($n =$ four biologically independent samples). (E) Immunoblot validating the siRNA knockdown corresponding to Fig. 7F. (F) Immunoblot of ribosome and polysome fractions collected with corresponding UV traces showing 40S, 60S, 80S and polysome fractions. RPL4 is a marker for the 60S subunit and RPS3 is a marker for the 40S subunit. Whole cell lysates (WCL) are shown for input. (G) UBXN1 KO cells have similar viability to wildtype cells under resting conditions. Data shown in Main Fig. 8B is shown here without normalization to untreated in each genotype. We observe no differences in viability between wildtype and UBXN1 KO cells in untreated conditions allowing for this normalization. Fold change of the fluorescence measured by fluorescence-based cytotoxicity assay. Cells were treated with 1.5 µM thapsigargin in combination with 10 µg/ml cycloheximide where indicated. Values were normalized to wildtype untreated. ($n =$ three biologically independent samples). Data information: Data are means ± SEM (*, **, *** and **** where $P < 0.05$, 0.01, 0.001 and 0.0001, respectively.) Unpaired two-tailed $t$ test (A). One-way ANOVA with Tukey's multiple comparisons test (D, G).

