## [Peer Review File · EMBO Reports]

UBXN1 maintains ER proteostasis and represses UPR activation by modulating translation

Malavika Raman, Brittany Ahlstedt, Rakesh Ganji, Sirisha Mukkavalli, Joao Paulo, and Steven Gygi
DOI: 10.15252/embr.202357115

Corresponding author(s): Malavika Raman (malavika.raman@tufts.edu)

Review Timeline:

Transfer from Review Commons:	3rd Mar 23
Editorial Decision:	10th Mar 23
Revision Received:	2nd Jul 23
Editorial Decision:	25th Aug 23
Revision Received:	17th Oct 23
Editorial Decision:	20th Nov 23
Revision Received:	24th Nov 23
Accepted:	30th Nov 23

Transaction Report: This manuscript was transferred to EMBO reports following peer review at Review Commons.

Review
COMMONS

Review #1

1. Evidence, reproducibility and clarity:

Evidence, reproducibility and clarity (Required)

In this manuscript, Ahlstedt et al. study UBXN1, an adaptor of the p97/VCP AAA ATPase, using a cell line deficient for UBXN1. They found that the knockout of UBXN1 activates ER stress and sensitizes cells to ER stress-induced cell death. They used a proteomic approach to analyze the change in the global proteome in UBXN1 knockout cells. Interestingly, they found many proteins are upregulated in UBXN1 knockout cells, which appears to be regulated at a post-transcriptional level. Using puromycin labeling, they found that protein translation appears to be upregulated in UBXN1 knockout cells.

****Major comments:****

The conclusions of the manuscript are generally well supported by experimental data, which are of high quality. The presentation is clear. In my opinion, a few issues need to be addressed to further strengthen their conclusions.

1. The authors need to express UBXN1 and mutants lacking either the UBX or UBA domain in UBXN1 knockout cells to test whether the ER stress phenotype (Figure 1) and the protein upregulation phenotype (Figure 5A-F) can be rescued. This would eliminate the possibility that the reported phenotypes are the off-target effects of CRISPR.
2. For Figure 2, please indicate whether the repeat is a biological replicate or a technical replicate from RT-PCR.
3. In Figure 1A, the authors show that the knockout of UBXN1 causes an upregulation of phosphorylated eIF2alpha, which is known to suppress protein translation globally. In this regard, it is surprising to see the authors also concluded from Figure 7 that there is an upregulation of protein translation in UBXN1 knockout cells. The authors do not provide any explanation on how these seemingly contradictory phenotypes could be seen in the same cells.

2. Significance:

Significance (Required)

p97/VCP is an important member of the AAA ATPase family that has a variety of functions. It interacts with a collection of adaptor proteins that all contain a UBX

domain. These adaptors help to link the ATPase to the correct substrate in cells. The best-established function of p97/VCP is its role in ERAD, in which it acts together with its adaptors Ufd1-Npl4 and UBXD8 to extract retrotranslocated proteins from the ER for proteasomal degradation. UBXN1 is not required for ERAD. Instead, it appears to be a negative regulator of ERAD. Previous studies have also implicated it in mitophagy (Mengus C., Autophagy, 2022) and aggresome formation (from this group). Overall, the published studies did not pinpoint the precise cellular function of UBXN1.

This work characterizes the cellular phenotypes associated with UBXN1 loss of function. The information reported here is important, but the biological significance is limited. This is mainly because the authors entirely rely on a genetic approach. While the reported phenotypes associated with UBXN1 deficiency is solid, it is unclear what the underlying mechanisms are. It is not clear whether or not these phenotypes are interconnected, nor is it clear whether UBXN1 is a direct regulator of these processes. Taking the increased protein translation phenotype as an example, does this indicate UBXN1 is a translation suppressor for those ER-associated proteins? How can UBXN1 selectively inhibit the translation of a subset of proteins? Any evidence that UBXN1 is associated with translating ribosomes?

In summary, because of the limited mechanistic insights on UBXN1 function, the study may only be interesting to a specialized audience.

3. How much time do you estimate the authors will need to complete the suggested revisions:

Estimated time to Complete Revisions (Required)

(Decision Recommendation)

Between 1 and 3 months

Yes

Review #2

1. Evidence, reproducibility and clarity:

Evidence, reproducibility and clarity (Required)

RC-2022-01803

"UBXN1 maintains ER proteostasis and represses UPR activation by modulating translation independently of the p97 ATPase"

By Ahlstedt et al.

****Comments to the Author****

UBXN1 is a VCP adaptor UBX domain protein which is known to be involved in elimination of ubiquitylated cytosolic proteins bound to the BAG6 complex. In this study, authors demonstrated that cells depleted of UBXN1 have elevated UPR activation, even without external ER stresses. Cells devoid of UBXN1 have significant and global up-regulation of UPR-specific target genes, and these cells are more sensitive to ER stress than their wildtype counterparts. Using quantitative tandem mass tag proteomics of UBXN1 deleted cells, authors found that significant enrichment of the abundance of ER proteins involved in protein translocation, protein folding, quality control, and the ER stress response in an ERAD-independent manner. Notably, they observed no change in the abundance of proteins in the cytosol or nucleus, and significant decrease in the expression of several mitochondrial proteins when UBXN1 was depleted. Authors further demonstrate that UBXN1 is a translation repressor, and its UBA domain is critical for suppressing protein synthesis. Thus, increased influx of proteins into the ER in UBXN1 KO cells causes UPR activation. Authors concluded that they have identified a new regulator of protein translation and ER proteostasis.

My specific comments were provided as follows.

****Comments****

1. Authors found that significant enrichment of the ER proteins in UBXM1 KO cells, while there is no change in the abundance of proteins in the cytosol or nucleus. Mitochondrial proteins are even down-regulated in UBXM1 KO cells. I found these observations very interesting. However, I was frustrated that authors did not investigate the reason why such differences are associated in UBXM1-suppressed cells. Authors demonstrate that depletion of UBXM1 resulted in suppression of protein synthesis, but did not address whether ER proteins are specifically repressed by UBXM1 or it represses translation globally, as noted in their Discussion section. Do the mRNAs encoding signal sequence at the N-terminus of their products are specifically translated in UBXM1-suppressed cells? Do the translations of mRNAs encoding mitochondria translocation signals are suppressed in UBXM1 KO cells? It should be possible to investigate these issues by using appropriate model ER- or mitochondrial proteins with or without specific signal sequences. Such kind of analysis should be necessary to support the claim of this manuscript.
2. Related to my previous comments, ER-targeted mRNAs are known to be degraded by a process termed RIDD in the case of ER stressed condition. Since the rapid degradation of mRNAs through RIDD functions to alleviate ER stress by preventing the continued influx of new polypeptides into the ER, I wondered why UBXM1 depletion greatly stimulates ER protein synthesis, escaping IRE1-dependent mRNA degradations. Does UBXM1 depletion suppress RIDD?
3. Authors mentioned that the elevated levels of ER proteins are not due to increased transcription of target genes. However, they only provided the quantification of prp transcript levels, which was unchanged between wildtype and UBXM1 KO cells. To support this important conclusion, it is necessary to provide whole transcriptome data to compare the expression levels of corresponding ER proteins (quantified by their proteomics data) and transcripts (quantified by, for an example, RNA-seq analysis).
4. Authors claimed that UBXM1 loss is detrimental to cell viability and have elevated levels of the apoptosis in the face of ER stress. However, authors did not examine apoptotic cell death in UBXM1 KO cells. They only provided evidence for defective proliferation of cells and transient induction of CHOP expression, but these are not enough to support the ER-stress induced apoptosis.
5. Authors showed that UBA domain of UBXM1 is critical for suppressing protein synthesis. Could you provide a bit more detailed discussion how UBA domain modulates protein translational events and promote expressions of ER-related proteins. Have you ever checked whether UBA domain of UBXM1 is necessary for suppressing UPR-specific target gene expressions?

2. Significance:

Significance (Required)

Although the discovery in this manuscript might be potentially interesting for broad audience, the presented study did not provide enough mechanistic insights and their data lacks vital evidences to support their conclusion. I found that the data are preliminary to discuss the validity of this finding. The inadequacy of these points makes this manuscript unsuitable for publication at this stage.

My expertise is cell biology and biochemistry for protein quality control.

3. How much time do you estimate the authors will need to complete the suggested revisions:

Estimated time to Complete Revisions (Required)

(Decision Recommendation)

More than 6 months

Yes

Review #3

1. Evidence, reproducibility and clarity:

Evidence, reproducibility and clarity (Required)

****Summary:****

Ahlstedt et al. investigate a new role for the p97 adapter protein UBXN1 in negatively regulating the ER unfolded protein response. The study starts from the observations that knockdown of UBXN1 in a previously generated HeLa cell line leads to induction of unfolded protein response markers, and the knockout cells display more pronounced UPR activation upon ER stress. This elevated UPR signaling renders the UBXN1 cells more prone to cell death. Global proteomics experiments similarly show an increased abundance of ER localized proteins, although it is not clearly delineated which of those are the result of UPR activation. The authors then probe the expression of two secretory client proteins, alpha-galactosidase (AGAL) and prion protein (PrP) and find that UBXN1 transient knockdown leads to ER accumulation of the two proteins and increased aggregation upon ER stress. The authors claim that degradation of these ER client proteins is unaffected by the UBXN1 knockdown, but accumulation may instead be due to increased protein translation. Indeed, they surprisingly find that UBXN1 knockout leads to constitutively elevated protein translation. This result points to a previously unknown role of UBXN1 in repressing protein synthesis. Complementation with UBXN1 mutants demonstrate that the translation repression is dependent on the ubiquitin binding activity of UBXN1 but that p97 is dispensable. Further investigation into the molecular mechanism for the translation repression remains reserved for a future manuscript.

****Major comments:****

1. My main reservation about the current manuscript is whether the UPR activation can be directly ascribed to the loss of UBXN1. The authors do not differentiate between acute depletion (through siRNA in Fig. 5) versus permanent UBXN1 knockout in most of the experiments. The latter may lead to extensive adaptation of the cellular proteome due to chronic stress. Prior studies from the authors have shown that UBXN1 knockout leads to loss of aggreasomes. This raises a major question whether the observed UPR activation can be directly attributed to UBXN1 loss or be an indirect result of adaptation in the knockout cells, for instance due to accumulation of BAG6 substrates in insoluble aggregates as the authors have shown previously (ref. 40). Along those lines, the authors already showed in the same study that UBXN1 knockout cells are more sensitive to proteotoxic stress.

2. The later results in the study nicely show that the repressed protein translation phenotype is dependent on the ubiquitin binding domain of UBXN1. These segregation-of-function mutants and complementation experiments could be easily used to more clearly distinguish whether the UPR activation can be directly attributed to UBXN1 and the increase in protein translation. For instance, can overexpression of UBXN1 in the knockout background suppress the UPR activation? Is the UBX-domain mutant capable of suppressing the UPR phenotype? These results would provide critical support as to whether the UPR activation is a direct result of the loss

of UBXN1.

3. Similarly, the authors use transient siRNA knockdown of UBXN1 in Fig. 5 and Supp. Fig. 4, but do not reassess the UPR activation under these conditions. It would be important to validate that the acute UBXN1 knockdown can recapitulate the UPR activation phenotype.

4. I am puzzled by the interpretation of the AGAL degradation experiments in Supplemental Figure 4F. Clearly, the rate of AGAL degradation is much faster in WT cells than in UBXN1 knockout cells as indicated by the slope of the curves between 2-4 hours. I disagree with the interpretation that UBXN1 knockout does not impact AGAL turnover. It is not valid to make the comparison at 9 hours because hardly any AGAL substrate is remaining. Importantly, this experiment raises a larger question: Are other ER client degradation rates affected by the UBXN1 knockout? And is the UPR activation more generally due to accumulation of misfolded ER proteins? Their prior publication (ref. 40) evaluated several ERAD clients where UBXN1 was dispensable, but it could be possible that UBXN1 has a more specialized client pool. Showing quantification of the PrP CHX chase would also be helpful - from the single replicate it looks like more PrP remaining in the UBXN1 knockout at 8 hours (Supp. Figure 4G).

5. It would be helpful for the manuscript to clearly distinguish between 1) upregulation of ER proteostasis factors because of ER stress/UPR, and 2) upregulation of secreted clients (AGAL, PrP) which may be partly due to increased translation rates but could also be due to reduced degradation. Many of the hits from the proteomics experiments are ER proteostasis factors that are part of the adaptive stress response (SEC61B, SEC63, CANX, SSR1/2/3, STT3B, RPN1, RPN2, SEC61A1 - compare to ref 12: most are direct IRE1/XBP1s targets). Their increased expression does not lead to increased ER stress as they are involved in the resolution of ER stress. It appears to be circular logic that increased expression of UPR targets would lead to more UPR activation. Currently, the authors do not clearly disentangle the increased expression of endogenous ER proteins from the proteomics experiment versus overexpression of exogenous secreted clients.

6. The authors should tone down on broad generalizations, for instance in lines 306-309: ER aggregation was only observed for a single client protein (AGAL). Further, only a single mitochondrial protein was observed to be downregulated (TOMM20).

****Minor comments****

- Does UBXN1 localization to the ER/microsomes fraction depend on p97? What happens in UBX-domain mutant?

- In Fig. 1A it is surprising that no BiP is detected at 0 hours as BiP is highly expressed even in the absence of ER stress. Can the authors comment on this discrepancy.

- The authors use different ER stressors interchangeably: DTT, Tunicamycin, Thapsigargin. While all results in UPR activation, they do so through different mechanisms and with slight nuances that may be worth considering for the experiments and interpretations.
- Line 198: "Hierarchical clustering analysis demonstrates that the gene expression pattern observed in UBXN1 KO cells more closely resembles wildtype cells stressed with DTT than untreated wildtype cells based on similar log2 fold change values (Figure 2)." Where is this clustering shown?
- What are the downregulated UPR genes in Fig. 2, and may this hold significance?

2. Significance:

Significance (Required)

General assessment: The authors broadly characterize the UPR activation in the UBXN1 knockout cells, looking both at gene targets by Western blot and qPCR, and characterize the activation of individual sensors (ATF6 cleavage and IRE1alpha clustering). Proteomics results further corroborate the upregulation of ER-localized proteins, although the robustness of the findings is surprising considering that only 2 replicates were included in the mass spectrometry experiment. Most other experiments are technically sound, for instance the puromycylation translation assays. One of the key limitations of the is that the authors fail to make use of their extensive prior toolset on UBXN1, particularly the segregation-of-function mutations for p97 and ubiquitin binding, as well as the knockdown cell lines with inducible overexpression of UBXN1 to rescue the phenotypes. These tools could probe a direct involvement of UBXN1 in the UPR repression, and whether this activity is truly independent of p97. A related limitation is that results are often over-interpreted and too far generalized (see examples above), or wrongly interpreted (see AGAL degradation rates).

Advance: The AAA+ ATPase VCP/p97 has many divergent cellular roles that are in part mediated by a variety of different adaptor proteins. The authors have previously discovered the important role for UBXN1 in recruiting p97 to mislocalized cytosolic proteins targeted to the BAG6 complex. The current study now aims to establish a new role for UBXN1 in regulating the unfolded protein response. As it stands, the findings that UBXN1 knockdown results in UPR activation and impacts translation rates are solid but largely descriptive in nature. These findings merit reporting but require that the authors tone down their conclusions about a direct role for UBXN1 as a regulator of the UPR. Alternatively, if the authors choose to stick with their current model for a direct involvement of UBXN1, they need to establish the mechanistic link more clearly.

Audience: In the current form, the manuscript should appeal to a broad biochemistry and cell biology readership interested in topics related to proteostasis, protein quality control, and stress signaling.

3. How much time do you estimate the authors will need to complete the suggested revisions:

Estimated time to Complete Revisions (Required)

(Decision Recommendation)

Between 3 and 6 months

Yes

Revision Plan

Manuscript number: RC-2022-01803

Corresponding author(s): Brittany A., Ahlstedt, Rakesh, Ganji, Sirisha Mukkavalli, Joao A., Paulo, Steve P., Gygi, Malavika, Raman

*If you wish to submit a full revision, please use our "Full Revision" template. **It is important to use the appropriate template to clearly inform the editors of your intentions.**]*

1. General Statements [optional]

This section is optional. Insert here any general statements you wish to make about the goal of the study or about the reviews.

We thank the reviewers for their insightful comments and agree that the many of these revision experiments will improve the strength of our manuscript. Some of these we have already completed or are in the process of completing which will be outlined below. In particular, the reviewers asked that we investigate the mechanistic link between increased translation and UPR induction. We have detailed the studies that we will perform to establish this connection in detail below.

2. Description of the planned revisions

Response to Reviewer 1:

1. The authors need to express UBXN1 and mutants lacking either the UBX or UBA domain in UBXN1 knockout cells to test whether the ER stress phenotype (Figure 1) and the protein upregulation phenotype (Figure 5A-F) can be rescued. This would eliminate the possibility that the reported phenotypes are the off-target effects of CRISPR.

- We will express the Myc-tagged wildtype UBXN1 and UBX or UBA point mutants (used in translation rescue studies in Figure 6) into UBXN1 knockout (KO) cells to determine whether the ER stress phenotype can be rescued. We will determine the level of *xbp1s* by real-time PCR and BiP by immunoblot.
- The studies in Figure 5 A-F were completed in cells depleted of UBXN1 with siRNA, not the CRISPR knockout cells. Thus, it is unlikely an off-target effect of CRISPR. We will attempt rescue of this phenotype with the wildtype and mutant constructs.

Revision Plan

2. **For Figure 2, please indicate whether the repeat is a biological replicate or a technical replicate from RT-PCR.**
 - We apologize for the omission. The data from the RT PCR studies in Figure 2 are biological replicates – the figure legend and main text of the manuscript will be edited to clarify this.

3. **In Figure 1A, the authors show that the knockout of UBXN1 causes an upregulation of phosphorylated eIF2alpha, which is known to suppress protein translation globally. In this regard, it is surprising to see the authors also concluded from Figure 7 that there is an upregulation of protein translation in UBXN1 knockout cells. The authors do not provide any explanation on how these seemingly contradictory phenotypes could be seen in the same cells.**
 - We will provide a detailed discussion of the apparent paradox between upregulation of phosphorylated eIF2 α and increased protein translation. Several prior studies have demonstrated that elevated expression of ATF4 (as we observe in UBXN1 KO cells) activates a transcriptional program that restarts translation. This occurs through the upregulation of the phosphatase PPP1R15a that dephosphorylates eIF2 α , as well as aminoacyl tRNA synthetases and ribosomal subunits. We propose that elevated ATF4 levels leads to premature translational restart in UBXN1 KO cells. In addition, our data suggests that UBXN1 represses translation upstream of UPR activation and thus an increase in protein translation dysregulates ER-proteostasis which hyperactivates the UPR.

4. **Any evidence that UBXN1 is associated with translating ribosomes?**
 - We now have new data that UBXN1 is associated with 40S, 60S, and 80S ribosomal fractions as well as actively translating polysome fractions that we isolated by polysome purification. In agreement with our finding that the role of UBXN1 in repressing translation is independent of p97, p97 appears to associate largely with the 40S, 60S, and 80S ribosomal fractions but not with the actively translating polysomes. This data will be included in the revised manuscript.

Response to Reviewer 2:

1. **Authors found that significant enrichment of the ER proteins in UBXN1 KO cells, while there is no change in the abundance of proteins in the cytosol or nucleus. Mitochondrial proteins are even down-regulated in UBXN1 KO cells. I found these observations very interesting. However, I was frustrated that authors did not investigate the reason why such differences are associated in**

UBXN1-suppressed cells. Authors demonstrate that depletion of UBXN1 resulted in suppression of protein synthesis, but did not address whether ER proteins are specifically repressed by UBXN1 or it represses translation globally, as noted in their Discussion section. Do the mRNAs encoding signal sequence at the N-terminus of their products are specifically translated in UBXN1-suppressed cells? Do the translations of mRNAs encoding mitochondria translocation signals are suppressed in UBXN1 KO cells? It should be possible to investigate these issues by using appropriate model ER- or mitochondrial proteins with or without specific signal sequences. Such kind of analysis should be necessary to support the claim of this manuscript.

- Previous studies by Luke Wiseman's group showed that PERK activation resulted in hyperfusion of the mitochondria and loss of Tim17 leading to decreased mitochondrial import. We already show that mitochondrial proteins are downregulated (by TMT proteomics and by immunoblotting). We now have preliminary data that mitochondria are more fused in UBXN1 KO cells consistent with data from the Wiseman group. We will include this in the resubmission.
 - In addition, we have re-analyzed our TMT proteomics data to parse out proteins with ER-signal sequences and define the topology of ER proteins (Type 1, 2, multimembrane spanning and luminal proteins) and those with mitochondrial targeting sequences. This data will be included in the revised manuscript.
- 2. Related to my previous comments, ER-targeted mRNAs are known to be degraded by a process termed RIDD in the case of ER stressed condition. Since the rapid degradation of mRNAs through RIDD functions to alleviate ER stress by preventing the continued influx of new polypeptides into the ER, I wondered why UBXN1 depletion greatly stimulates ER protein synthesis, escaping IRE1-dependent mRNA degradations. Does UBXN1 depletion suppress RIDD?**
- In the revised manuscript, we will determine the relative mRNA abundance of the bona fide RIDD targets BLOC1S1 and CD59 by quantitative PCR in cells stressed with dithiothreitol (DTT). We will utilize previously published and validated primers for each target to quantify RIDD activity in wildtype and UBXN1 KO cells. These studies will address whether loss of UBXN1 impacts IRE1-dependent RIDD.
- 3. Authors mentioned that the elevated levels of ER proteins are not due to increased transcription of target genes. However, they only provided the quantification of prp transcript levels, which was unchanged between wildtype and UBXN1 KO cells. To support this important conclusion, it is necessary to**

Revision Plan

provide whole transcriptome data to compare the expression levels of corresponding ER proteins (quantified by their proteomics data) and transcripts (quantified by, for an example, RNA-seq analysis).

- We thank the reviewer for this comment. Currently, we show that mRNA levels of *Prp* do not significantly change between control and siUBXN1 cells (Supplementary Figure 4). For a more comprehensive analysis, we will additionally assess the mRNA levels of the proteins we determined to be significantly enriched in Figure 5 (AGAL, ALPP2 and TRAP α). RNA sequencing is currently beyond the scope of this study.
4. **Authors claimed that UBXN1 loss is detrimental to cell viability and have elevated levels of the apoptosis in the face of ER stress. However, authors did not examine apoptotic cell death in UBXN1 KO cells. They only provided evidence for defective proliferation of cells and transient induction of CHOP expression, but these are not enough to support the ER-stress induced apoptosis.**
- We will address the levels of apoptotic cell death in wildtype and UBXN1 KO cells by assessing PARP, caspase-3, or caspase-8 cleavage in these cells by immunoblot.
5. **Authors showed that UBA domain of UBXN1 is critical for suppressing protein synthesis. Could you provide a bit more detailed discussion how UBA domain modulates protein translational events and promote expressions of ER-related proteins. Have you ever checked whether UBA domain of UBXN1 is necessary for suppressing UPR-specific target gene expressions?**
- We will express the Myc-tagged wildtype UBXN1 and UBX or UBA point mutants (used in translation rescue studies in Figure 6) into UBXN1 knockout (KO) cells to determine whether the ER stress phenotype can be rescued.
 - We will also include a discussion on how the UBA domain in UBXN1 may recognize distinct ubiquitylation events on ribosomes that modulate their abundance and function.

Response to Reviewer 3: (Major comments)

1. **My main reservation about the current manuscript is whether the UPR activation can be directly ascribed to the loss of UBXN1. The authors do not differentiate between acute depletion (through siRNA in Fig. 5) versus permanent UBXN1 knockout in most of the experiments. The latter may lead to**

extensive adaptation of the cellular proteome due to chronic stress. Prior studies from the authors have shown that UBXN1 knockout leads to loss of aggreasomes. This raises a major question whether the observed UPR activation can be directly attributed to UBXN1 loss or be an indirect result of adaptation in the knockout cells, for instance due to accumulation of BAG6 substrates in insoluble aggregates as the authors have shown previously (ref. 40). Along those lines, the authors already showed in the same study that UBXN1 knockout cells are more sensitive to proteotoxic stress.

- We agree with the reviewer that cells can adapt to CRISPR knockout. However, the IRE1 α clustering studies found in Figure 1 were completed in the context of acute depletion of UBXN1 by siRNA and demonstrate a significant increase in IRE1 α clustering when UBXN1 is depleted.
 - We now have new data that that acute depletion of UBXN1 with siRNA results in a significant increase in BiP and ATF4 expression as well as ATF6 N-terminal processing.
 - Furthermore, we have new data that acute depletion of UBXN1 with siRNA phenocopies UBXN1 KO in terms of increased puromycin incorporation into newly synthesized proteins.
 - Thus, we will have both genetic knockout as well as siRNA acute depletion for all major studies. We will include these new studies in the revised manuscript.
- 2. The later results in the study nicely show that the repressed protein translation phenotype is dependent on the ubiquitin binding domain of UBXN1. These segregation-of-function mutants and complementation experiments could be easily used to more clearly distinguish whether the UPR activation can be directly attributed to UBXN1 and the increase in protein translation. For instance, can overexpression of UBXN1 in the knockout background suppress the UPR activation? Is the UBX-domain mutant capable of suppressing the UPR phenotype? These results would provide critical support as to whether the UPR activation is a direct result of the loss of UBXN1.**
- We will express the Myc-tagged wildtype UBXN1 and UBX or UBA point mutants (used in translation rescue studies in Figure 6) into UBXN1 knockout (KO) cells to determine whether the ER stress phenotype can be rescued. We will determine the level of *xbp1s* by real-time PCR and BiP by immunoblot.
 - To delineate the relationship between UPR activation and protein translation, we will halt protein synthesis with the translational elongation inhibitor cycloheximide and assess UPR activation in wildtype and UBXN1 KO cells. If increased protein translation in UBXN1 KO cells is what causes UPR

Revision Plan

activation, we anticipate that cycloheximide will rescue UPR activation in UBXN1 KO cells back to wildtype levels.

3. **Similarly, the authors use transient siRNA knockdown of UBXN1 in Fig. 5 and Supp. Fig. 4, but do not reassess the UPR activation under these conditions. It would be important to validate that the acute UBXN1 knockdown can recapitulate the UPR activation phenotype.**
 - Please see comment 1 above.

4. **I am puzzled by the interpretation of the AGAL degradation experiments in Supplemental Figure 4F. Clearly, the rate of AGAL degradation is much faster in WT cells than in UBXN1 knockout cells as indicated by the slope of the curves between 2-4 hours. I disagree with the interpretation that UBXN1 knockout does not impact AGAL turnover. It is not valid to make the comparison at 9 hours because hardly any AGAL substrate is remaining. Importantly, this experiment raises a larger question: Are other ER client degradation rates affected by the UBXN1 knockout? And is the UPR activation more generally due to accumulation of misfolded ER proteins? Their prior publication (ref. 40) evaluated several ERAD clients where UBXN1 was dispensable, but it could be possible that UBXN1 has a more specialized client pool. Showing quantification of the PrP CHX chase would also be helpful - from the single replicate it looks like more PrP remaining in the UBXN1 knockout at 8 hours (Supp. Figure 4G).**
 - Our previous ERAD reporter study using three distinct ERAD clients that are routinely used to assess ERAD found no role for UBXN1 in ERAD (Ganji et al MCB 2018). We do agree with the reviewer that UBXN1 may have discrete roles in regulating the degradation of select p97 ER clients. Determining this in an unbiased and comprehensive manner would require pulse chase SILAC proteomics or similar methodologies which are beyond the scope of the current study. We will therefore evaluate whether loss of UBXN1 affects the rate of degradation of additional ER-client proteins that we identified via TMT. Additionally, we will include a quantification of PrP cycloheximide chase.

5. **It would be helpful for the manuscript to clearly distinguish between 1) upregulation of ER proteostasis factors because of ER stress/UPR, and 2) upregulation of secreted clients (AGAL, PrP) which may be partly due to increased translation rates but could also be due to reduced degradation. Many of the hits from the proteomics experiments are ER proteostasis factors that are part of the adaptive stress response (SEC61B, SEC63, CANX, SSR1/2/3, STT3B, RPN1, RPN2, SEC61A1 - compare to ref 12: most are direct IRE1/XBP1s targets). Their increased expression does not lead to increased**

ER stress as they are involved in the resolution of ER stress. It appears to be circular logic that increased expression of UPR targets would lead to more UPR activation. Currently, the authors do not clearly disentangle the increased expression of endogenous ER proteins from the proteomics experiment versus overexpression of exogenous secreted clients.

- We identify many ER proteins with increased abundance in UBXN1 KO cells that are not transcriptional targets of the IRE1-UPR pathway. We will re-format the TMT data to more comprehensively characterize the proteins that we identify (known UPR transcriptional targets, membrane embedded, soluble clients etc.).
- We will change the language in the current manuscript to clearly demarcate the difference between an increase of ER proteostasis factors in response to ER stress, and the upregulation of secreted proteins. Additionally, we will emphasize the secretory proteins that are significantly enriched in UBXN1 KO cells in our proteomics figures to demonstrate the increase of non-ER stress responsive clients.

6. The authors should tone down on broad generalizations, for instance in lines 306-309: ER aggregation was only observed for a single client protein (AGAL). Further, only a single mitochondrial protein was observed to be downregulated (TOMM20).

- We have included the quantifications of the relative expression levels of three mitochondrial proteins, two of which are significantly reduced (TOMM20 and CYC1).
- Additionally, we have new data where we immunoblotted for additional mitochondrial import factors and observed significant reduction of the mitochondrial proteins TIMM23 and TOMM70A which will be included in the revised manuscript.
- We also plan to examine the levels of the TIMM17A subunit of the TIMM23 complex in UBXN1-depleted cells. TIMM17A is degraded in response to ER stress to prevent protein import into the mitochondria. (Rainbolt, T. et al. Cell Metab 2013)
- The language of the manuscript will be changed to tone down on broad generalizations.

(Minor comments)

7. Does UBXN1 localization to the ER/microsomes fraction depend on p97? What happens in UBX-domain mutant?

Revision Plan

- We will isolate ER-microsomes from UBXN1 KO cells where we have expressed wildtype and UBX/UBA domain mutants to address if localization is dependent on ubiquitin or p97 interaction.
8. **In Fig. 1A it is surprising that no BiP is detected at 0 hours as BiP is highly expressed even in the absence of ER stress. Can the authors comment on this discrepancy.**
- We provide low exposures of the immunoblots as the UBXN1 KO cells have very high levels of BiP compared to control. We will provide alternative blots where the BiP levels at t=0 in control cells is more obvious.
9. **The authors use different ER stressors interchangeably: DTT, Tunicamycin, Thapsigargin. While all results in UPR activation, they do so through different mechanisms and with slight nuances that may be worth considering for the experiments and interpretations.**
- We thank the reviewer for this comment and agree that these stressors can impact the ER and UPR activation in distinct ways. Our rationale for using these agents interchangeably was to demonstrate the UPR induction in UBXN1 null cells occurs irrespective of the type of stress.
 - DTT is a severe stressor and we used tunicamycin and thapsigargin in some assays (imaging etc.) as they are less toxic and more amenable to downstream analysis. We will include text that explains our rationale better.
10. **Line 198: "Hierarchical clustering analysis demonstrates that the gene expression pattern observed in UBXN1 KO cells more closely resembles wildtype cells stressed with DTT than untreated wildtype cells based on similar log2 fold change values (Figure 2)." Where is this clustering shown?**
- We apologize that this was not clear in the figure. We will edit the figure to make the clustering more obvious.
11. **What are the downregulated UPR genes in Fig. 2, and may this hold significance?**
- The reviewer points out an interesting observation. Many of the downregulated transcripts are ERAD components. The significance of this is presently unclear and we would require RNA-seq analysis to make a more educated conclusion. However, this finding may point to an environment that

Revision Plan

has a greater need to induce folding than degradative components. We will include a discussion of this in the revision.

3. Description of the revisions that have already been incorporated in the transferred manuscript.

Please insert a point-by-point reply describing the revisions that were already carried out and included in the transferred manuscript. If no revisions have been carried out yet, please leave this section empty.

4. Description of analyses that authors prefer not to carry out

Please include a point-by-point response explaining why some of the requested data or additional analyses might not be necessary or cannot be provided within the scope of a revision. This can be due to time or resource limitations or in case of disagreement about the necessity of such additional data given the scope of the study. Please leave empty if not applicable.

Response to Reviewer 1:

1. Taking the increased protein translation phenotype as an example, does this indicate UBXN1 is a translation suppressor for those ER-associated proteins?

- We thank the reviewer for this comment. We are indeed very interested in determining whether UBXN1 represses the translation of ER proteins. We are in the process of identifying proteins that are translated in UBXN1 null cells using O-propargyl-puromycin (OPP) labelling and mass spectrometry. However, given the timeframe for these studies, this cannot be accomplished in this revision.

2. How can UBXN1 selectively inhibit the translation of a subset of proteins?

- Recent studies suggest that ribosome populations are quite heterogeneous, and ribosome associated proteins can help tune translation of select proteins. For example, pyruvate kinase muscle (PKM) associates with ER docked ribosomes to regulate the translation of ER proteins in particular. We find that UBXN1 is present on ER membranes and localizes to polysomes and thus may regulate the translation of specific proteins. Studies are underway to test this hypothesis but are beyond the scope for this present study.

Response to Reviewer 2:

Revision Plan

- 1. Authors found that significant enrichment of the ER proteins in UBXN1 KO cells, while there is no change in the abundance of proteins in the cytosol or nucleus. Mitochondrial proteins are even down-regulated in UBXN1 KO cells. I found these observations very interesting. However, I was frustrated that authors did not investigated the reason why such differences are associated in UBXN1-suppressed cells. Authors demonstrate that depletion of UBXN1 resulted in suppression of protein synthesis, but did not address whether ER proteins are specifically repressed by UBXN1 or it represses translation globally, as noted in their Discussion section. Do the mRNAs encoding signal sequence at the N-terminus of their products are specifically translated in UBXN1-suppressed cells? Do the translations of mRNAs encoding mitochondria translocation signals are suppressed in UBXN1 KO cells? It should be possible to investigate these issues by using appropriate model ER- or mitochondrial proteins with or without specific signal sequences. Such kind of analysis should be necessary to support the claim of this manuscript.**

- Please see comment 1 above.

Dear Dr. Raman

Thank you for the submission of your research manuscript to our journal. I have now had the chance to read your manuscript, the referee reports from Review Commons and your revision plan. I agree that the proposed role of UBXN1 in translational repression is potentially interesting and I invite you to revise your manuscript along the lines detailed in your revision plan for potential publication in EMBO reports. It will be important to provide further data that links the proposed role of UBXN1 in translation to the observed increase in ER proteins and ER stress, as planned.

Please address all referee concerns in a complete point-by-point response. Acceptance of the manuscript will depend on a positive outcome of a second round of review. It is EMBO reports policy to allow a single round of revision only and acceptance or rejection of the manuscript will therefore depend on the completeness of your responses included in the next, final version of the manuscript.

We realize that it is difficult to revise to a specific deadline. In the interest of protecting the conceptual advance provided by the work, we recommend a revision within 3 months (June 10th). Please discuss the revision progress ahead of this time with the editor if you require more time to complete the revisions.

I am also happy to discuss the revision further via e-mail or a video call, if you wish.

*******IMPORTANT NOTE:**

We perform an initial quality control of all revised manuscripts before re-review. Your manuscript will FAIL this control and the handling will be DELAYED if the following APPLIES:

- 1) A data availability section providing access to data deposited in public databases is missing. If you have not deposited any data, please add a sentence to the data availability section that explains that.
- 2) Your manuscript contains statistics and error bars based on $n=2$. Please use scatter blots in these cases. No statistics should be calculated if $n=2$.

When submitting your revised manuscript, please carefully review the instructions that follow below. Failure to include requested items will delay the evaluation of your revision.*****

- 1) a .docx formatted version of the manuscript text (including legends for main figures, EV figures and tables). Please make sure that the changes are highlighted to be clearly visible.
- 2) individual production quality figure files as .eps, .tif, .jpg (one file per figure). Please download our Figure Preparation Guidelines (figure preparation pdf) from our Author Guidelines pages <https://www.embopress.org/page/journal/14693178/authorguide> for more info on how to prepare your figures.
- 3) a .docx formatted letter INCLUDING the reviewers' reports and your detailed point-by-point responses to their comments. As part of the EMBO Press transparent editorial process, the point-by-point response is part of the Review Process File (RPF), which will be published alongside your paper.
- 4) a complete author checklist, which you can download from our author guidelines (). Please insert information in the checklist that is also reflected in the manuscript. The completed author checklist will also be part of the RPF.
- 5) Please note that all corresponding authors are required to supply an ORCID ID for their name upon submission of a revised manuscript (). Please find instructions on how to link your ORCID ID to your account in our manuscript tracking system in our Author guidelines

()

6) We replaced Supplementary Information with Expanded View (EV) Figures and Tables that are collapsible/expandable online. A maximum of 5 EV Figures can be typeset. EV Figures should be cited as 'Figure EV1, Figure EV2' etc... in the text and their respective legends should be included in the main text after the legends of regular figures.

7) Before submitting your revision, primary datasets (and computer code, where appropriate) produced in this study need to be deposited in an appropriate public database (see < <https://www.embopress.org/page/journal/14693178/authorguide#dataavailability>>).

Specifically, we would kindly ask you to provide public access to the proteomics datasets.

The accession numbers and database should be listed in a formal "Data Availability " section (placed after Materials & Method) that follows the model below (see also < <https://www.embopress.org/page/journal/14693178/authorguide#dataavailability>>). Please note that the Data Availability Section is restricted to new primary data that are part of this study.

Data availability

Additional information on source data and instruction on how to label the files are available .

10) Figure legends and data quantification:

- the name of the statistical test used to generate error bars and P values,
- the number (n) of independent experiments (please specify technical or biological replicates) underlying each data point,
- the nature of the bars and error bars (s.d., s.e.m.)

- If the data are obtained from n {less than or equal to} 5, show the individual data points in addition to the SD or SEM.
- If the data are obtained from n {less than or equal to} 2, use scatter blots showing the individual data points.

11) Our journal encourages inclusion of *data citations in the reference list* to directly cite datasets that were re-used and obtained from public databases. Data citations in the article text are distinct from normal bibliographical citations and should directly link to the database records from which the data can be accessed. In the main text, data citations are formatted as follows: "Data ref: Smith et al, 2001" or "Data ref: NCBI Sequence Read Archive PRJNA342805, 2017". In the Reference list, data citations must be labeled with "[DATASET]". A data reference must provide the database name, accession number/identifiers and a resolvable link to the landing page from which the data can be accessed at the end of the reference. Further instructions are available at .

12) As part of the EMBO publication's Transparent Editorial Process, EMBO reports publishes online a Review Process File to accompany accepted manuscripts. This File will be published in conjunction with your paper and will include the referee reports, your point-by-point response and all pertinent correspondence relating to the manuscript.

I look forward to seeing a revised form of your manuscript when it is ready.

Yours sincerely,

Response to Review

We thank the reviewers for their insightful comments that have improved the strength of our manuscript. We have addressed the majority of the comments posed by the reviewers and have added substantial new data to this revision which are detailed below. Importantly, we address and confirm that the UPR activation can be directly attributed to UBXM1 through acute depletion and rescue experiments. Additionally, we link the increase in translation in UBXM1 KO cells to UPR activation by rescuing the increase in UPR activation and cell death in the UBXM1 KO cells to wildtype levels by inhibiting protein translation. We also extend the analysis of depleted mitochondrial proteins in UBXM1 deleted cells. We hope that the reviewers agree that the studies reported here demonstrate a new role for the p97 adaptor UBXM1 in ER-proteostasis and translation repression. Reviewer comments are reported in **bold**, and our responses are reported in blue.

Reviewer #1

In this manuscript, Ahlstedt et al. study UBXM1, an adaptor of the p97/VCP AAA ATPase, using a cell line deficient for UBXM1. They found that the knockout of UBXM1 activates ER stress and sensitizes cells to ER stress-induced cell death. They used a proteomic approach to analyze the change in the global proteome in UBXM1 knockout cells. Interestingly, they found many proteins are upregulated in UBXM1 knockout cells, which appears to be regulated at a post-transcriptional level. Using puromycin labeling, they found that protein translation appears to be upregulated in UBXM1 knockout cells.

Major comments:

The conclusions of the manuscript are generally well supported by experimental data, which are of high quality. The presentation is clear. In my opinion, a few issues need to be addressed to further strengthen their conclusions.

1. The authors need to express UBXM1 and mutants lacking either the UBX or UBA domain in UBXM1 knockout cells to test whether the ER stress phenotype (Figure 1) and the protein upregulation phenotype (Figure 5A-F) can be rescued. This would eliminate the possibility that the reported phenotypes are the off-target effects of CRISPR.

We now provide rescue data with wildtype and UBA or UBX point mutants (previously validated to lose interaction with ubiquitin and p97 (Ganji et al MCB 2018)). We measured *xbp1s* levels by real-time PCR and GFP-IRE1 α foci formation by immunofluorescence and imaging. We chose these studies as they are the most quantitative. We found that wildtype UBXM1 rescues the increase in GFP-IRE1 α foci formation (Figure 2B) and *xbp1s* expression (Figure 2C) to wildtype levels. In both cases, the UBX and UBA point mutants failed to rescue the UPR activation (Figure 2B and C) suggesting that an interaction with p97 and ubiquitin is important for the role of UBXM1 in suppressing UPR activation.

The studies in Figure 5A-F were completed in cells depleted of UBXM1 with siRNA, not the CRISPR knockout cells. Thus, it is unlikely an off-target effect of CRISPR. In any case, the protein upregulation phenotype can be rescued, with wildtype UBXM1 and the UBX point mutant but not the UBA point mutant (Figure 7). It is possible that some ER proteins may be degraded by p97-UBXM1 and thus loss of p97 interaction would contribute to UPR activation. Importantly, we find that translational repression by UBXM1 is an important mechanism that represses UPR activation

as inhibition of translation rescues UPR activation and consequent cell death in UBXN1 KO cells (Figure 8).

2. For Figure 2, please indicate whether the repeat is a biological replicate or a technical replicate from RT-PCR.

We apologize for the omission. The data from the RT-PCR studies in Figure 2 are biological duplicates – the figure legend has been edited to clarify this (Figure 2A).

3. In Figure 1A, the authors show that the knockout of UBXN1 causes an upregulation of phosphorylated eIF2alpha, which is known to suppress protein translation globally. In this regard, it is surprising to see the authors also concluded from Figure 7 that there is an upregulation of protein translation in UBXN1 knockout cells. The authors do not provide any explanation on how these seemingly contradictory phenotypes could be seen in the same cells.

While we don't fully understand this paradox, one possibility is that elevated ATF4-CHOP levels could result in pre-mature translation restart even in the face of ER stress. Several prior studies have demonstrated that elevated expression of ATF4-CHOP (as we observe in UBXN1 KO cells) activates a transcriptional program that restarts translation. This occurs through the upregulation of the phosphatase PPP1R15a that dephosphorylates eIF2 α , as well as aminoacyl tRNA synthetases and ribosomal subunits (Han, J., et al. *Nat Cell Biol* 2013). We are currently investigating this pathway but is presently beyond the scope of this study. We have added a detailed discussion in the revised text.

Minor Comments:

This work characterizes the cellular phenotypes associated with UBXN1 loss of function. The information reported here is important, but the biological significance is limited. This is mainly because the authors entirely rely on a genetic approach. While the reported phenotypes associated with UBXN1 deficiency is solid, it is unclear what the underlying mechanisms are. It is not clear whether or not these phenotypes are interconnected, nor is it clear whether UBXN1 is a direct regulator of these processes.

Taking the increased protein translation phenotype as an example, does this indicate UBXN1 is a translation suppressor for those ER-associated proteins?

We thank the reviewer for this comment. We are indeed very interested in determining whether UBXN1 specifically represses the translation of ER proteins. We are in the process of identifying proteins that are translated in UBXN1 null cells using O-propargyl-puromycin (OPP) labelling and mass spectrometry. However, given the timeframe for these studies, this cannot be accomplished in this revision.

Additionally, we have added data to connect the UPR activation to translation by rescuing the increase in UPR activation and cell death in the UBXN1 KO cells to wildtype levels by inhibiting protein translation (Figure 8).

How can UBXN1 selectively inhibit the translation of a subset of proteins?

Recent studies suggest that ribosome populations are quite heterogeneous, and ribosome associated proteins can help tune translation of select proteins. For example, pyruvate kinase muscle (PKM) associates with ER docked ribosomes to regulate the translation of ER proteins in particular (Simsek, D., et al. *Cell*. 2017). We find that UBXM1 is present on ER membranes (Figure 3G) and localizes to polysomes (Figure EV5F) and thus may regulate the translation of specific proteins. Studies are underway to test this hypothesis but are beyond the scope for this present study. We have included a more detailed explanation of this in our Discussion section.

Any evidence that UBXM1 is associated with translating ribosomes?

We thank the reviewer for this suggestion. We have added new data demonstrating that UBXM1 associates with 40S, 60S, and 80S ribosomal fractions as well as actively translating polysomes (Figure EV5F). This supports the hypothesis that UBXM1 may be physically repressing protein translation at the polysome. In agreement with our finding that the role of UBXM1 in repressing translation is independent of p97, p97 associates largely with the 40S, 60S, and 80S ribosomal fractions but not with the actively translating polysomes (Figure EV5F). We have initiated studies to identify the ribosome component that associates with UBXM1 by proximity proteomics.

Reviewer #2

UBXM1 is a VCP adaptor UBX domain protein which is known to be involved in elimination of ubiquitylated cytosolic proteins bound to the BAG6 complex. In this study, authors demonstrated that cells depleted of UBXM1 have elevated UPR activation, even without external ER stresses. Cells devoid of UBXM1 have significant and global up-regulation of UPR-specific target genes, and these cells are more sensitive to ER stress than their wildtype counterparts. Using quantitative tandem mass tag proteomics of UBXM1 deleted cells, authors found that significant enrichment of the abundance of ER proteins involved in protein translocation, protein folding, quality control, and the ER stress response in an ERAD-independent manner. Notably, they observed no change in the abundance of proteins in the cytosol or nucleus, and significant decrease in the expression of several mitochondrial proteins when UBXM1 was depleted. Authors further demonstrate that UBXM1 is a translation repressor, and its UBA domain is critical for suppressing protein synthesis. Thus, increased influx of proteins into the ER in UBXM1 KO cells causes UPR activation. Authors concluded that they have identified a new regulator of protein translation and ER proteostasis.

My specific comments were provided as follows.

Comments;

(1)

Authors found that significant enrichment of the ER proteins in UBXM1 KO cells, while there is no change in the abundance of proteins in the cytosol or nucleus. Mitochondrial proteins are even down-regulated in UBXM1 KO cells. I found these observations very interesting. However, I was frustrated that authors did not investigate the reason why such differences are associated in UBXM1-suppressed cells. Authors demonstrate that depletion of UBXM1 resulted in suppression of protein synthesis, but did not address whether ER proteins are specifically repressed by UBXM1 or it represses translation globally, as noted in their Discussion section. Do the mRNAs encoding signal sequence at the N-terminus of their products are specifically translated in UBXM1-suppressed cells? Do the translations of mRNAs encoding mitochondria translocation signals are

suppressed in UBXN1 KO cells? It should be possible to investigate these issues by using appropriate model ER- or mitochondrial proteins with or without specific signal sequences. Such kind of analysis should be necessary to support the claim of this manuscript.

In the revised manuscript, we expanded our mitochondrial protein expression study to include additional proteins (TOMM70, TIMM23, and TIMM17A) and found that they have significantly decreased expression when UBXN1 was depleted and were further depleted by ER stress (Figure 6C and D). Previous studies by Luke Wiseman's group showed that ER-stress induced PERK-pelF2 α activation resulted in two distinct events to protect mitochondrial integrity. First, mitochondrial hyperfusion prevents premature mitochondrial fragmentation, while in parallel the protective degradation of the TIMM17A subunit of the TIMM23 import complex decreases mitochondrial protein import. Indeed, we observe significant degradation of TIMM17A in UBXN1 depleted cells (Figure 6D) and demonstrate that mitochondria are significantly hyperfused in resting UBXN1 KO cells (Figure 6F) that is further enhanced upon ER-stress induction. This suggests that the ER stress experienced by UBXN1 KO cells is transmitted to mitochondria as a mechanism to preserve mitochondrial integrity. A more complete discussion on this topic is found in the results section of the revised manuscript.

In addition, we have now re-analyzed our TMT proteomics data to parse out proteins with ER-signal sequences and define the topology of ER proteins (Type 1, 2, multimembrane spanning and luminal proteins) and those with mitochondrial targeting sequences to provide a more comprehensive view of which proteins are increased or decreased in UBXN1 KO cells (Figure 3C). This analysis suggests that a wide cohort of ER clients including diverse membrane spanning and secreted factors are increased in UBXN1 KO cells.

Additionally, we are interested in determining whether UBXN1 specifically represses the translation of ER proteins. We are in the process of identifying proteins that are translated in UBXN1 null cells using O-propargyl-puromycin (OPP) labelling and mass spectrometry. However, given the timeframe for these studies, this cannot be accomplished in this revision.

(2)

Related to my previous comments, ER-targeted mRNAs are known to be degraded by a process termed RIDD in the case of ER stressed condition. Since the rapid degradation of mRNAs through RIDD functions to alleviate ER stress by preventing the continued influx of new polypeptides into the ER, I wondered why UBXN1 depletion greatly stimulates ER protein synthesis, escaping IRE1-dependent mRNA degradations. Does UBXN1 depletion suppress RIDD?

We measured the relative mRNA abundance of the *bona fide* RIDD targets *bloc1s1* and *cd59* by quantitative PCR in wildtype and UBXN1 KO cells. We quantified mRNA degradation in dithiothreitol (DTT) treated cells to induce RIDD. We observed increased degradation of both *bloc1s1* and *cd59* in UBXN1 KO cells, denoting an increase in the RIDD activity of IRE1 α (Figure 1H) which is expected given elevated IRE1 α activity in UBXN1 KO cells (Figure 1E-F). Currently, only a handful of mRNAs have been characterized as *bona fide* RIDD targets and does not include all ER-targeted mRNAs. Along with *xbp1*, RIDD target mRNAs are characterized by a stem loop structure containing a set of conserved residues that enhance their accessibility to the endoribonuclease domain of IRE1 α (Moore, K. et al. *Mol Biol Cell* 2015) . Thus, we do not expect

that all the potential ER-targeted mRNAs affected by UBXN1 loss would be degraded by RIDD before translation.

(3)

Authors mentioned that the elevated levels of ER proteins are not due to increased transcription of target genes. However, they only provided the quantification of prp transcript levels, which was unchanged between wildtype and UBXN1 KO cells. To support this important conclusion, it is necessary to provide whole transcriptome data to compare the expression levels of corresponding ER proteins (quantified by their proteomics data) and transcripts (quantified by, for an example, RNA-seq analysis).

We thank the reviewer for this comment. Since we already assessed ~80 UPR transcripts by real time PCR (Figure 2), we felt RNA-Seq would be of incremental advance to this study. However, the reviewer's point is well taken and instead we assessed transcript levels of additional ER proteins we used throughout our study. In our revision, we show that the mRNA levels of 5 out of the 6 genes (*agal*, *alpp2*, *scd1*, *trap α* , and *prp*) did not change between wildtype and UBXN1 loss of function cells (Figure EV4A).

(4)

Authors claimed that UBXN1 loss is detrimental to cell viability and have elevated levels of the apoptosis in the face of ER stress. However, authors did not examine apoptotic cell death in UBXN1 KO cells. They only provided evidence for defective proliferation of cells and transient induction of CHOP expression, but these are not enough to support the ER-stress induced apoptosis.

We previously demonstrated by crystal violet staining that UBXN1 KO cells have reduced growth in the face of ER stress (Figure 3A and B). We now provide additional quantitative data using CellTox Green Cytotoxicity assay and show that ER stress increases cell death in UBXN1 KO cells (Figure 3C). Increased CHOP expression (Figure 3D and E) suggests that this could be due to apoptosis, but we were unable to resolve cleaved caspase-3 in immunoblots in either the wildtype or UBXN1 KO over a 48-hour period. Given time constraints, we felt this was a minor issue and instead modified the language in the manuscript to reflect an increase in cell death (rather than apoptosis).

(5)

Authors showed that UBA domain of UBXN1 is critical for suppressing protein synthesis. Could you provide a bit more detailed discussion how UBA domain modulates protein translational events and promote expressions of ER-related proteins. Have you ever checked whether UBA domain of UBXN1 is necessary for suppressing UPR-specific target gene expressions?

We expressed the Myc-tagged UBA point mutant (used in translation rescue studies in Figure 7) into UBXN1 knockout (KO) cells as well as cells depleted of UBXN1 with siRNA and found that the UBA domain is necessary for suppressing IRE1 α clustering (Figure 2B) and *xbp1s* expression (Figure 2C). This suggests that an interaction with ubiquitin is important for the role of UBXN1 in maintaining ER-proteostasis. It is possible that UBXN1 is recruited to a ubiquitylated ribosomal protein to halt translation under certain circumstances. Several distinct ubiquitination events on ribosomal proteins regulate ribosome abundance via degradation by the 26S proteasome or

autophagy, ribosome disassembly through ribosome quality control pathways, and translation initiation on problematic mRNAs. It is possible UBXN1 could function to destabilize the ribosome to prevent translation of ER-targeted mRNAs and is an area of current study. We now include this in the discussion.

Reviewer #3

Summary:

Ahlstedt et al. investigate a new role for the p97 adapter protein UBXN1 in negatively regulating the ER unfolded protein response. The study starts from the observations that knockdown of UBXN1 in a previously generated HeLa cell line leads to induction of unfolded protein response markers, and the knockout cells display more pronounced UPR activation upon ER stress. This elevated UPR signaling renders the UBXN1 cells more prone to cell death. Global proteomics experiments similarly show an increased abundance of ER localized proteins, although it is not clearly delineated which of those are the result of UPR activation. The authors then probe the expression of two secretory client proteins, alpha-galactosidase (AGAL) and prion protein (PrP) and find that UBXN1 transient knockdown leads to ER accumulation of the two proteins and increased aggregation upon ER stress. The authors claim that degradation of these ER client proteins is unaffected by the UBXN1 knockdown, but accumulation may instead be due to increased protein translation. Indeed, they surprisingly find that UBXN1 knockout leads to constitutively elevated protein translation. This result points to a previously unknown role of UBXN1 in repressing protein synthesis. Complementation with UBXN1 mutants demonstrate that the translation repression is dependent on the ubiquitin binding activity of UBXN1 but that p97 is dispensable. Further investigation into the molecular mechanism for the translation repression remains reserved for a future manuscript.

Major comments:

1. My main reservation about the current manuscript is whether the UPR activation can be directly ascribed to the loss of UBXN1. The authors do not differentiate between acute depletion (through siRNA in Fig. 5) versus permanent UBXN1 knockout in most of the experiments. The latter may lead to extensive adaptation of the cellular proteome due to chronic stress. Prior studies from the authors have shown that UBXN1 knockout leads to loss of aggreasomes. This raises a major question whether the observed UPR activation can be directly attributed to UBXN1 loss or be an indirect result of adaptation in the knockout cells, for instance due to accumulation of BAG6 substrates in insoluble aggregates as the authors have shown previously (ref. 40). Along those lines, the authors already showed in the same study that UBXN1 knockout cells are more sensitive to proteotoxic stress.

We agree with the reviewer that cells can adapt to CRISPR knockout and therefore expanded all our major studies to include acute depletion of UBXN1 with siRNA. We demonstrate that siRNA-mediated depletion of UBXN1 results in a significant increase in BiP and ATF4 levels (Figure EV1G and H) as well as ATF6 N-terminal processing (Figure EV1I and J). In our first submission,

the IRE1 α clustering studies (Figure 1E and F) were completed in the context of acute depletion of UBXM1 by siRNA and demonstrate a significant increase in IRE1 α clustering when UBXM1 is depleted (Figure 1E and F). Additionally, we now provide new rescue experiments wherein re-expression of wildtype UBXM1 into UBXM1 loss of function cells (CRISPR KO and siRNA depletion) rescues IRE1 α clustering (Figure 2B) and *xbp1s* (Figure 2C) to wildtype levels. Similarly, in our first submission we show through rescue experiments that wildtype-UBXM1 expression can rescue the increase in puromycin incorporation to wildtype levels (Figure 7C and D). Furthermore, we depleted UBXM1 with siRNA alongside p97 depletion and measured puromycin incorporation into these cells. We found that acute depletion phenocopies UBXM1 KO in terms of increased puromycin incorporation into newly synthesized proteins (Figure EV5C and D). Therefore, we consider it unlikely that the phenotypes we observe are a result of long-term CRISPR adaptation.

2. The later results in the study nicely show that the repressed protein translation phenotype is dependent on the ubiquitin binding domain of UBXM1. These segregation-of-function mutants and complementation experiments could be easily used to more clearly distinguish whether the UPR activation can be directly attributed to UBXM1 and the increase in protein translation. For instance, can overexpression of UBXM1 in the knockout background suppress the UPR activation? Is the UBXM-domain mutant capable of suppressing the UPR phenotype? These results would provide critical support as to whether the UPR activation is a direct result of the loss of UBXM1.

We thank the reviewer for this suggestion. We now provide new data where we expressed wildtype UBXM1 and UBXM or UBA point mutants (previously validated to lose interaction with ubiquitin and p97 (Ganji et al MCB 2018)) into UBXM1 knockout (KO) cells or cells depleted of UBXM1 with siRNA. To assess UPR activation, we measured *xbp1s* expression levels by real-time PCR and GFP-IRE1 α foci formation by immunofluorescence imaging. We chose these studies as they are the most quantitative. We found that wildtype UBXM1 rescues the increase in GFP-IRE1 α foci formation (Figure 2B) and *xbp1s* expression (Figure 2C) to wildtype levels. In both cases, the UBXM and UBA point mutants failed to rescue the UPR activation (Figure 2B and C) suggesting that an interaction with p97 and ubiquitin is important for the role of UBXM1 in suppressing UPR activation. While we validated the turnover of several TMT hits and found no change in degradation in UBXM1 KO cells, it is possible that some ER proteins may be degraded in a p97-UBXM1 dependent manner and thus loss of p97 interaction would contribute to UPR activation. Nevertheless, in new data to delineate the relationship between UPR activation and protein translation, we halted protein synthesis with the translational elongation inhibitor cycloheximide and measured *xbp1s* expression by quantitative PCR. We found that inhibiting translation in UBXM1 KO cells rescued *xbp1s* expression back to wildtype levels (Figure 8A), suggesting that the increase in translation contributes to UPR activation. Supporting this, cycloheximide treatment rescues cell death in UBXM1 KO cells to wildtype levels (Figure 8B). This new data links the elevated translation to UPR activation in UBXM1 KO cells.

3. Similarly, the authors use transient siRNA knockdown of UBXM1 in Fig. 5 and Supp. Fig. 4, but do not reassess the UPR activation under these conditions. It would be important to validate that the acute UBXM1 knockdown can recapitulate the UPR activation phenotype.

Please see comment 1 above.

4. I am puzzled by the interpretation of the AGAL degradation experiments in

Supplemental Figure 4F. Clearly, the rate of AGAL degradation is much faster in WT cells than in UBXN1 knockout cells as indicated by the slope of the curves between 2-4 hours. I disagree with the interpretation that UBXN1 knockout does not impact AGAL turnover. It is not valid to make the comparison at 9 hours because hardly any AGAL substrate is remaining. Importantly, this experiment raises a larger question: Are other ER client degradation rates affected by the UBXN1 knockout? And is the UPR activation more generally due to accumulation of misfolded ER proteins? Their prior publication (ref. 40) evaluated several ERAD clients where UBXN1 was dispensable, but it could be possible that UBXN1 has a more specialized client pool. Showing quantification of the PrP CHX chase would also be helpful - from the single replicate it looks like more PrP remaining in the UBXN1 knockout at 8 hours (Supp. Figure 4G).

We agree with the reviewer that UBXN1 may have roles in regulating the degradation of select p97 ER clients. Determining this in an unbiased and comprehensive manner would require pulse chase SILAC proteomics or similar methodologies which are beyond the scope of the current study. Nevertheless, we evaluated the turnover of additional ER-resident proteins that we identified via TMT to be enriched in UBXN1 KO cells. We demonstrate that the ER-resident proteins SQLE and SCD1 are degraded at the same rate in control and UBXN1 depleted cells even though their abundance is increased in UBXN1 depleted cells (Figure 5H and I). Additionally, we show that the rate of PrP turnover is also unaffected by UBXN1 depletion (Figure EV4B and C). As noted by the reviewer our previous ERAD reporter study using three distinct ERAD clients routinely used to assess ERAD found no role for UBXN1 in ERAD (Ganji et al MCB 2018). In the revised manuscript, we have removed the AGAL turnover studies. Also, because TRAP α , and ALPP2 are long lived and not amenable to turnover studies these substrates have also been removed.

5. It would be helpful for the manuscript to clearly distinguish between 1) upregulation of ER proteostasis factors because of ER stress/UPR, and 2) upregulation of secreted clients (AGAL, PrP) which may be partly due to increased translation rates but could also be due to reduced degradation. Many of the hits from the proteomics experiments are ER proteostasis factors that are part of the adaptive stress response (SEC61B, SEC63, CANX, SSR1/2/3, STT3B, RPN1, RPN2, SEC61A1 - compare to ref 12: most are direct IRE1/XBP1s targets). Their increased expression does not lead to increased ER stress as they are involved in the resolution of ER stress. It appears to be circular logic that increased expression of UPR targets would lead to more UPR activation. Currently, the authors do not clearly disentangle the increased expression of endogenous ER proteins from the proteomics experiment versus overexpression of exogenous secreted clients.

In the revised manuscript, we re-formatted the TMT data to more comprehensively characterize the proteins that we identified by proteomics to be increased in abundance in UBXN1 KO cells. We distinguish between single- and multi-pass membrane proteins as well as secreted clients, many of which are not ER-resident proteins nor ER-stress responsive targets (Figure 3C). We also demonstrate an increase in expression of the ER-membrane resident proteins involved in cholesterol metabolism, SQLE and SCD1, in UBXN1 depleted cells and show that their turnover is not affected (Figure 5H and I). This analysis suggests that a wide cohort of ER clients (not limited to UPR transcript upregulation) that includes diverse membrane spanning and secreted factors are increased in UBXN1 KO cells. In the revised manuscript we clearly demarcate the difference between an increase of ER proteostasis factors in response to ER stress, and the upregulation of ER-targeted client proteins (Figure 3C).

6. The authors should tone down on broad generalizations, for instance in lines 306-309: ER aggregation was only observed for a single client protein (AGAL). Further, only a single mitochondrial protein was observed to be downregulated (TOMM20).

We have expanded the mitochondrial protein down regulation to include 5 proteins and observe significantly reduced protein expression in UBXN1 depleted cells relative to control (CYC1, TOMM20, TOMM70, TIMM23, TIMM17A) (Figure 5 A-D). Previous studies by Luke Wiseman's group showed that ER-stress induced PERK-peIF2 α activation resulted in two distinct events to protect mitochondrial integrity. First, mitochondrial hyperfusion prevents premature mitochondrial fragmentation, while in parallel the protective degradation of the TIMM17A subunit of the TIMM23 import complex decreases mitochondrial protein import. Indeed, we observe significant degradation of TIMM17A in UBXN1 depleted cells (Figure 6D) and demonstrate that mitochondria are significantly hyperfused in resting UBXN1 KO cells (Figure 6F) that is further enhanced upon ER-stress induction. This suggests that the ER stress experienced by UBXN1 KO cells is transmitted to mitochondria as a mechanism to preserve mitochondrial integrity.

In addition, in cases where we only study a single substrate (AGAL aggregation), we have changed the language of the revised manuscript to tone down any broad generalizations.

Minor comments

• Does UBXN1 localization to the ER/microsomes fraction depend on p97? What happens in UBX-domain mutant?

We tried to address this comment by expressing the domain mutants in UBXN1 KO cells and fractionating microsomes to look at their localization. In general, we observe decreased localization of the UBA but not the UBX mutant to microsomes. However due to technical difficulties in (a) expressing these constructs at identical levels from experiment to experiment and (b) fractionation efficiencies changing between experiments, it was difficult to unequivocally determine which domain was important for ER localization. For this reason, we have chosen not to include these studies in the revised manuscript.

• In Fig. 1A it is surprising that no BiP is detected at 0 hours as BiP is highly expressed even in the absence of ER stress. Can the authors comment on this discrepancy.

We now provide an alternative blot where the BiP levels at t=0 in control cells is more obvious (Figure 1A).

• The authors use different ER stressors interchangeably: DTT, Tunicamycin, Thapsigargin. While all results in UPR activation, they do so through different mechanisms and with slight nuances that may be worth considering for the experiments and interpretations.

We thank the reviewer for this comment and agree that these stressors can impact the ER and UPR activation in distinct ways. We included explanations in the methods section of our text that explains our rationale for alternating between the different ER stressors (DTT, tunicamycin, thapsigargin). For example, DTT is a severe stressor and can impact cell morphology therefore we used tunicamycin and thapsigargin for imaging studies. Importantly, we demonstrate through

the utilization of these various stressors that the UPR induction in UBXN1 null cells occurs irrespective of the mode of ER stress induction.

• Line 198: "Hierarchical clustering analysis demonstrates that the gene expression pattern observed in UBXN1 KO cells more closely resembles wildtype cells stressed with DTT than untreated wildtype cells based on similar log₂ fold change values (Figure 2)." Where is this clustering shown?

We apologize that this was not clear in the figure. We edited the figure to make the clustering more obvious (Figure 2A).

• What are the downregulated UPR genes in Fig. 2, and may this hold significance?

The reviewer points out an interesting observation. Many of the downregulated transcripts are ERAD components. The significance of this is presently unclear and we would require RNA-seq analysis to make a more educated conclusion. However, this finding may point to an environment that has a greater need to induce folding than degradative components. We included an expanded discussion about this in our results section regarding Figure 2A.

Dear Malavika,

Thank you for the submission of your revised manuscript to EMBO reports. As I already told you, we have now received the full set of referee reports that is copied below.

As you will see, the referees find that the study has been significantly improved during revision and referee 1 and 3 recommend publication after some remaining concerns have been addressed. Referee 2 asks for further control experiments to strengthen some of the data and conclusions, that I also ask you to address.

The further revised manuscript will be sent back to referee 2 for a final evaluation of the newly added data.

There are also a few things from the editorial side that need your attention:

- Please provide up to 5 keywords.
- Please update the references to the alphabetical Harvard style. The abbreviation 'et al' should be used if more than 10 authors. You can download the respective EndNote file from our Guide to Authors https://endnote.com/style_download/embo-reports/
- Please update the 'Conflict of interest' paragraph to our new 'Disclosure and competing interests statement'. For more information see <https://www.embopress.org/page/journal/14693178/authorguide#conflictsofinterest>
- Please remove the Author Contributions from the manuscript file and make sure that the author contributions in our online submission system are correct and up-to-date. The information you specified in the system will be automatically retrieved and typeset into the article. You can enter additional information in the free text box provided, if you wish.
- Please provide the callouts for Fig. EV4 panels in an alphabetical order
- Author Checklist: please update the manuscript number to the one assigned by EMBO Reports.
- Dataset EV legends: Supplementary Table 1 and 2 should be renamed to Dataset EV1-EV2 with the legends uploaded as a separate sheet in each Excel file, and appropriate callouts in the text.
- Source data: Please put all files for one Figure into one folder and then zip it. This means splitting the quantification in the .xls file into one file per figure panel and add this to the subfolders. E.g., the data for Fig. 1B, 1D, 1F, and 1G needs to be split and then put into the respective panel subfolders in the source data folder for Fig. 1
- The manuscript sections are in the wrong order. Please order them like this:
Title page - Abstract - Introduction - Results - Discussion - Materials and Methods - Acknowledgements - Disclosure and competing interests statement - References - Figure legends - Tables and their legends (not EV tables) - Expanded View Figure legends
- If you cite preprints (ref 75, 115) please add 'preprint' to the in-text citation. E.g., (preprint: Perea et al, 2022).
- I attach to this email a related manuscript file with comments by our data editors. Please address all comments and upload a revised file with tracked changes with your final manuscript submission.
- Finally, EMBO Reports papers are accompanied online by A) a short (1-2 sentences) summary of the findings and their significance, B) 2-3 bullet points highlighting key results and C) a synopsis image that is 550x300-600 pixels large (width x height) in PNG for JPG format. You can either show a model or key data in the synopsis image. Please note that the size is rather small and that text needs to be readable at the final size. Please send us this information along with the revised manuscript.
- On a different note, I would like to alert you that EMBO Press offers a new format for a video-synopsis of work published with us, which essentially is a short, author-generated film explaining the core findings in hand drawings, and, as we believe, can be very useful to increase visibility of the work. This has proven to offer a nice opportunity for exposure i.p. for the first author(s) of the study. Please see the following link for representative examples and their integration into the article web page:

<https://www.embopress.org/doi/full/10.15252/embo.2019103932>

Kind regards,
Martina

Referee #1:

Thank you to the authors for their extensive revisions to address our reviewer comments. This updated manuscript is now strengthened in several aspects by additional data:

- Important new complementation data with UBXN1 mutants in Fig. 2B-C clearly highlight that activity of both UBA and UBX domains are required for UPR activation. Hence, UBXN1 does not fully activate the UPR in an p97-independent manner as was even suggested in the title of the prior version (this part has now been removed). Later complementation assays showed that the translation repression by UBXN1 remains solely dependent on the UBA domain and independent of p97. However, the authors now adequately discuss that the loss of p97-UBXN1 dependent degradation may at least partially contribute to the UPR activation.
- New data shows that the UBXN1 knockout not only decreases mitochondrial protein levels but this also impacts mitochondrial hyperfusion.
- New data that UBXN1 is associated with polysomes, but not p97, strengthens the mechanistic link for a direct involvement of UBXN1 in regulating ribosome activity.

There are a few minor issues outlined below that should be easily addressable. After that, I am generally satisfied with the way the authors addressed my concerns and support publication:

- The authors now show new data (Fig. 5I) that degradation rates of two of ER clients from their proteomics dataset, SQLE and SCD1 are not affected by UBXN1 knockdown. However, AGAL degradation was slowed by UBXN1 in their prior version (previous Suppl. Fig. 4). The authors removed this data. In their response letter and in the discussion (lines 459-463), the authors outline a possible explanation that UBXN1 may still have a role in the degradation of select ER clients, such as AGAL, which could be the case. Therefore, the authors should add the AGAL data back into the manuscript.
- I was only able to find quantification source data for new Fig. 2B-C but not (representative) images. Check whether these are necessary to include.
- Line 269: is the p-value meant to be < 0.05 ?

Referee #2:

The revised manuscript by Ahlstedt and colleagues includes many new results to address the criticisms by the reviewers. Concerning my comments, I appreciate the efforts the authors have put in. Below are a few remaining questions that need to be addressed.

To exclude the potential off-target effect of Ubxn1 knockdown or knockout, the authors performed rescue experiments by putting back wild-type or mutant UBXN1. They showed in Figure 2B, C that only wild-type but not the mutants reduce GFP-IRE1 positive foci. It is unclear why the authors only did the rescue experiment in cells treated with the ER stress-inducer Tg. The same question also applies to Fig 2C, where the authors show that UBXN1 knockout reduces spliced Xbp1 after treatment with Tg. Since they could see a profound ER stress activation in UBXN1 knockout cells (Figure 1B, ATF4 induction and eIF2alpha phosphorylation), why don't they test whether UBXN1 could rescue these phenotypes?

From Figure 1, it seems that loss of Ubxn1 only selectively activates the ATF4 and PERK branches of UPR, whereas Xbp1 splicing is not affected unless cells are treated with an ER stressor. The model in Figure 8C cannot explain why certain UPR branches are selectively activated if the knockout of UBXN1 increases ER flow and, thus, misfolded proteins. Additionally, the phenotype description for Figure 1F, G needs to be clarified. It reads as if the knockout of Ubxn1 alone is sufficient to activate the IRE1 branch.

Figure EV5 shows that a fraction of UBXN1 appears co-sedimented with the polysome, but the lane labeled 40S is confusing.

There is not much RPS3 detected in that lane. The gel should include the ribosome-free fractions to show the relative amount of the proteins associated with the ribosome.

The new Figure 8 shows that Ubxn1 depletion-induced sensitivity to ER stress can be rescued by translation inhibition. The data needs to be more convincing for several reasons. First, it is unusual that Tg and CHX individual treatment each causes some degree of cytotoxicity in Ubxn1 KO cells, but the combined treatment was better tolerated. Second, the reported drug doses are quite high. Significant cytotoxicity is expected for individual treatment at these doses for 72 h, but the authors did not observe this. Is there an error in the reported amounts? This type of experiment would be more convincing if the authors could carefully choose multiple drug concentrations and combine them to reveal a rescue window. BTW, the authors should not normalize untreated Ubxn1 KO data to 1. They should report if there is any difference between WT and Ubxn1 KO cells.

Minor points:

Line 472 should be "ER-bound protein" or "translation of ER-bound RNA".

Referee #3:

EMBOR-2023-57115V2

"UBXN1 maintains ER proteostasis and represses UPR activation by modulating translation"
by Ahlstedt et al.

Comments to the Author

Authors have addressed most the concerns suggested by reviewers and added new appropriate data to improve the manuscript. They expanded their analyses to include additional mitochondrial and ER protein expressions and their additional experiments support the validity of their findings. I think that the manuscript is now suitable for publication.

We would like to thank all three reviewers for their comments and are gratified that they are appreciated the effort put into the revision. The first author has since graduated from the lab, and we have done our best to address the comments given her departure. We hope these revisions will be satisfactory to the reviewer. We address the remaining comments brought up by reviewer 2, Comments are in **bold** and responses are in blue.

Referee #1

The authors now show new data (Fig. 5I) that degradation rates of two of ER clients from their proteomics dataset, SQLE and SCD1 are not affected by UBXN1 knockdown. However, AGAL degradation was slowed by UBXN1 in their prior version (previous Suppl. Fig. 4). The authors removed this data. In their response letter and in the discussion (lines 459-463), the authors outline a possible explanation that UBXN1 may still have a role in the degradation of select ER clients, such as AGAL, which could be the case. Therefore, the authors should add the AGAL data back into the manuscript.

The AGAL data has now been added back (EV4 G and H) and included in the discussion.

I was only able to find quantification source data for new Fig. 2B-C but not (representative) images. Check whether these are necessary to include.

Source data has been included.

Line 269: is the p-value meant to be < 0.05 ?

Corrected

Referee #2

The revised manuscript by Ahlstedt and colleagues includes many new results to address the criticisms by the reviewers. Concerning my comments, I appreciate the efforts the authors have put in. Below are a few remaining questions that need to be addressed.

To exclude the potential off-target effect of Ubxn1 knockdown or knockout, the authors performed rescue experiments by putting back wild-type or mutant UBXN1. They showed in Figure 2B, C that only wild-type but not the mutants reduce GFP-IRE1 positive foci. It is unclear why the authors only did the rescue experiment in cells treated with the ER stress-inducer Tg. The same question also applies to Fig 2C, where the authors show that UBXN1 knockout reduces spliced Xbp1 after treatment with Tg.

We have attempted to use a variety of ER stressors (DTT, tunicamycin [Tu] and thapsigargin [Tg] throughout the manuscript to demonstrate the generality of the UBXN1 deletion phenotype. In the course of our studies, we found that the *xbp1s* qPCR assay produced the greatest signal to noise at low doses of Tg which is why we use this dose and drug for these studies. We have also performed this assay with Tu and observed similar results (however we do not have adequate replicates to report here). Further the qPCR screen depicted in Figure 2 used DTT and levels of *xbp1* were increased in UBXN1 KO cells.

Since they could see a profound ER stress activation in UBXN1 knockout cells (Figure 1B, ATF4 induction and eIF2alpha phosphorylation), why don't they test whether UBXN1 could rescue these phenotypes?

This is an excellent and relevant point and as we mentioned in the manuscript, we chose IRE1 foci formation and *xbp1s* qPCR as they are unbiased and quantitative. We are currently investigating how UBXM1 associates with polysomes and regulates translation. We are examining whether the ATF4 pathway contributes to the increased translation we observe in UBXM1 KO cells through induction of the protein phosphatase GADD34 that dephosphorylates eIF2a enabling translation to resume. Indeed, we find that GADD34 levels are dramatically elevated in UBXM1 KO cells, as are ribosomal and tRNA transcripts that are known targets of ATF4 (unpublished results). We now provide new data to show rescue of ATF6 processing by re-expression of wildtype UBXM1 in UBXM1 KO cells (New figure 2B). We are still working on the rescue studies with the UBA and UBX point mutants in eIF2a/ATF4 signaling and feel this is more pertinent to the new translation studies that are ongoing. We respectfully chose not to include it here.

From Figure 1, it seems that loss of *Ubx1* only selectively activates the ATF4 and PERK branches of UPR, whereas *Xbp1* splicing is not affected unless cells are treated with an ER stressor. The model in Figure 8C cannot explain why certain UPR branches are selectively activated if the knockout of UBXM1 increases ER flow and, thus, misfolded proteins.

As noted above, we are investigating the role of UBXM1 in regulating translation at the ribosome as well as through the eIF2a-ATF4 pathway. The basal increase in ATF4 induction in UBXM1 KO cells maybe due to a direct role for UBXM1 in translation regulation at the ribosome. However, delineating this mechanism will involve identifying how UBXM1 associates with the ribosome and how this in turn regulates translation. This is the focus of our efforts currently, but we have not made enough progress to report results here and is thus beyond the scope for this study. We have modified the model (Fig. 8C) and the discussion to reflect this.

Additionally, the phenotype description for Figure 1F, G needs to be clarified. It reads as if the knockout of *Ubx1* alone is sufficient to activate the IRE1 branch.

We apologize for this error, and have corrected the text appropriately to indicate that the increase in UBXM1 KO cells is after ER stress induction.

Figure EV5 shows that a fraction of UBXM1 appears co-sedimented with the polysome, but the lane labeled 40S is confusing. There is not much RPS3 detected in that lane. The gel should include the ribosome-free fractions to show the relative amount of the proteins associated with the ribosome.

We thank the reviewer for this comment. Upon closer inspection of the absorbance trace, this error can be attributed to the close proximity of the 40S and 60S peaks. We have now included the trace for clarity and re-labelled the lanes (EV5F). In the ribosome free fractions, the low amounts of ribosomal proteins relative the 40S, 60S, 80S and polysome fractions is apparent. We have also included the whole cell lysate (input) for these studies for comparison.

The new Figure 8 shows that *Ubx1* depletion-induced sensitivity to ER stress can be rescued by translation inhibition. The data needs to be more convincing for several reasons. First, it is unusual that Tg and CHX individual treatment each causes some degree of cytotoxicity in *Ubx1* KO cells, but the combined treatment was better tolerated.

Second, the reported drug doses are quite high. Significant cytotoxicity is expected for individual treatment at these doses for 72 h, but the authors did not observe this. Is there an error in the reported amounts? This type of experiment would be more convincing if the authors could carefully choose multiple drug concentrations and combine them to reveal a rescue window.

We now provide a graph in new EV6 that shows the cytotoxicity data without normalization. As can be seen UBXN1 KO cell viability is equivalent to wildtype cells without any perturbation. We normalized the data in the Figure 8 for ease in interpretation and have kept it as such. Previous studies have found that inhibiting protein synthesis in the face of ER stress alleviates cell death (Refs 62, 69 and 70), thus this is not altogether surprising. We have included text in the discussion referring to these previous studies. In Figure 3C we use multiple drug concentrations and find increased cytotoxicity in UBXN1 KO cells compared to wildtype even at lower doses of Tg (250nM). Furthermore, this dose of Tg is very low many published studies use concentrations as high as 20-30mM (PMID: 29467921, PMID: 21537829). The CHX concentration is a standard dose that is used by us and many other groups without overt toxicity (at least in HeLa cells as used here).

BTW, the authors should not normalize untreated Ubxn1 KO data to 1. They should report if there is any difference between WT and Ubxn1 KO cells.

New EV6 now shows the data without normalization. We kept the normalized data in figure 8 as it is easier to interpret.

Minor points:
Line 472 should be "ER-bound protein" or "translation of ER-bound RNA".
Corrected.

Manuscript number: EMBOR-2023-57115V3

Title: UBXN1 maintains ER proteostasis and represses UPR activation by modulating translation

Author(s): Malavika Raman, Brittany Ahlstedt, Rakesh Ganji, Sirisha Mukkavalli, Joao Paulo, and Steven Gygi

Dear Mali,

Thank you for your patience while we have reviewed your revised manuscript. It has been evaluated by former referee #2, who now also supports publication (see below). I am therefore writing with an 'accept in principle' decision, which means that I will be happy to accept your manuscript for publication once a few minor issues/corrections have been addressed, as follows.

- Please note that we can only typeset up to 5 EV figures. Maybe you can merge Fig. EV6 with another one, or alternatively move some figures to an Appendix? For the latter: it is a single .pdf file, with a table of content incl. page numbers. The nomenclature is Appendix Fig. S#.

- Reference list: et al needs to be used after 10 author names.

- Please shorten the abstract to 175 words. I also suggest removing the "(data available via ProteomeXchange with identifier PXD043686)" as the accession number is anyway given in the manuscript.

- I attach to this email a manuscript file with further comments. Some of the requests by our data editors in the figure legends have not been addressed and there are also some further edits, e.g. of preprint citations. Please address all comments and upload a revised file with tracked changes with your final manuscript submission.

- The synopsis image needs to be in jpeg, TIFF or png format and sized 550 pixels wide x 200-400 pixels high.

- We also need a draft for the summary text: A) a short (1-2 sentences) summary of the findings and their significance, B) 2-3 bullet points highlighting key results. Please send us this information along with the revised manuscript.

Once you have made these minor revisions, please use the following link to submit your corrected manuscript:

Link Not Available

If all remaining corrections have been attended to, you will then receive an official decision letter from the journal accepting your manuscript for publication in the next available issue of EMBO reports. This letter will also include details of the further steps you need to take for the prompt inclusion of your manuscript in our next available issue.

Thank you for your contribution to EMBO reports.

Best wishes,

Martina

Referee #2:

The authors have addressed my comments.

All editorial and formatting issues were resolved by the authors.

Dr. Malavika Raman
Tufts University School of Medicine
DMCB
150 Harrison Avenue
Boston, MA 02111
United States

Dear Mali,

I am very pleased to accept your manuscript for publication in the next available issue of EMBO Reports. Thank you for your contribution to our journal.

Kind regards,

Martina

Rev_Com_number: RC-2022-01803
New_manu_number: EMBOR-2023-57115V4
Corr_author: Raman
Title: UBXN1 maintains ER proteostasis and represses UPR activation by modulating translation